# THE GEOMETRY OF LLM QUANTIZATION: GPTQ AS BABAI'S NEAREST PLANE ALGORITHM

**Jiale Chen[1], Yalda Shabanzadeh[1], Elvir Crnčević[2], Torsten Hoefler[3], Dan Alistarh[1,2]**
[1]Institute of Science and Technology Austria (ISTA), [2]Red Hat, Inc., [3]ETH Zürich
`jiale.chen@ist.ac.at`

## ABSTRACT

Quantizing the weights of large language models (LLMs) from 16-bit to lower bitwidth is the de facto approach to deploy massive transformers onto more affordable accelerators. While GPTQ emerged as one of the standard methods for one-shot post-training quantization at LLM scale, its inner workings are described as a sequence of algebraic updates that obscure geometric meaning or worst-case guarantees. In this work, we show that, when executed back-to-front (from the last to first dimension) for a linear layer, GPTQ is mathematically identical to Babai's nearest plane algorithm for the classical closest vector problem (CVP) on a lattice defined by the Hessian matrix of the layer's inputs. This equivalence is based on a sophisticated mathematical argument, and has two analytical consequences: first, the GPTQ error propagation step gains an intuitive geometric interpretation; second, GPTQ inherits the error upper bound of Babai's algorithm under the assumption that no weights are clipped. Leveraging this bound, we design post-training quantization methods that avoid clipping, and outperform the original GPTQ. In addition, we provide efficient GPU inference kernels for the resulting representation. Taken together, these results place GPTQ on a firm theoretical footing and open the door to importing decades of progress in lattice algorithms towards the design of future quantization algorithms for billion-parameter models. Source code is available at `https://github.com/IST-DASLab/GPTQ-Babai`.

## 1 INTRODUCTION

Generative pre-trained transformers (GPT) models contain hundreds of billions of parameters and have massive computational and memory costs (Luccioni et al., 2024). Post-training quantization (PTQ) has emerged as a practical solution for reducing their footprint (Gholami et al., 2021). Among a growing family of methods, GPTQ (Frantar et al., 2023) was the first to push one-shot quantization down to the 4-bit regime, while retaining near-baseline accuracies. GPTQ is still very popular nowadays and yields state-of-the-art results in some regimes (Kurtic et al., 2025).

Despite its empirical success, the GPTQ algorithm was only presented as a sequence of greedily applied algebraic operations: the procedure picks one weight at a time, quantizes it via rounding or clipping, and then optimally updates the not-yet-quantized weights to correct for the remaining per-layer loss; it then continues with the next weight, and so on. This procedure leaves an obvious open question: why does a local greedy rule work so well globally? Current literature does not answer this question, leaving little guidance for principled extensions or failure case analysis.

**Our contribution.** This paper is the first[1] to provide a geometric interpretation for GPTQ, which implies a layer-wise global error bound. Our main theoretical results (Section 4) are (i) the GPTQ optimization problem, i.e., linear-layer quantization with the L2 objective on the output, is equivalent to the closest vector problem (CVP) w.r.t. L2 distance; (ii) the GPTQ algorithm executed from the last to first dimension is the same as Babai's nearest plane algorithm on the basis of the factorized Hessian matrix, without LLL basis reduction, and this finding holds independently of whether large weights are clipped to the quantization grid (a process known as weight clipping); and (iii) the worst-case layer-wise error in the no-clipping setting is bound tightly by the trace of the diagonal matrix of the

---

[1]The concurrent work of Birnick (2025) appeared on arXiv slightly later than our preprint (Chen et al., 2025).

LDL decomposition of the Hessian matrix. In addition (Section 5), we tie our theoretical findings to practical quantization by introducing new no-clipping methods of better accuracy than the original GPTQ, together with efficient GPU inference kernels for the resulting representation.

## 2    RELATED WORK

**Second-order compression (pruning and quantization).** The idea of using Hessian information to guide parameter removal dates back to Optimal Brain Damage (LeCun et al., 1989) and Optimal Brain Surgeon (OBS) (Hassibi et al., 1993). Optimal Brain Compression (OBC) (Frantar & Alistarh, 2022) generalizes OBS to the post-training setting and unifies structured pruning and quantization (also called Optimal Brain Quantizer, OBQ) under a single exact solver. GPTQ (Frantar et al., 2023) inherits OBQ's error propagation method but applies it in a fixed order, so that the inverse Hessian can be shared and only needs to be computed once. GPTQ only has cubic computational complexity in the column/row dimension, making it suitable for LLMs. QuIP (Chee et al., 2023) proves an error guarantee for GPTQ and proposes the LDLQ method as an equivalent variant of GPTQ.

**Lattices, CVP algorithms, and hardness.** The closest vector problem (CVP) is NP-complete to approximate within any constant factor under polynomial-time reductions (van Emde Boas, 1981; Micciancio & Goldwasser, 2002; Dinur et al., 2003), motivating decades of approximation algorithms. Babai's nearest plane heuristic (Babai, 1986) delivers a solution in polynomial time and, when preceded by LLL basis reduction (Lenstra et al., 1982), enjoys a $2^{O(n)}$ approximation. BKZ basis reduction (Kannan, 1987) further tightens the constant in an exponential-time solver.

## 3    PRELIMINARIES AND NOTATIONS

We use Python-style indexing inside square brackets to select elements and sub-matrices from a tensor, e.g., $[j, :]$ selects the $j$-th row vector, $[:, j]$ selects the $j$-th column vector, and $[j :, j]$ selects the sub-column consisting of rows after the $j$-th (inclusive) row in the $j$-th column, $[:, J]$ selects the column vectors indexed by set $J$ as a sub-matrix, etc[2].

### 3.1    LINEAR-LAYER QUANTIZATION PROBLEM

**Problem.** Let $\boldsymbol{X} = [\boldsymbol{x}_1, \ldots, \boldsymbol{x}_n]^\top \in \mathbb{R}^{n \times c}$ be the sampled calibration input data of batch size $n$ and input dimension $c$ with $\boldsymbol{x}_i \in \mathbb{R}^c$ and $n \geq c = \mathrm{rank}\,(\boldsymbol{X})$. Let $\boldsymbol{W} = [\boldsymbol{w}_1, \ldots, \boldsymbol{w}_r] \in \mathbb{R}^{c \times r}$ be the linear layer weights of input dimension $c$ and output dimension $r$ with $\boldsymbol{w}_i \in \mathbb{R}^c$. Let $\boldsymbol{S} = [\boldsymbol{s}_1, \ldots, \boldsymbol{s}_r] \in \mathbb{R}_{\neq 0}^{c \times r}$ be the non-zero quantization scales with $\boldsymbol{s}_i \in \mathbb{R}_{\neq 0}^c$. Here we consider a general case that applies to any grouping pattern: each weight element $\boldsymbol{w}_i[j]$ has its own scaling factor $\boldsymbol{s}_i[j]$. Assume $\boldsymbol{S}$ is statically computed using methods like AbsMax or MSE before any weight updates. Let $\mathbb{Z}_\dagger \subseteq \mathbb{Z}$ be the quantization grid (representable integers). In the clipping setting, e.g., for INT4 format, $\mathbb{Z}_\dagger = \{-8, \ldots, -1, 0, 1, \ldots, 7\}$. In the no-clipping setting, $\mathbb{Z}_\dagger = \mathbb{Z}$, which allows any integer as the quantization results. Let $\boldsymbol{Z} = [\boldsymbol{z}_1, \ldots, \boldsymbol{z}_r] \in \mathbb{Z}_\dagger^{c \times r}$ be the (unknown) quantized integers with $\boldsymbol{z}_i \in \mathbb{Z}_\dagger^c$. Denote $\boldsymbol{Q} = [\boldsymbol{q}_1, \ldots, \boldsymbol{q}_r] \in \mathbb{R}^{c \times r}$ as the dequantized weights with $\boldsymbol{q}_i = \mathrm{diag}\,(\boldsymbol{s}_i)\,\boldsymbol{z}_i \in \mathbb{R}^c$. The goal is to minimize the L2 error on the layer output $\boldsymbol{X}\boldsymbol{W} \in \mathbb{R}^{n \times r}$: $\|\boldsymbol{X}\boldsymbol{Q} - \boldsymbol{X}\boldsymbol{W}\|_{\mathrm{F}}^2 = \sum_{i=1}^r \|\boldsymbol{X}\,\mathrm{diag}\,(\boldsymbol{s}_i)\,\boldsymbol{z}_i - \boldsymbol{X}\boldsymbol{w}_i\|^2$, i.e, finding $\mathrm{argmin}_{\boldsymbol{z}_i \in \mathbb{Z}_\dagger^c} \|\boldsymbol{X}\,\mathrm{diag}\,(\boldsymbol{s}_i)\,\boldsymbol{z}_i - \boldsymbol{X}\boldsymbol{w}_i\|^2$ for all $1 \leq i \leq r$.

**OBQ algorithm.** Let set $J_i$ initialized to $\{1, \ldots, c\}$ be the set of not-yet-quantized indices of $\boldsymbol{w}_i$. We denote $J_i$ as $J$ as a short-hand notation. For each weight vector $\boldsymbol{w}_i$, OBQ chooses

$$j \leftarrow \mathrm{argmin}_{j \in J} \frac{(\boldsymbol{q}_i[j] - \boldsymbol{w}_i[j])^2}{(\boldsymbol{X}[:, J]^\top \boldsymbol{X}[:, J])^{-1}\,[j, j]} \qquad (1)$$

as the next dimension to quantize. OBQ quantizes the chosen element $\boldsymbol{w}_i[j]$ as $\boldsymbol{q}_i[j] \leftarrow \boldsymbol{s}_i[j] \cdot \mathrm{ROUND}\left(\frac{\boldsymbol{w}_i[j]}{\boldsymbol{s}_i[j]}, \mathbb{Z}_\dagger\right)$ via the $\mathrm{ROUND}\,(\cdot, \mathbb{Z}_\dagger)$ function which rounds the inputs to the nearest values

---

[2]For more details, please see (NumPy): `https://numpy.org/doc/stable/user/basics.ind exing.html`

in $\mathbb{Z}_\dagger$. OBQ then optimally updates the subset of weights $\boldsymbol{w}_i[J]$ via an error propagation step $\boldsymbol{w}_i[j'] \leftarrow \boldsymbol{w}_i[j'] + \Delta\boldsymbol{w}_i[j']$ for all $j' \in J$ with

$$\Delta\boldsymbol{w}_i[j'] \leftarrow \frac{\left(\boldsymbol{X}[:,J]^\top \boldsymbol{X}[:,J]\right)^{-1}[j',j]}{\left(\boldsymbol{X}[:,J]^\top \boldsymbol{X}[:,J]\right)^{-1}[j,j]}\left(\boldsymbol{q}_i[j] - \boldsymbol{w}_i[j]\right). \tag{2}$$

OBQ continues iteration with $J \leftarrow J \setminus \{j\}$ until $J$ is empty.

**GPTQ algorithm.** GPTQ reduces the computational complexity of OBQ by applying the OBQ quantization and error propagation steps in a fixed dimensional order, e.g., from the first to last dimension ($j \leftarrow 1$ to $c$), instead of dynamically determined orders (Eq. 1). The fixed order is independent of the output channel $i$, thus the Hessian information $\left(\boldsymbol{X}[:,J]^\top \boldsymbol{X}[:,J]\right)^{-1}[:,j]$ can be shared across $\boldsymbol{w}_i$ for all $i$, without recomputation. Furthermore, the Hessian information for all $j$ can be precomputed at once using Cholesky or LDL decomposition of the Hessian matrix $\boldsymbol{X}^\top \boldsymbol{X}$.

Algorithm 1 is the pseudocode of GPTQ. The algorithm is identical to the original GPTQ paper (Frantar et al., 2023) except for missing the blocking mechanism that only affects the memory access pattern and computational speed, but not the numerical results. Additional notations are as follows. $\boldsymbol{P} \in \{0,1\}^{c \times c}$ is a permutation matrix that modifies the dimensional order of GPTQ quantization. The default order is front-to-back (from the first to last dimension), i.e., $\boldsymbol{P} = \mathbf{I}$. $\lambda \in \mathbb{R}_+$ is a small damping factor for computing the Hessian matrix, ensuring the matrix is of full rank. A typical choice is $\lambda = \frac{1}{100c}\sum_{j=1}^c \left(\boldsymbol{X}^\top \boldsymbol{X}\right)[j,j] = \frac{1}{100c}\|\boldsymbol{X}\|_\mathrm{F}^2$. Function LDL returns the lower triangular matrix in the LDL decomposition. Symbols $*$ and $/$ denote element-wise multiplication and division, respectively.

---

**Algorithm 1:** GPTQ

    **Input:** original weights $\boldsymbol{W} \in \mathbb{R}^{c \times r}$, per-coordinate scales $\boldsymbol{S} \in \mathbb{R}_{\neq 0}^{c \times r}$, calibration activation
        $\boldsymbol{X} \in \mathbb{R}^{n \times c}$, permutation $\boldsymbol{P} \in \{0,1\}^{c \times c}$, damping ratio $\lambda > 0$, integer grid $\mathbb{Z}_\dagger \subseteq \mathbb{Z}$
    **Output:** quantized weights $\boldsymbol{Z} \in \mathbb{Z}_\dagger^{c \times r}$, dequantized weights $\boldsymbol{Q} \in \mathbb{R}^{c \times r}$

1   $\boldsymbol{H} \leftarrow \boldsymbol{P}^\top \left(\boldsymbol{X}^\top \boldsymbol{X} + \lambda \mathbf{I}\right) \boldsymbol{P}$ // dampen and reorder Hessian
2   $\boldsymbol{L} \leftarrow \text{LDL}\left(\boldsymbol{H}^{-1}\right)$ // factorize (take the L matrix from the LDL decomposition) the inversed
       Hessian as the shared coefficients for error propagation
3   $\boldsymbol{W}, \boldsymbol{S} \leftarrow \boldsymbol{P}^{-1}\boldsymbol{W}, \boldsymbol{P}^{-1}\boldsymbol{S}$ // reorder weights and scales
4   $\boldsymbol{Q}, \boldsymbol{Z} \leftarrow \boldsymbol{W}, \boldsymbol{0}$ // initialize dequantized and quantized weights
5   **for** $j \leftarrow 1$ to $c$ **do**
6       $\boldsymbol{\zeta} \leftarrow \boldsymbol{W}[j,:]/\boldsymbol{S}[j,:]$ // element-wise divide current row by its scales
7       $\boldsymbol{Z}[j,:] \leftarrow \text{ROUND}\left(\boldsymbol{\zeta}, \mathbb{Z}_\dagger\right)$ // quantize coefficients to the target grid
8       $\boldsymbol{Q}[j,:] \leftarrow \boldsymbol{Z}[j,:] * \boldsymbol{S}[j,:]$ // dequantize current row back to weight space
9       $\varepsilon \leftarrow \boldsymbol{Q}[j,:] - \boldsymbol{W}[j,:]$ // quantization error for current row
10     $\boldsymbol{W}[j:,:] \leftarrow \boldsymbol{W}[j:,:] + \boldsymbol{L}[j:,j]\varepsilon$ // propagate error to not-yet-quantized rows; broadcast
        over columns
11 **end**
12 $\boldsymbol{Z}, \boldsymbol{Q} \leftarrow \boldsymbol{P}\boldsymbol{Z}, \boldsymbol{P}\boldsymbol{Q}$ // undo reorder to restore original input order; return integers and
    dequantized weights

---

## 3.2 THE CLOSEST VECTOR PROBLEM (CVP)

**Problem.** Let $\boldsymbol{B} = [\boldsymbol{b}_1, \dots, \boldsymbol{b}_c] \in \mathbb{R}^{n \times c}$ be a set of $c$ basis vectors of dimension $n$ with $\boldsymbol{b}_j \in \mathbb{R}^n$ and $n \geq c = \text{rank}(\boldsymbol{B})$. Let $\boldsymbol{y} \in \mathbb{R}^n$ be an external target vector to approximate. Let $\boldsymbol{z} \in \mathbb{Z}^c$ be the (unknown) integer vector representing the basis combinations of the lattice vector. The goal is to find the vector on the lattice defined by the basis $\boldsymbol{B}$ that is closest to the target vector $\boldsymbol{y}$, i.e., finding $\arg\min_{\boldsymbol{z} \in \mathbb{Z}^c} \|\boldsymbol{B}\boldsymbol{z} - \boldsymbol{y}\|^2$. A visualization of a two-dimensional CVP is shown in Figure 1 (a).

**Babai's nearest plane algorithm.** Babai's algorithm iteratively projects the target vector onto the nearest hyperplane of an LLL-reduced lattice and rounds the corresponding coefficient. Figure 1 (b) visualizes the basis reduction step and Figure 1 (c-d) visualize the projection steps.

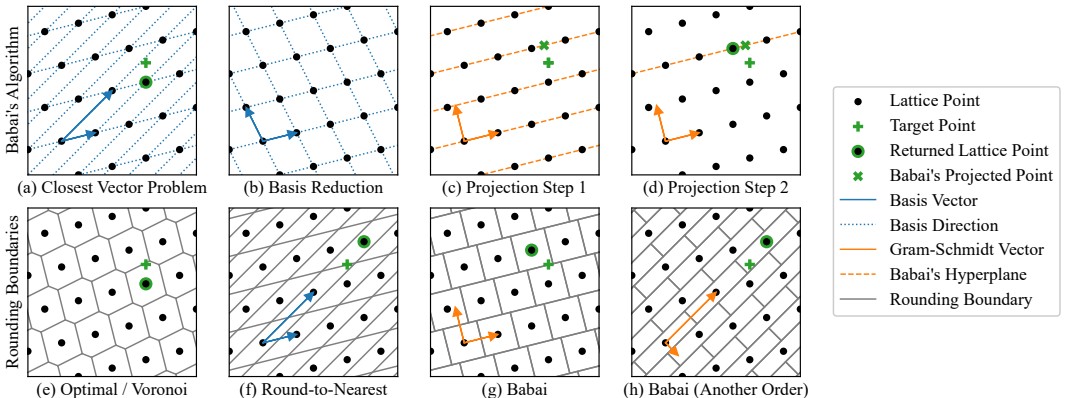

Figure 1: **Top row:** (a) CVP in a two-dimensional lattice; (b) Basis reduction can find a shorter, more orthogonal basis that can potentially improve the results; (c-d) The projection steps in Babai's nearest plane algorithm. **Bottom row:** rounding boundaries of (e) optimal rounding or Voronoi cells; (f) round-to-nearest (RTN); (g) Babai's nearest plane algorithm without basis reduction; (h) Babai's algorithm without basis reduction under the reversely ordered basis.

Algorithm 2 is the pseudocode of Babai's nearest plane algorithm to solve CVP. For better computational efficiency, the pseudocode uses a conceptually equivalent approach. Instead of projecting the target vector onto the nearest hyperplane, it moves the target vector along the basis direction towards the hyperplane where the origin lies. The projection error is retained in the updated target vector as it is orthogonal to the hyperplane and will not affect subsequent projections. Additional notations are as follows. Function LLL returns the transformation matrix of the LLL reduction with the parameter delta defaulting to $\frac{3}{4}$. Function QR returns the orthogonal matrix in QR decomposition, which is the same as the normalized Gram-Schmidt orthogonalization process. $\langle \cdot, \cdot \rangle$ denotes the vector dot product. Function ROUND is defined as in the GPTQ algorithm.

---

**Algorithm 2:** Babai's Nearest Plane

---

**Input:** lattice basis (column vectors) $\boldsymbol{B} \in \mathbb{R}^{n \times c}$, target vector $\boldsymbol{y} \in \mathbb{R}^n$
**Output:** closest lattice vector's basis coefficients $\boldsymbol{z} \in \mathbb{Z}^c$

1 $\boldsymbol{T} \leftarrow \mathrm{LLL}\,(\boldsymbol{B})$ // unimodular transformation matrix from LLL basis reduction
2 $\boldsymbol{A} \leftarrow \boldsymbol{BT}$ // reduce the basis
3 $\boldsymbol{\Phi} \leftarrow \mathrm{QR}\,(\boldsymbol{A})$ // normalized Gram-Schmidt process (take the Q matrix from the QR decomposition)
4 $\boldsymbol{y}', \boldsymbol{z} \leftarrow \boldsymbol{y}, \boldsymbol{0}$ // initialize residual target and integer solution in reduced basis
5 **for** $j \leftarrow c$ to $1$ **do**
6     $\zeta \leftarrow \langle \boldsymbol{\Phi}[:,j], \boldsymbol{y}' \rangle \,/\, \langle \boldsymbol{\Phi}[:,j], \boldsymbol{A}[:,j] \rangle$ // exact coefficient along the unnormalized Gram-Schmidt vector; ratio between the projections of residual and the reduced basis on the Gram-Schmidt direction
7     $\boldsymbol{z}[j] \leftarrow \mathrm{ROUND}\,(\zeta, \mathbb{Z})$ // round to the nearest plane
8     $\boldsymbol{y}' \leftarrow \boldsymbol{y}' - \boldsymbol{A}[:,j]\boldsymbol{z}[j]$ // update the residual
9 **end**
10 $\boldsymbol{z} \leftarrow \boldsymbol{T}\boldsymbol{z}$ // map integer solution back to the original basis and return

---

**Babai's error bound.** Figure 1 shows the rounding boundaries of the optimal (e), round-to-nearest (RTN) (f), and Babai's algorithm without basis reduction (g-h). Compared to RTN, Babai's algorithm generates rectangular partitions and thus has a smaller worst-case error. The error bound has been proven in Babai (1986). Formally, let $\boldsymbol{\Phi} = [\boldsymbol{\phi}_1, \ldots, \boldsymbol{\phi}_c]$ be the set of normalized Gram-Schmidt vectors of the LLL-reduced basis $\boldsymbol{A} = [\boldsymbol{a}_1, \ldots, \boldsymbol{a}_c]$. Let $\tilde{\boldsymbol{A}} = [\tilde{\boldsymbol{a}}_1, \ldots, \tilde{\boldsymbol{a}}_c]$ denote the unnormalized Gram-Schmidt vectors with $\tilde{\boldsymbol{a}}_j = \langle \boldsymbol{\phi}_j, \boldsymbol{a}_j \rangle \boldsymbol{\phi}_j$. At iteration $j$, the algorithm replaces the exact coefficient $\zeta$ with the closest integer, so the deviation satisfies $|\zeta - \boldsymbol{z}[j]| \leq \frac{1}{2}$. Hence, the error component along $\tilde{\boldsymbol{a}}_j$ has norm at most $\frac{1}{2}\|\tilde{\boldsymbol{a}}_j\|$. Because the $\tilde{\boldsymbol{A}}$ is orthogonal, these error components

add in Euclidean norm, giving a bound on the residual (error) vector $\boldsymbol{y}'$: $\|\boldsymbol{y}'\|^2 \leq \frac{1}{4}\sum_{j=1}^{c}\|\tilde{\boldsymbol{a}}_j\|^2 = \frac{1}{4}\sum_{j=1}^{c}\langle\boldsymbol{\phi}_j,\boldsymbol{a}_j\rangle^2$. Babai's algorithm guarantees to return the center vector of the hyper-cuboid (Figure 1 (g)) constructed by the unnormalized Gram-Schmidt vectors $\tilde{\boldsymbol{A}}$ where the target $\boldsymbol{y}$ is located. Equality is attained when the target $\boldsymbol{y}$ lies at the corner of the hyper-cuboid, so the bound is tight. Babai (1986) additionally proved a relative error bound for $\gamma$ with $\|\boldsymbol{B}\boldsymbol{z}-\boldsymbol{y}\| \leq \gamma\cdot\min_{\boldsymbol{z}'\in\mathbb{Z}^c}\|\boldsymbol{B}\boldsymbol{z}'-\boldsymbol{y}\|$.
The bound is $1 \leq \gamma \leq \sqrt{1+\max_{1\leq j\leq c}\frac{\sum_{j'=1}^{j}\|\tilde{\boldsymbol{a}}_{j'}\|^2}{\|\tilde{\boldsymbol{a}}_j\|^2}} \leq \sqrt{c+1}\cdot\max_{1\leq j'\leq j\leq c}\frac{\|\tilde{\boldsymbol{a}}_{j'}\|}{\|\tilde{\boldsymbol{a}}_j\|}$.

# 4 Theoretical Results

We first show that weight quantization is an instance of the classical closest vector problem (CVP) in Section 4.1, which allows us to work in a lattice defined by the Hessian. We then reinterpret OBQ's, equivalently GPTQ's, error propagation step as a nearest hyperplane projection in Section 4.2, establishing our main equivalence in Section 4.3: GPTQ, running back-to-front, coincides exactly with Babai's nearest plane algorithm. This equivalence allows us to import Babai's guarantees to obtain a tight, layer-wise error bound in the no-clipping setting in Section 4.4. Finally, we analyze how quantization order influences this bound in Section 4.5.

## 4.1 Equivalence Between L2 Quantization and CVP

A quantization problem with the L2 objective $\operatorname{argmin}_{\boldsymbol{z}_i\in\mathbb{Z}_{\dagger}^c}\|\boldsymbol{X}\operatorname{diag}(\boldsymbol{s}_i)\boldsymbol{z}_i - \boldsymbol{X}\boldsymbol{w}_i\|^2$ and a CVP with the L2 distance $\operatorname{argmin}_{\boldsymbol{z}\in\mathbb{Z}^c}\|\boldsymbol{B}\boldsymbol{z}-\boldsymbol{y}\|^2$ share the same solution ($\boldsymbol{z} = \boldsymbol{z}_i$) whenever the structural conditions $\boldsymbol{B} = \boldsymbol{X}\operatorname{diag}(\boldsymbol{s}_i)$ and $\boldsymbol{y} = \boldsymbol{X}\boldsymbol{w}_i$ hold and the solution domain matches. To ensure the solution domain matches, we can either disable the clipping in the quantization setup (setting $\mathbb{Z}_{\dagger} = \mathbb{Z}$) or enable the clipping in the CVP setup (making $\boldsymbol{z}\in\mathbb{Z}_{\dagger}^c$). Table 1 is a take-away dictionary showing the correspondence between the quantization and CVP concepts.

Table 1: Quantization-CVP dictionary for the output channel $i$.

| Quantization symbol | CVP interpretation |
|---|---|
| Input activation $\boldsymbol{X}\in\mathbb{R}^{n\times c}$ | Basis directions (columns are generators) |
| Scale $\boldsymbol{s}_i\in\mathbb{R}_{\neq 0}^c$ | Basis stretches |
| $\boldsymbol{B}_{(i)} = \boldsymbol{X}\operatorname{diag}(\boldsymbol{s}_i)\in\mathbb{R}^{n\times c}$ | Lattice basis (columns are generators) |
| Weight $\boldsymbol{w}_i\in\mathbb{R}^c$ | Floating-point coordinates on the unstretched basis |
| Integer weight representation $\boldsymbol{z}_i\in\mathbb{Z}_{\dagger}^c$ | Integer coordinates on the lattice basis |
| Dequantized weight $\boldsymbol{q}_i = \operatorname{diag}(\boldsymbol{s}_i)\boldsymbol{z}_i\in\mathbb{R}^c$ | Dequantized coordinates on the unstretched basis |
| Target output activation $\boldsymbol{y}_{(i)} = \boldsymbol{X}\boldsymbol{w}_i\in\mathbb{R}^n$ | External target vector to approximate |

We can introduce a factor of the Hessian matrix, $\boldsymbol{\mathcal{X}} = [\boldsymbol{\chi}_1,\ldots,\boldsymbol{\chi}_c]$ with $\boldsymbol{X}^\top\boldsymbol{X} = \boldsymbol{\mathcal{X}}^\top\boldsymbol{\mathcal{X}}$. The loss can then be reformulated as $\|\boldsymbol{\mathcal{X}}\operatorname{diag}(\boldsymbol{s}_i)\boldsymbol{z}_i - \boldsymbol{\mathcal{X}}\boldsymbol{w}_i\|^2$.

**Theorem 1 (Quantization and CVP)** *The CVPs using any possible factors $\boldsymbol{\mathcal{X}}$ of the Hessian matrix $\boldsymbol{X}^\top\boldsymbol{X}$ are equivalent under an orthogonal transformation (rotation and reflection) of the lattice and external target vector.*

**Proof** Let $\boldsymbol{\mathcal{X}}$ and $\boldsymbol{\mathcal{X}}'$ be two possible factors of the Hessian matrix with $\boldsymbol{\mathcal{X}}^\top\boldsymbol{\mathcal{X}} = \boldsymbol{\mathcal{X}}'^\top\boldsymbol{\mathcal{X}}'$. The inner products $\langle\boldsymbol{\chi}_{j_1},\boldsymbol{\chi}_{j_2}\rangle$ and $\langle\boldsymbol{\chi}'_{j_1},\boldsymbol{\chi}'_{j_2}\rangle$ must be equal for all $1\leq j_1,j_2\leq c$. In other words, the lengths $\|\boldsymbol{\chi}_{j_1}\| = \|\boldsymbol{\chi}'_{j_1}\|$, and the angles $\angle(\boldsymbol{\chi}_{j_1},\boldsymbol{\chi}_{j_2}) = \angle(\boldsymbol{\chi}'_{j_1},\boldsymbol{\chi}'_{j_2})$, for all $1\leq j_1,j_2\leq c$. ∎

According to Theorem 1, any decomposition factor $\boldsymbol{\mathcal{X}}$ of the Hessian matrix $\boldsymbol{X}^\top\boldsymbol{X}$ can be used instead of $\boldsymbol{X}$ without changing the geometric properties of the CVP and its associated quantization problem. This is useful for reducing the computational cost, e.g., we may use a square matrix $\boldsymbol{\mathcal{X}}\in\mathbb{R}^{c\times c}$ instead of the rectangular matrix $\boldsymbol{X}\in\mathbb{R}^{n\times c}$.

## 4.2 OBQ's GEOMETRIC INTERPRETATION

We first demonstrate the geometric interpretation of OBQ (GPTQ's slower predecessor) to facilitate our equivalence proof of GPTQ and Babai's algorithm in Section 4.3.

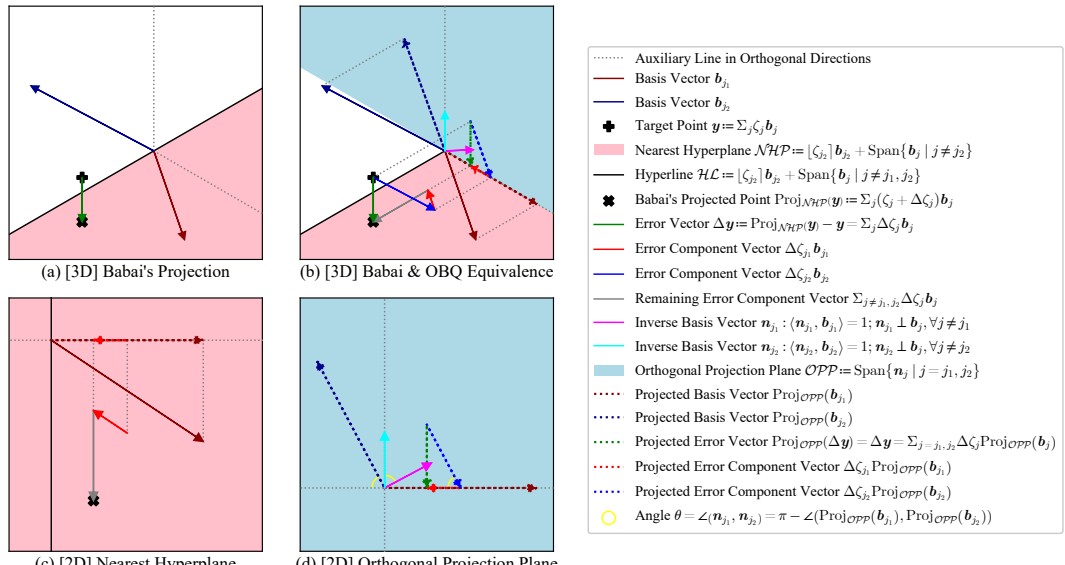

(a) [3D] Babai's Projection
(b) [3D] Babai & OBQ Equivalence
(c) [2D] Nearest Hyperplane
(d) [2D] Orthogonal Projection Plane

Figure 2: Equivalence of OBQ's error propagation and Babai's projection. **(a)** 3D plot showing the target being projected onto the nearest plane. **(b)** 3D plot showing how the projection error is propagated. **(c)** 2D plot showing the vectors on the nearest hyperplane in (a-b). **(d)** 2D plot showing the vectors on the orthogonal projection plane in (b).

**Theorem 2 (Error Propagation and Babai's Projection)** *Babai's nearest plane algorithm iteratively projects the target vector onto the nearest hyperplane and rounds the coefficient. The OBQ error propagation step (Eq. 2) is exactly this projection on the original basis $\boldsymbol{B} = \boldsymbol{X} \operatorname{diag}(\boldsymbol{s}_i)$ without basis reduction.*

**Proof** Let $\boldsymbol{B} = [\boldsymbol{b}_1, \ldots, \boldsymbol{b}_c]$ be the basis with $\boldsymbol{b}_j$ being a basis vector. Let $J$ be the set of unprojected indices with $j_1, j_2 \in J$ and $j_1 \neq j_2$. Let $\boldsymbol{y} = \sum_{j \in J} \zeta_j \boldsymbol{b}_j$ be the current residual target where $\zeta_j \in \mathbb{R}$ is a real number to be rounded to integers. Let $\mathcal{NHP} := \lfloor \zeta_{j_2} \rceil \boldsymbol{b}_{j_2} + \operatorname{Span}\{\boldsymbol{b}_j \mid j \neq j_2\}$ be the nearest hyperplane that is orthogonal to the Gram-Schmidt vector $\boldsymbol{b}_{j_2} - \sum_{j \neq j_2} \operatorname{Proj}_{\boldsymbol{b}_j}(\boldsymbol{b}_{j_2})$. Figure 2 (a) is a 3D plot showing the projection error vector $\Delta \boldsymbol{y} = \operatorname{Proj}_{\mathcal{NHP}}(\boldsymbol{y}) - \boldsymbol{y}$. We focus on analyzing the error propagation in the direction of basis $\boldsymbol{b}_{j_1}$ induced by the projection of basis $\boldsymbol{b}_{j_2}$ and collapse the span of other basis vectors to a single dimension as illustrated by the hyperline $\mathcal{HL} := \lfloor \zeta_{j_2} \rceil \boldsymbol{b}_{j_2} + \operatorname{Span}\{\boldsymbol{b}_j | j \neq j_1, j_2\}$. Figure 2 (b) is a 3D plot showing the decomposition of the error $\Delta \boldsymbol{y} = \sum_{j \in J} \Delta \zeta_j \boldsymbol{b}_j$ as the error component vectors in the basis directions. Figure 2 (c) is a 2D plot showing the vectors on plane $\mathcal{NHP}$. The number $\zeta_j$ will be updated to $\zeta_j + \Delta \zeta_j$ such that $\operatorname{Proj}_{\mathcal{NHP}}(\boldsymbol{y}) = \sum_{j \in J} (\zeta_j + \Delta \zeta_j) \boldsymbol{b}_j$. Next, let $\boldsymbol{N} = \boldsymbol{B}^{-\top} = [\boldsymbol{n}_1, \ldots, \boldsymbol{n}_c]$ be the inverse basis. Then, we have $\langle \boldsymbol{n}_j, \boldsymbol{b}_j \rangle = 1$ and $\boldsymbol{n}_j \perp \boldsymbol{b}_{j'}, \forall j \neq j'$. We project all the vectors in Figure 2 (b) onto the orthogonal projection plane $\mathcal{OPP} := \operatorname{Span}\{\boldsymbol{n}_j | j = j_1, j_2\}$ that is orthogonal to the hyperline $\mathcal{HL}$, and continue the proof in the 2D geometry in Figure 2 (d). Denote the angle $\theta = \angle(\boldsymbol{n}_{j_1}, \boldsymbol{n}_{j_2}) = \pi - \angle(\operatorname{Proj}_{\mathcal{OPP}}(\boldsymbol{b}_{j_1}), \operatorname{Proj}_{\mathcal{OPP}}(\boldsymbol{b}_{j_2}))$. Then, $\frac{\Delta \zeta_{j_1} \|\operatorname{Proj}_{\mathcal{OPP}}(\boldsymbol{b}_{j_1})\|}{\Delta \zeta_{j_2} \|\operatorname{Proj}_{\mathcal{OPP}}(\boldsymbol{b}_{j_2})\|} = \cos \theta = \frac{\langle \boldsymbol{n}_{j_1}, \boldsymbol{n}_{j_2} \rangle}{\|\boldsymbol{n}_{j_1}\| \|\boldsymbol{n}_{j_2}\|} = \frac{\|\boldsymbol{n}_{j_2}\|}{\|\boldsymbol{n}_{j_1}\|} \frac{\langle \boldsymbol{n}_{j_1}, \boldsymbol{n}_{j_2} \rangle}{\langle \boldsymbol{n}_{j_2}, \boldsymbol{n}_{j_2} \rangle}$. For $j = j_1, j_2$, $\|\operatorname{Proj}_{\mathcal{OPP}}(\boldsymbol{b}_j)\| \|\boldsymbol{n}_j\| = \frac{\langle \operatorname{Proj}_{\mathcal{OPP}}(\boldsymbol{b}_j), \boldsymbol{n}_j \rangle}{\cos(\frac{\pi}{2} - \theta)} = \frac{\langle \boldsymbol{b}_j, \boldsymbol{n}_j \rangle}{\cos(\frac{\pi}{2} - \theta)} = \frac{1}{\cos(\frac{\pi}{2} - \theta)}$. For $j, j' \in \{j_1, j_2\}$, $\langle \boldsymbol{n}_j, \boldsymbol{n}_{j'} \rangle = (\boldsymbol{N}^\top \boldsymbol{N})[j, j'] = (\boldsymbol{B}^\top \boldsymbol{B})^{-1}[j, j']$. Combining the above equations, $\Delta \zeta_{j_1} = \frac{\|\operatorname{Proj}_{\mathcal{OPP}}(\boldsymbol{b}_{j_2})\| \|\boldsymbol{n}_{j_2}\|}{\|\operatorname{Proj}_{\mathcal{OPP}}(\boldsymbol{b}_{j_1})\| \|\boldsymbol{n}_{j_1}\|} \frac{\langle \boldsymbol{n}_{j_1}, \boldsymbol{n}_{j_2} \rangle}{\langle \boldsymbol{n}_{j_2}, \boldsymbol{n}_{j_2} \rangle} \Delta \zeta_{j_2} = \frac{\langle \boldsymbol{n}_{j_1}, \boldsymbol{n}_{j_2} \rangle}{\langle \boldsymbol{n}_{j_2}, \boldsymbol{n}_{j_2} \rangle} \Delta \zeta_{j_2} =$

$\frac{\left(\boldsymbol{B}^{\top}\boldsymbol{B}\right)^{-1}[j_1,j_2]}{\left(\boldsymbol{B}^{\top}\boldsymbol{B}\right)^{-1}[j_2,j_2]}\Delta\zeta_{j_2}$. Finally, substituting $\boldsymbol{B} = \left(\boldsymbol{X}\operatorname{diag}\left(\boldsymbol{s}_i\right)\right)[:,J]$ and $\zeta_j = \frac{\boldsymbol{w}_i[j]}{\boldsymbol{s}_i[j]}$ completes the proof. ∎

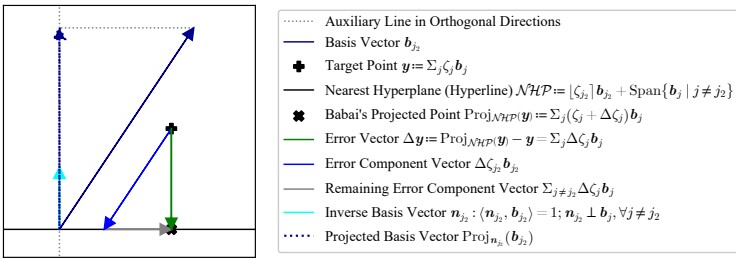

Figure 3: Geometric interpretation of OBQ's quantization order. This 2D plot shows the target being projected onto the nearest plane.

**Corollary 3 (OBQ Dimension Selection)** *At each dimension selection step (Eq. 1), OBQ selects the not-yet-quantized dimension $j$ such that the nearest hyperplane of dimension $j$ is closest to the target residual vector.*

**Proof** We use the same notations defined in Theorem 2. Figure 3 is a 2D plot showing the distance (projection error or quantization error) between the target residual vector $\boldsymbol{y}$ and the nearest hyperplane $\mathcal{NHP}$ of the basis $\boldsymbol{b}_{j_2}$. For better illustration, we collapse $\mathcal{NHP}$ into a single dimension. The distance $\|\Delta\boldsymbol{y}\|$ can be expressed as $\|\Delta\boldsymbol{y}\| = \left\|\operatorname{Proj}_{\boldsymbol{n}_{j_2}}\left(\Delta\boldsymbol{y}\right)\right\| = |\Delta\zeta_{j_2}|\left\|\operatorname{Proj}_{\boldsymbol{n}_{j_2}}\left(\boldsymbol{b}_{j_2}\right)\right\| = \frac{|\Delta\zeta_{j_2}||\langle\boldsymbol{b}_{j_2},\boldsymbol{n}_{j_2}\rangle|}{\|\boldsymbol{n}_{j_2}\|} = \frac{|\Delta\zeta_{j_2}|}{\|\boldsymbol{n}_{j_2}\|}$. For each $\boldsymbol{w}_i$, OBQ independently selects $j = \operatorname{argmin}_{j\in J}\frac{(\boldsymbol{q}_i[j]-\boldsymbol{w}_i[j])^2}{(\boldsymbol{X}[:,J]^{\top}\boldsymbol{X}[:,J])^{-1}[j,j]} = \operatorname{argmin}_{j\in J}\frac{(\Delta\zeta_j)^2}{\langle\boldsymbol{n}_j,\boldsymbol{n}_j\rangle} = \operatorname{argmin}_{j\in J}\frac{|\Delta\zeta_j|}{\|\boldsymbol{n}_j\|}$ as the next dimension to quantize, which is exactly minimizing this distance. ∎

### 4.3 GPTQ AND BABAI'S ALGORITHM

Originally, GPTQ (Algorithm 1) runs from the first to the last dimension ($j \leftarrow 1$ to $c$) while Babai's algorithm (Algorithm 2) runs from the last to the first dimension ($j \leftarrow c$ to 1). This is the only (superficial) difference between the two algorithms, as formalized below.

**Theorem 4 (GPTQ and Babai)** *GPTQ and Babai's algorithm without basis reduction will have the same results if we align the dimensional order of these two algorithms, e.g., running GPTQ from the last to the first dimension.*

**Proof** We prove this theorem both geometrically and algebraically. We first present the geometric proof. Theorem 2 shows that each intermediate weight vector produced by OBQ, equivalently GPTQ, can be viewed as Babai's residual vector in the activation space. At step $j$ (running from the last to the first dimension, $j \leftarrow c$ to 1), GPTQ's error propagation update is exactly Babai's projection at step $j$, which projects the current residual of the target vector onto the hyperplane orthogonal to the $j$-th Gram-Schmidt vector.

Alternatively, we present a more rigorous algebraic proof. Section B describes the exact quantization procedures using Babai's algorithm in more detail, with the pseudocode in Algorithm 4. Section C contains the equivalence proof, in which we proceed in three steps. First, we rewrite GPTQ to track the cumulative quantization error and show that this form is algebraically equivalent to the standard implementation. Second, we run GPTQ in the back-to-front order and replace the lower triangular factor with an upper triangular one so that each update affects only the not-yet-quantized coordinates. Third, we prove that the step-wise rounding decisions of the back-to-front GPTQ coincide with those of Babai's algorithm. ∎

**Geometric interpretation of GPTQ.** Theorem 4 shows that, if we regard the activations as the lattice basis and transform the floating-point weight vector to a target vector in the activation space,

GPTQ performs an *orthogonal walk* through a nested sequence of affine subspaces in a pre-computed dimensional order.

**Ineffectiveness of composing algorithms.** A seemingly appealing idea is to take the solution returned by any Babai iteration and then perform one further GPTQ-style error propagation step on the weights in the activation space, hoping to push the approximation even closer to the optimum. However, as proven in Section C.4, such an extra update vanishes: the final results of $\boldsymbol{Z}$ and $\boldsymbol{Q}$ remain unchanged. In other words, once Babai's projection has been executed, any subsequent GPTQ-style correction is algebraically redundant. This confirms that the equivalence in Theorem 4 is already tight; neither algorithm can be strengthened by composition.

### 4.4 GPTQ'S ERROR BOUND

Having established the correspondence between GPTQ and Babai's nearest plane algorithm, we can now import Babai's approximation guarantee to obtain an upper bound on the layer-wise quantization error in the no-clipping setting.

**Theorem 5 (GPTQ Error Bound)** *Assume no clipping ($\mathbb{Z}_\dagger = \mathbb{Z}$) and let $\boldsymbol{T}$ be the permutation matrix of the reversed GPTQ quantization order (equivalently $\boldsymbol{P}$ with the reversed column order). Let $\boldsymbol{D}$ be the diagonal matrix of the LDL decomposition of the permuted Hessian matrix $\boldsymbol{T}^\top \boldsymbol{X}^\top \boldsymbol{X} \boldsymbol{T}$. For every output channel $i$ ($1 \le i \le r$) produced by Babai's algorithm, or equivalently the GPTQ algorithm executed back-to-front, the (absolute) quantization error has a tight upper bound: $\|\boldsymbol{X} \operatorname{diag}(\boldsymbol{s}_i) \boldsymbol{z}_i - \boldsymbol{X} \boldsymbol{w}_i\|^2 \le \frac{1}{4} (\boldsymbol{T}^{-1} \boldsymbol{s}_i)^\top \boldsymbol{D} (\boldsymbol{T}^{-1} \boldsymbol{s}_i)$. For the relative bound for $\gamma$ with $\|\boldsymbol{X} \operatorname{diag}(\boldsymbol{s}_i) \boldsymbol{z}_i - \boldsymbol{X} \boldsymbol{w}_i\| \le \gamma \cdot \min_{\boldsymbol{z}_i' \in \mathbb{Z}^c} \|\boldsymbol{X} \operatorname{diag}(\boldsymbol{s}_i) \boldsymbol{z}_i' - \boldsymbol{X} \boldsymbol{w}_i\|$, we have $1 \le \gamma \le \sqrt{1 + \max_{1 \le j \le c} \frac{\sum_{j'=1}^{j} d_{j'}^2}{d_j^2}} \le \sqrt{c+1} \cdot \max_{1 \le j' \le j \le c} \frac{d_{j'}}{d_j}$ where $d_j = \sqrt{\boldsymbol{D}[j,j]} \left| (\boldsymbol{T}^{-1} \boldsymbol{s}_i)[j] \right|$.*

**Proof** The full proof of Theorem 5 is presented in Section D.1. ∎

If the scales $\boldsymbol{s}_i$ are small enough, we may assume the weights $\boldsymbol{w}_i$ are nearly uniformly distributed within the hyper-cuboid constructed by Babai's orthogonalized basis vectors, the expected absolute error will be $\frac{1}{3}$ of the worst-case bound. See Section D.2 for a proof.

### 4.5 THE ROLE OF QUANTIZATION ORDER IN GPTQ

The quadratic form on the right-hand side of the absolute error bound in Theorem 5 is sensitive to the pivot order of the LDL decomposition of the Hessian matrix; this is the quantization order. Reordering the dimensions changes the entries of the diagonal matrix $\boldsymbol{D}$ before the scale $\boldsymbol{s}_i$ is "weighted" by them. A poor order may place large $\boldsymbol{D}$ entries against large $\boldsymbol{s}_i$ entries and hence inflate the bound. For a batched quantization algorithm like GPTQ, the order should be independent of the output channel $i$. To develop a good heuristic order, a reasonable approximation to make, especially for large quantization group sizes, is that the elements of $\boldsymbol{s}_i[j]$ are equal for all $1 \le j \le c$. Then we can focus on finding the optimal pivot order for the LDL decomposition of the Hessian matrix $\boldsymbol{X}^\top \boldsymbol{X}$ to minimize $\operatorname{tr}(\boldsymbol{D})$.

Finding the optimal order is NP-hard (Rose et al., 1976). However, heuristics often effectively reduce the trace term in practice. Even with clipping, heuristics can reduce the error. GPTQ introduces the act-order, the descending order of the Hessian diagonal, i.e., the ascending order of the Hessian diagonal when applied to Babai's algorithm.

To improve upon act-order, we propose the min-pivot order, which is essentially taking the minimum diagonal entry at each LDL (or Cholesky) decomposition step. This order can be calculated by Algorithm 3, which has cubic time complexity and does not increase the overall time complexity of quantization. This order also has a geometric interpretation, as the order of the Gram-Schmidt orthogonalization process of the basis: always taking the shortest residual vector as the next one to orthogonalize, agreeing with Babai's relative error bound. Across our preliminary runs (Section D.3), min-pivot *consistently* reduces $\operatorname{tr}(\boldsymbol{D})$ relative to act-order, but the downstream accuracy gains are modest. We nevertheless report min-pivot as a principled choice, and view act-order as a cheap approximation that only considers the Hessian diagonal, which already captures most of the benefit when the Hessian matrix is well-conditioned.

---

**Algorithm 3:** Min-Pivot

---

**Input:** Hessian $\boldsymbol{H} \in \mathbb{R}^{c \times c}$

**Output:** order encoded as a permutation matrix $\boldsymbol{T} \in \{0, 1\}^{c \times c}$

1   $J \leftarrow \{1, \ldots, c\}$ // initialize the not-yet-pivoted indices
2   $\boldsymbol{T} \leftarrow \boldsymbol{0}$ // initialize the output permutation matrix
3   **for** $j \leftarrow 1$ to $c$ **do**
4      $j' \leftarrow \arg\min_{j' \in J} \boldsymbol{H}[j', j']$ // choose next index with the smallest current diagonal
5      $\boldsymbol{H} \leftarrow \boldsymbol{H} - \boldsymbol{H}[:, j']\boldsymbol{H}[j', :]/\boldsymbol{H}[j', j']$ // updates remaining entries with rank-1 Schur complement
6      $\boldsymbol{T}[j', j] \leftarrow 1$ // record the index
7      $J \leftarrow J \setminus \{j'\}$ // mark pivot as used
8   **end**

---

## 5   Applications

The original GPTQ algorithm clips the overflowed integers at the rounding step, introducing large errors that violate the error bound in Theorem 5. In this section, we explore error-guaranteed variants of GPTQ that work in the no-clipping regime.

We notice that enforcing no-clipping by simply increasing scales is counterproductive: larger scales enlarge the bound, and the resulting errors can exceed those of a clipped scheme such as MSE. Hence, any practical no-clipping design must account for the weight distributions that are known to have heavy outliers (Li et al., 2025). We would still like to apply small scales, but use small bitwidths for the bulk of inliers while handling the overflowed outliers with more storage budget without clipping them. We therefore propose two overflow-tolerant schemes.

**Scale-adjusted SpQR (SSQR).** SpQR (Dettmers et al., 2024) keeps a small set of outliers in full precision, but it still leaves clipping in place: weights are grouped, the outliers and a shared scale are chosen per group before the GPTQ updates, and there is no guarantee the updated inlier weights stay within the representable range. We design SSQR with a scale-adjustment mechanism to fix this issue. For simplicity, we discard SpQR's second-level quantization for the scales. For a weight vector $\boldsymbol{w}_i \in \mathbb{R}^c$, we represent the quantized weight $\boldsymbol{q}_i \in \mathbb{R}^c$ as $\operatorname{diag}(\boldsymbol{s}_i)\boldsymbol{z}_i + \boldsymbol{\xi}_i$ where $\boldsymbol{z} \in \mathbb{Z}_{\dagger}^c$ is the low-bitwidth integer weight vector, $\boldsymbol{s}_i \in \mathbb{R}_{\neq 0}^c$ is the floating-point scale vector with each scale shared per group (only one number per group is actually stored), and $\boldsymbol{\xi}_i \in \mathbb{R}^c$ is the sparse floating-point outlier vector (stored in the compressed sparse row format, CSR) that captures all the overflowed weights after GPTQ's error propagation. The scale-adjustment mechanism tunes the scale $\boldsymbol{s}_i$ until the density of $\boldsymbol{\xi}_i$ satisfies the specified rate. Because exhaustive trial-and-error over per-group scales is infeasible in large layers, the mechanism only proportionally changes $\boldsymbol{s}_i$ so that the search space reduces to one dimension. With the observation that the outlier rate is negatively related to the scales in general, this can be done via binary search: initialize $\boldsymbol{s}_i$ using MSE, quantize $\boldsymbol{w}_i$ with the specified format using GPTQ without clipping, calculate the density of $\boldsymbol{\xi}_i$, and adjust $\boldsymbol{s}_i$ and iterate. Section E.1 Algorithm 9 is the pseudocode.

**Huffman-encoded post-training quantization (HPTQ).** To better align with the infinite, unconstrained lattice in CVP, we design HPTQ, which represents both inliers and outliers in a unified, equal-spaced integer grid. The idea is to use Huffman encoding, which was also explored for network compression by Choi et al. (2017). We quantize the weight matrix $\boldsymbol{W} \in \mathbb{R}^{c \times r}$ as $\boldsymbol{Q} = s\boldsymbol{Z}$ with a single scalar $s \in \mathbb{R}_{\neq 0}$ and integers $\boldsymbol{Z} \in \mathbb{Z}^{c \times r}$. We select $s$ via an entropy-guided binary search: initialize a range proportional to the maximum weight, quantize to unclipped integers with GPTQ, measure the Huffman coding cost of $\boldsymbol{Z}$, and adjust $s$ until the encoded bits meet a target average bitwidth. This yields uneven-bitwidth representations that preserve accuracy while meeting a compression budget. Section E.1 Algorithm 11 is the pseudocode.

Experiments compare round-to-nearest (RTN), original GPTQ, HPTQ, and SSQR with 1~5% outliers. We also include Huffman-encoded RTN (HRTN) as a baseline to HPTQ, which mirrors HPTQ but replaces GPTQ with RTN (pseudocode: Section E.1 Algorithm 12). The quantization order is act-order for all methods. RTN, GPTQ, and SSQR use group size 128. RTN and GPTQ calculate the scales with the MSE method. Figure 4 (a-b) shows that HPTQ sustains low perplexity on Qwen3-8B

at reduced bitwidths and scales favorably across model sizes, with 3.125-bit emerging as Pareto optimal in terms of perplexity vs compression. Further information can be found in Section E, including the experimental setup (Section E.2), additional metrics such as benchmark results (Qwen3 models: Section E.3; Llama models: Section E.4), and comparison with other methods (Section E.5).

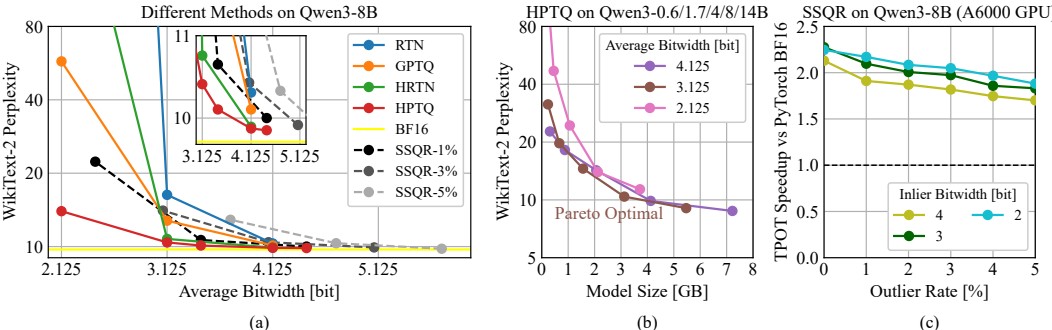

Figure 4: **(a)** Comparison of quantization methods (RTN, GPTQ, HRTN, HPTQ, and SSQR with 1~5% outliers) on Qwen3-8B evaluated on WikiText-2. Perplexity is plotted against the average effective bitwidth per weight, with the BF16 baseline shown as a horizontal line. HPTQ has the best (lowest) perplexity. See Section E.3 for zero-shot evaluation results. **(b)** Scaling behavior of HPTQ across multiple model sizes (0.6B, 1.7B, 4B, 8B, 14B) and bitwidths (4.125, 3.125, 2.125). The x-axis denotes the effective model size after quantization, and the y-axis shows perplexity on WikiText-2. Each curve corresponds to a fixed bitwidth, while points along a curve represent different model scales. Using our HPTQ method, 3.125-bit stands out as the Pareto optimal bitwidth (optimal perplexity vs compression trade-offs). **(c)** End-to-end inference speedups of our SSQR kernel vs the PyTorch BF16 matrix multiplication kernel on NVIDIA RTX A6000 GPU. We run the Qwen3-8B model across multiple outlier rates (0%~5%) and inlier bitwidths (4, 3, 2) and measure the TPOT (time per output token) metric. Our kernel achieves about $2\times$ speedup end-to-end.

**CUDA inference kernel.** We implement an inference kernel for SSQR in CUDA/C++, optimized for low-batch latency, handling both dense inliers and sparse outliers while targeting the Ampere platform. The kernel supports group-quantized inlier weights in the 2-4-bit range with scales in 16 bits and support for unstructured sparsity, used to avoid weight clipping. Figure 4 (c) visualizes the end-to-end speedup in the LLM decoding phase vs the PyTorch BF16 kernel. Our kernel achieves about $2\times$ speedup across different bitwidth and outlier rate settings when generating 128 new tokens at a batch size of 1. Technical details and layer-wise speedups are described in Section E.6.

## 6 CLOSING REMARKS

**Summary.** We have shown that GPTQ, when executed back-to-front, is mathematically identical to Babai's nearest plane algorithm applied to the lattice defined by a layer's Hessian without basis reduction. Based on this theory, we propose error-guaranteed practical methods and provide optimized CUDA kernels that deliver low-latency inferences. More broadly, the lattice perspective opens a two-way channel: decades of the closest vector problem (CVP) heuristics can refine practical quantizers, while the behavior of massive neural networks may, in turn, inspire new questions for lattice theory.

**Future work.** Looking ahead, extending the analysis to clipped grids and exploring (scale-aware) basis reductions are the immediate next steps. However, we emphasize that the state-of-the-art 4-bit floating-point formats (e.g., MXFP4 and NVFP4) are essentially no-clipping (Egiazarian et al., 2025; Chen et al., 2026): since they use very small quantization groups (32 and 16, respectively), the near-optimal choice of scale is AbsMax per-group, which leads to no weight being clipped. As such, a no-clipping analysis of these formats would directly apply to actual practice. We will also extend the lattice view beyond weight-only linear layers to activation and KV-cache quantization.

ETHICS STATEMENT

Throughout this work, we have strictly adhered to the ICLR Code of Ethics. All datasets utilized in our experiments are publicly available and widely recognized within the scientific community. We ensure that these datasets do not contain any personally identifiable information or sensitive content. Our work does not involve human subjects, animals, or any form of personal data collection. We have thoroughly considered potential dual-use concerns and do not foresee any harmful applications of our methods. There are no conflicts of interest to declare, and no external sponsorship influenced the outcomes of this research. All experiments were conducted with integrity and transparency.

REPRODUCIBILITY STATEMENT

We are committed to ensuring that our work is transparent and reproducible. To facilitate this, clear explanations of any assumptions and a complete proof of the claims have been included in the main text and appendix. We also share the source code as part of the supplementary materials. The code is documented and includes instructions for setting up the environment, running the simulations, and reproducing the results presented in our paper. By making our resources openly available and providing detailed explanations, we aim to enable the research community to validate and build upon our findings.

ACKNOWLEDGMENTS

This research was supported by the Scientific Service Units (SSU) of ISTA through resources provided by Scientific Computing (SciComp). The ISTA team was supported by generous grants from Google and NVIDIA. The authors thank Vage Egiazarian for the discussions on this work.

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

# A    APPENDIX TABLE OF CONTENTS

# B  Babai's Algorithm for Batched Quantization

Given the equivalence we have shown in Section 4.1, the quantization problem can be converted to CVP, allowing us to apply Babai's nearest plane algorithm in the context of quantization. A naive way is to compute $\boldsymbol{B}_{(i)} = \boldsymbol{\mathcal{X}} \operatorname{diag}(\boldsymbol{s}_i)$ and $\boldsymbol{y}_{(i)} = \boldsymbol{\mathcal{X}} \boldsymbol{w}_i$ and run Babai's algorithm independently for all $1 \leq i \leq r$. However, this is computationally inefficient, as we will need to compute the expensive $(O(c^4))$ LLL basis reduction transformation $\boldsymbol{T}_{(i)}$ for the basis $\boldsymbol{B}_{(i)}$ and the expensive $(O(c^3))$ QR decomposition of $\boldsymbol{A}_{(i)} = \boldsymbol{B}_{(i)} \boldsymbol{T}_{(i)}$ for $r$ times. However, a few adjustments can be made to simplify the computation and enable batched processing.

**Disabling basis reduction.** The LLL basis reduction is unfortunately scale-sensitive, generating different transformations $\boldsymbol{T}_{(i)}$ for different scales $\boldsymbol{s}_i$ (unless all the $\boldsymbol{s}_i$ vectors are parallel), which prohibits the reuse of QR decomposition results. Furthermore, LLL basis reduction is incompatible with clipping, as the roundings are performed in another basis, and there is no easy way to do the clipping for the original basis.

**Changing quantization order.** Quantization order is a feature in GPTQ that controls the rounding and clipping order of the dimensions. This order influences the quantization error, as we discuss in Section 4.5. In the context of Babai's algorithm, this corresponds to the order of the basis in the Gram-Schmidt orthogonalization and the hyperplane projections, as shown in Figure 1 (g-h). To do so, we can replace the LLL basis reduction in Babai's algorithm with a permutation by setting the transformation matrix $\boldsymbol{T}$ to a permutation matrix that is independent of $i$.

**Theorem 6 (Babai's Quantization Order)** *If $\boldsymbol{T}$ is a permutation matrix that does not depend on $i$, the orthogonal matrix $\boldsymbol{\Phi}$ can be reused without recomputing the QR decomposition for each $i$.*

**Proof** The permutation matrix $\boldsymbol{T} \in \{0, 1\}^{c \times c}$ has exactly one non-zero element in each row and column. Scaling the rows of $\boldsymbol{T}$ can also be interpreted as scaling the columns of $\boldsymbol{T}$, therefore its multiplication with a diagonal matrix has property: $\operatorname{diag}(\boldsymbol{s}_i) \boldsymbol{T} = \boldsymbol{T} \operatorname{diag}(\boldsymbol{T}^{-1} \boldsymbol{s}_i)$. Let $\boldsymbol{A} = \boldsymbol{\mathcal{X}} \boldsymbol{T}$, $\boldsymbol{A}_{(i)} = \boldsymbol{\mathcal{X}} \operatorname{diag}(\boldsymbol{s}_i) \boldsymbol{T}$. Denote the QR decomposition of $\boldsymbol{A}$ as $\boldsymbol{A} = \boldsymbol{\Phi} \boldsymbol{R}$ with $\boldsymbol{\Phi}$ being an orthogonal matrix and $\boldsymbol{R}$ being an upper triangular matrix. Then, the QR decomposition of $\boldsymbol{A}_{(i)}$ becomes $\boldsymbol{A}_{(i)} = \boldsymbol{\mathcal{X}} \operatorname{diag}(\boldsymbol{s}_i) \boldsymbol{T} = \boldsymbol{\mathcal{X}} \boldsymbol{T} \operatorname{diag}(\boldsymbol{T}^{-1} \boldsymbol{s}_i) = \boldsymbol{A} \operatorname{diag}(\boldsymbol{T}^{-1} \boldsymbol{s}_i) = \boldsymbol{\Phi}(\boldsymbol{R} \operatorname{diag}(\boldsymbol{T}^{-1} \boldsymbol{s}_i))$. Therefore, the QR decompositions of $\boldsymbol{A}_{(i)}$ share the same orthogonal matrix $\boldsymbol{\Phi}$ for all $1 \leq i \leq r$. ∎

As shown in Theorem 6, changing quantization order does not require repeated computation of the QR decomposition. Note that, we also need to permute the scale $\boldsymbol{S}$ accordingly to $\boldsymbol{T}^{-1} \boldsymbol{S}$.

**Selecting basis.** Putting things together, we are interested in $\boldsymbol{A} = \boldsymbol{\mathcal{X}} \boldsymbol{T}$ and its QR decomposition $\boldsymbol{\Phi}$. Theorem 1 allows us to choose any Hessian factor $\boldsymbol{\mathcal{X}}$ while keeping the result intact. Without loss of generality, we can choose a $\boldsymbol{\mathcal{X}}$ such that $\boldsymbol{A}$ is an upper triangular matrix and the QR decomposition becomes trivial: $\boldsymbol{\Phi} = \mathbf{I}$, which simplifies the computation. The upper triangular matrix $\boldsymbol{A}$ can be directly computed from the Cholesky decomposition of the permuted Hessian matrix $\boldsymbol{A}^\top \boldsymbol{A} = \boldsymbol{T}^\top \boldsymbol{X}^\top \boldsymbol{X} \boldsymbol{T}$.

Applying all the above considerations, we construct Algorithm 4 for batched quantization using Babai's algorithm.

---

**Algorithm 4:** Babai's Quantize

  **Input:** $\boldsymbol{W}, \boldsymbol{S}, \boldsymbol{X}, \boldsymbol{T}, \lambda, \mathbb{Z}_\dagger$
  **Output:** $\boldsymbol{Z}, \boldsymbol{Q}$

1 $\boldsymbol{H} \leftarrow \boldsymbol{T}^\top (\boldsymbol{X}^\top \boldsymbol{X} + \lambda \mathbf{I}) \boldsymbol{T}$
2 $\boldsymbol{A} \leftarrow \text{Cholesky}(\boldsymbol{H})^\top$
3 $\boldsymbol{W}, \boldsymbol{S} \leftarrow \boldsymbol{T}^{-1} \boldsymbol{W}, \boldsymbol{T}^{-1} \boldsymbol{S}$
4 $\boldsymbol{Y}, \boldsymbol{Q}, \boldsymbol{Z} \leftarrow \boldsymbol{A} \boldsymbol{W}, \boldsymbol{W}, \boldsymbol{0}$
5 **for** $j \leftarrow c$ to $1$ **do**
6   $\quad \omega \leftarrow \boldsymbol{Y}[j, :] / \boldsymbol{A}[j, j]$
7   $\quad \zeta \leftarrow \omega / \boldsymbol{S}[j, :]$
8   $\quad \boldsymbol{Z}[j, :] \leftarrow \text{Round}(\zeta, \mathbb{Z}_\dagger)$
9   $\quad \boldsymbol{Q}[j, :] \leftarrow \boldsymbol{Z}[j, :] * \boldsymbol{S}[j, :]$
10  $\quad \boldsymbol{Y} \leftarrow \boldsymbol{Y} - \boldsymbol{A}[:, j] \boldsymbol{Q}[j, :]$
11 **end**
12 $\boldsymbol{Z}, \boldsymbol{Q} \leftarrow \boldsymbol{T} \boldsymbol{Z}, \boldsymbol{T} \boldsymbol{Q}$

---

## C  ALGEBRAIC EQUIVALENCE PROOF OF GPTQ AND BABAI'S ALGORITHM

In this section, we prove Theorem 4 that GPTQ (Algorithm 1) and Babai's algorithm (Algorithm 4) are equivalent if the dimensional orders are opposite.

Because a permutation matrix acts only as re-ordering coordinates, we may apply the permutation once at the beginning (to $\boldsymbol{W}$, $\boldsymbol{S}$, and $\boldsymbol{X}$) and once at the end (to $\boldsymbol{Z}$ and $\boldsymbol{Q}$) without affecting any intermediate arithmetic. Hence, all algebra performed inside the two algorithms can be analyzed on the permuted basis where the permutation matrix is the identity. On that basis, the sole distinction between GPTQ and Babai's algorithm lies in the direction of the iterations. Proving that GPTQ running back-to-front ($j \leftarrow c$ to 1) reproduces Babai's updates in Babai's default iteration direction would complete the equivalence proof.

We follow a three-step proof scheme.

- **Step 1.** Proving that the original GPTQ algorithm (Algorithm 5) that uses relative quantization error row vector $\boldsymbol{\varepsilon} \in \mathbb{R}^{1 \times r}$ is equivalent to a new algorithm (Algorithm 6) using the absolute quantization error matrix $\boldsymbol{\Delta} \in \mathbb{R}^{c \times r}$.

- **Step 2.** Reversing the iteration in Algorithm 6 and writing the reversed-iteration algorithm as Algorithm 7.

- **Step 3.** Proving that the reversed-iteration algorithm Algorithm 7 is equivalent to Babai's algorithm Algorithm 8.

Algorithms 5 to 8 are intentionally written in the linear algebra form. $\mathbf{e}_j \in \mathbb{R}^c$ is the standard basis vector whose elements are 0 except the $j$-th element being 1, which is used as the row or column selector of a matrix. The superscripts in parentheses denote the versions of the variables during the iterations. $\boldsymbol{\omega}, \boldsymbol{\zeta} \in \mathbb{R}^{1 \times r}$ are intermediate row vectors. Additionally, $\boldsymbol{L}$ is the LDL decomposition of the Hessian inverse $\boldsymbol{H}^{-1} = \boldsymbol{L} \boldsymbol{D}_{\mathrm{L}}^{\frac{1}{2}} \boldsymbol{D}_{\mathrm{L}}^{\frac{1}{2}} \boldsymbol{L}^{\top}$ where $\boldsymbol{L}$ is a lower triangular matrix with all diagonal elements being 1, and $\boldsymbol{D}_{\mathrm{L}}^{\frac{1}{2}}$ is a non-negative diagonal matrix. Similarly, $\boldsymbol{U}$ is the "UDU" decomposition of the Hessian inverse $\boldsymbol{H}^{-1} = \boldsymbol{U} \boldsymbol{D}_{\mathrm{U}}^{\frac{1}{2}} \boldsymbol{D}_{\mathrm{U}}^{\frac{1}{2}} \boldsymbol{U}^{\top}$ where $\boldsymbol{U}$ is an upper triangular matrix with all diagonal elements being 1, and $\boldsymbol{D}_{\mathrm{U}}^{\frac{1}{2}}$ is a non-negative diagonal matrix.

Note: the symbols are overloaded in Algorithms 5 to 8, and the variables using the same symbols may carry different values, even if the inputs to the algorithms are the same.

### C.1  STEP 1

To distinguish the variables using the same symbol in Algorithms 5 and 6, we use symbols without ˆ to denote the symbols in Algorithm 5, and use the symbols with ˆ for Algorithm 6.

**Claim**

$$\boldsymbol{\omega}_j = \hat{\boldsymbol{\omega}}_j, \quad 1 \leq j \leq c, \tag{3}$$

and consequently,

$$\boldsymbol{Z}^{(j)} = \hat{\boldsymbol{Z}}^{(j)}, \quad 0 \leq j \leq c, \tag{4}$$

and

$$\boldsymbol{Q}^{(j)} = \hat{\boldsymbol{Q}}^{(j)}, \quad 0 \leq j \leq c. \tag{5}$$

**Proof Eq. 3 by Induction**

The following equalities are held by the design of Algorithms 5 and 6:

$$\boldsymbol{Q}^{(0)} = \hat{\boldsymbol{Q}}^{(0)} = \boldsymbol{W}^{(0)} = \hat{\boldsymbol{W}}^{(0)}. \tag{6}$$

$$\boldsymbol{\omega}^{(j)} = \mathbf{e}_j^{\top} \boldsymbol{W}^{(j-1)}, \quad 1 \leq j \leq c. \tag{7}$$

$$\hat{\boldsymbol{\omega}}^{(j)} = \mathbf{e}_j^{\top} \hat{\boldsymbol{W}}^{(j-1)}, \quad 1 \leq j \leq c. \tag{8}$$

$$\boldsymbol{Q}^{(j)} = \boldsymbol{Q}^{(j-1)} + \mathbf{e}_j \left( \mathbf{e}_j^{\top} \boldsymbol{Z}^{(j)} \operatorname{diag}\left(\boldsymbol{S}^{\top} \mathbf{e}_j\right) - \mathbf{e}_j^{\top} \boldsymbol{Q}^{(j-1)} \right), \quad 1 \leq j \leq c. \tag{9}$$

---

**Algorithm 5:** GPTQ Original (Front-to-Back)

---

**Input:** $W, S, X, \lambda, \mathbb{Z}_\dagger$
**Output:** $Z, Q$

**1** $H \leftarrow X^\top X + \lambda \mathbf{I}$
**2** $L \leftarrow \text{LDL}\left(H^{-1}\right)$
**3** $W^{(0)} \leftarrow W$
**4** $Q^{(0)}, Z^{(0)} \leftarrow W^{(0)}, \mathbf{0}$
**5** **for** $j \leftarrow 1$ to $c$ **do**
**6** $\quad \omega^{(j)} \leftarrow \mathbf{e}_j^\top W^{(j-1)}$
**7** $\quad \zeta^{(j)} \leftarrow \omega^{(j)} \operatorname{diag}\left(S^\top \mathbf{e}_j\right)^{-1}$
**8** $\quad Z^{(j)} \leftarrow Z^{(j-1)} + \mathbf{e}_j \left( \text{ROUND}\left(\zeta^{(j)}, \mathbb{Z}_\dagger\right) - \mathbf{e}_j^\top Z^{(j-1)}\right)$
**9** $\quad Q^{(j)} \leftarrow Q^{(j-1)} + \mathbf{e}_j \left(\mathbf{e}_j^\top Z^{(j)} \operatorname{diag}\left(S^\top \mathbf{e}_j\right) - \mathbf{e}_j^\top Q^{(j-1)}\right)$
**10** $\quad \varepsilon^{(j)} \leftarrow \mathbf{e}_j^\top Q^{(j)} - \omega^{(j)}$
**11** $\quad W^{(j)} \leftarrow W^{(j-1)} + L\mathbf{e}_j \varepsilon^{(j)}$
**12** **end**
**13** $Z, Q \leftarrow Z^{(c)}, Q^{(c)}$

---

**Algorithm 6:** GPTQ Type-2 (Front-to-Back)

---

**Input:** $W, S, X, \lambda, \mathbb{Z}_\dagger$
**Output:** $Z, Q$

**1** $H \leftarrow X^\top X + \lambda \mathbf{I}$
**2** $L \leftarrow \text{LDL}\left(H^{-1}\right)$
**3** $W^{(0)} \leftarrow W$
**4** $Q^{(0)}, Z^{(0)} \leftarrow W^{(0)}, \mathbf{0}$
**5** **for** $j \leftarrow 1$ to $c$ **do**
**6** $\quad \omega^{(j)} \leftarrow \mathbf{e}_j^\top W^{(j-1)}$
**7** $\quad \zeta^{(j)} \leftarrow \omega^{(j)} \operatorname{diag}\left(S^\top \mathbf{e}_j\right)^{-1}$
**8** $\quad Z^{(j)} \leftarrow Z^{(j-1)} + \mathbf{e}_j \left( \text{ROUND}\left(\zeta^{(j)}, \mathbb{Z}_\dagger\right) - \mathbf{e}_j^\top Z^{(j-1)}\right)$
**9** $\quad Q^{(j)} \leftarrow Q^{(j-1)} + \mathbf{e}_j \left(\mathbf{e}_j^\top Z^{(j)} \operatorname{diag}\left(S^\top \mathbf{e}_j\right) - \mathbf{e}_j^\top Q^{(j-1)}\right)$
**10** $\quad \Delta^{(j)} \leftarrow Q^{(j)} - W^{(0)}$ // new
**11** $\quad W^{(j)} \leftarrow W^{(0)} - L^{-1}\Delta^{(j)}$ // new
**12** **end**
**13** $Z, Q \leftarrow Z^{(c)}, Q^{(c)}$

---

---

**Algorithm 7:** GPTQ Type-2 (Back-to-Front)

---

**Input:** $\boldsymbol{W}, \boldsymbol{S}, \boldsymbol{X}, \lambda, \mathbb{Z}_\dagger$
**Output:** $\boldsymbol{Z}, \boldsymbol{Q}$

1   $\boldsymbol{H} \leftarrow \boldsymbol{X}^\top \boldsymbol{X} + \lambda \mathbf{I}$
2   $\boldsymbol{U} \leftarrow \text{UDU}\left(\boldsymbol{H}^{-1}\right)$ // new
3   $\boldsymbol{W}^{(c+1)} \leftarrow \boldsymbol{W}$
4   $\boldsymbol{Q}^{(c+1)}, \boldsymbol{Z}^{(c+1)} \leftarrow \boldsymbol{W}^{(c+1)}, \mathbf{0}$
5   **for** $j \leftarrow c$ to $1$ **do**
6      $\boldsymbol{\omega}^{(j)} \leftarrow \mathbf{e}_j^\top \boldsymbol{W}^{(j+1)}$
7      $\boldsymbol{\zeta}^{(j)} \leftarrow \boldsymbol{\omega}^{(j)} \operatorname{diag}\left(\boldsymbol{S}^\top \mathbf{e}_j\right)^{-1}$
8      $\boldsymbol{Z}^{(j)} \leftarrow \boldsymbol{Z}^{(j+1)} + \mathbf{e}_j \left(\text{ROUND}\left(\boldsymbol{\zeta}^{(j)}, \mathbb{Z}_\dagger\right) - \mathbf{e}_j^\top \boldsymbol{Z}^{(j+1)}\right)$
9      $\boldsymbol{Q}^{(j)} \leftarrow \boldsymbol{Q}^{(j+1)} + \mathbf{e}_j \left(\mathbf{e}_j^\top \boldsymbol{Z}^{(j)} \operatorname{diag}\left(\boldsymbol{S}^\top \mathbf{e}_j\right) - \mathbf{e}_j^\top \boldsymbol{Q}^{(j+1)}\right)$
10     $\boldsymbol{\Delta}^{(j)} \leftarrow \boldsymbol{Q}^{(j)} - \boldsymbol{W}^{(c+1)}$
11     $\boldsymbol{W}^{(j)} \leftarrow \boldsymbol{W}^{(c+1)} - \boldsymbol{U}^{-1} \boldsymbol{\Delta}^{(j)}$ // new
12   **end**
13   $\boldsymbol{Z}, \boldsymbol{Q} \leftarrow \boldsymbol{Z}^{(1)}, \boldsymbol{Q}^{(1)}$

---

**Algorithm 8:** Babai-Quantize (Default Order)

---

**Input:** $\boldsymbol{W}, \boldsymbol{S}, \boldsymbol{X}, \lambda, \mathbb{Z}_\dagger$
**Output:** $\boldsymbol{Z}, \boldsymbol{Q}$

1   $\boldsymbol{H} \leftarrow \boldsymbol{X}^\top \boldsymbol{X} + \lambda \mathbf{I}$
2   $\boldsymbol{A} \leftarrow \text{CHOLESKY}\left(\boldsymbol{H}\right)^\top$
3   $\boldsymbol{Y}^{(c+1)}, \boldsymbol{Q}^{(c+1)}, \boldsymbol{Z}^{(c+1)} \leftarrow \boldsymbol{A}\boldsymbol{W}, \boldsymbol{W}, \mathbf{0}$
4   **for** $j \leftarrow c$ to $1$ **do**
5      $\boldsymbol{\omega}^{(j)} \leftarrow \dfrac{\mathbf{e}_j^\top \boldsymbol{Y}^{(j+1)}}{\mathbf{e}_j^\top \boldsymbol{A} \mathbf{e}_j}$
6      $\boldsymbol{\zeta}^{(j)} \leftarrow \boldsymbol{\omega}^{(j)} \operatorname{diag}\left(\boldsymbol{S}^\top \mathbf{e}_j\right)^{-1}$
7      $\boldsymbol{Z}^{(j)} \leftarrow \boldsymbol{Z}^{(j+1)} + \mathbf{e}_j \left(\text{ROUND}\left(\boldsymbol{\zeta}^{(j)}, \mathbb{Z}_\dagger\right) - \mathbf{e}_j^\top \boldsymbol{Z}^{(j+1)}\right)$
8      $\boldsymbol{Q}^{(j)} \leftarrow \boldsymbol{Q}^{(j+1)} + \mathbf{e}_j \left(\mathbf{e}_j^\top \boldsymbol{Z}^{(j)} \operatorname{diag}\left(\boldsymbol{S}^\top \mathbf{e}_j\right) - \mathbf{e}_j^\top \boldsymbol{Q}^{(j+1)}\right)$
9      $\boldsymbol{Y}^{(j)} \leftarrow \boldsymbol{Y}^{(j+1)} - \boldsymbol{A} \mathbf{e}_j \mathbf{e}_j^\top \boldsymbol{Q}^{(j)}$
10   **end**
11   $\boldsymbol{Z}, \boldsymbol{Q} \leftarrow \boldsymbol{Z}^{(1)}, \boldsymbol{Q}^{(1)}$

---

$$\hat{\boldsymbol{Q}}^{(j)} = \hat{\boldsymbol{Q}}^{(j-1)} + \mathbf{e}_j \left( \mathbf{e}_j^\top \hat{\boldsymbol{Z}}^{(j)} \operatorname{diag} \left( \boldsymbol{S}^\top \mathbf{e}_j \right) - \mathbf{e}_j^\top \hat{\boldsymbol{Q}}^{(j-1)} \right), \quad 1 \le j \le c. \tag{10}$$

$$\boldsymbol{\varepsilon}^{(j)} = \mathbf{e}_j^\top \boldsymbol{Q}^{(j)} - \boldsymbol{\omega}^{(j)}, \quad 1 \le j \le c. \tag{11}$$

$$\boldsymbol{\Delta}^{(j)} = \hat{\boldsymbol{Q}}^{(j)} - \hat{\boldsymbol{W}}^{(0)}, \quad 1 \le j \le c. \tag{12}$$

$$\boldsymbol{W}^{(j)} = \boldsymbol{W}^{(j-1)} + \boldsymbol{L}\mathbf{e}_j\boldsymbol{\varepsilon}^{(j)}, \quad 1 \le j \le c. \tag{13}$$

$$\hat{\boldsymbol{W}}^{(j)} = \hat{\boldsymbol{W}}^{(0)} - \boldsymbol{L}^{-1}\boldsymbol{\Delta}^{(j)}, \quad 1 \le j \le c. \tag{14}$$

Extend the definition of $\boldsymbol{\Delta}^{(j)}$ (Eq. 12) for $j = 0$,

$$\boldsymbol{\Delta}^{(j)} = \hat{\boldsymbol{Q}}^{(j)} - \hat{\boldsymbol{W}}^{(0)}, \quad 0 \le j \le c. \tag{15}$$

Then we have $\boldsymbol{\Delta}^{(0)} = \hat{\boldsymbol{Q}}^{(0)} - \hat{\boldsymbol{W}}^{(0)} = \hat{\boldsymbol{W}}^{(0)} - \hat{\boldsymbol{W}}^{(0)} = \boldsymbol{0}$ , so that Eq. 14 can also be extended for $j = 0$,

$$\hat{\boldsymbol{W}}^{(j)} = \hat{\boldsymbol{W}}^{(0)} - \boldsymbol{L}^{-1}\boldsymbol{\Delta}^{(j)}, \quad 0 \le j \le c. \tag{16}$$

(1) Eq. 3 holds for $j = 1$:

Using Eqs. 6, 7, 8,

$$\boldsymbol{\omega}^{(1)} = \mathbf{e}_1^\top \boldsymbol{W}^{(0)} = \mathbf{e}_1^\top \hat{\boldsymbol{W}}^{(0)} = \hat{\boldsymbol{\omega}}^{(1)}. \tag{17}$$

(2) Assume Eq. 3 holds for all $j \le j_*, 1 \le j_* < c$.

Because $\boldsymbol{L}$ is a lower triangular matrix with all diagonal elements being 1, $\boldsymbol{L}^{-1}$ is also a lower triangular matrix with all diagonal elements being 1.

For $1 \le j < k \le c$,

$$\mathbf{e}_j^\top \boldsymbol{L}\mathbf{e}_k = \mathbf{e}_j^\top \boldsymbol{L}^{-1}\mathbf{e}_k = 0. \tag{18}$$

For $1 \le j \le c$,

$$\mathbf{e}_j^\top \boldsymbol{L}\mathbf{e}_j = \mathbf{e}_j^\top \boldsymbol{L}^{-1}\mathbf{e}_j = 1. \tag{19}$$

For $1 \le j < c$,

$$\begin{aligned}
&\mathbf{e}_{j+1}^\top \boldsymbol{L} \left( \sum_{k=1}^{j} \mathbf{e}_k \mathbf{e}_k^\top \right) \\
=&\mathbf{e}_{j+1}^\top \boldsymbol{L} \left( \left( \sum_{k=1}^{c} \mathbf{e}_k \mathbf{e}_k^\top \right) - \mathbf{e}_{j+1}\mathbf{e}_{j+1}^\top - \left( \sum_{k=j+2}^{c} \mathbf{e}_k \mathbf{e}_k^\top \right) \right) \\
=&\mathbf{e}_{j+1}^\top \boldsymbol{L} \left( \sum_{k=1}^{j+1} \mathbf{e}_k \mathbf{e}_k^\top \right) - \mathbf{e}_c^\top \boldsymbol{L}\mathbf{e}_{j+1}\mathbf{e}_{j+1}^\top - \mathbf{e}_{j+1}^\top \boldsymbol{L} \left( \sum_{k=j+2}^{c} \mathbf{e}_k \mathbf{e}_k^\top \right) \\
=&\mathbf{e}_{j+1}^\top \boldsymbol{L}\mathbf{I} - \mathbf{e}_{j+1}^\top - \left( \sum_{k=j+2}^{c} \mathbf{e}_{j+1}^\top \boldsymbol{L}\mathbf{e}_k \mathbf{e}_k^\top \right) \quad \text{(Eq. 19)} \\
=&\mathbf{e}_{j+1}^\top \boldsymbol{L} - \mathbf{e}_{j+1}^\top - \left( \sum_{k=j+2}^{c} 0\mathbf{e}_k^\top \right) \quad \text{(Eq. 18)} \\
=&\mathbf{e}_{j+1}^\top \left( \boldsymbol{L} - \mathbf{I} \right).
\end{aligned} \tag{20}$$

With Eq. 9, for $1 \le j \le c, 1 \le k \le c$ and $j \ne k$,

$$\begin{aligned}
\mathbf{e}_k^\top \boldsymbol{Q}^{(j)} =&\mathbf{e}_k^\top \left( \boldsymbol{Q}^{(j-1)} + \mathbf{e}_j \left( \mathbf{e}_j^\top \boldsymbol{Z}^{(j)} \operatorname{diag} \left( \boldsymbol{S}^\top \mathbf{e}_j \right) - \mathbf{e}_j^\top \boldsymbol{Q}^{(j-1)} \right) \right) \\
=&\mathbf{e}_k^\top \boldsymbol{Q}^{(j-1)} + \mathbf{e}_k^\top \mathbf{e}_j \left( \mathbf{e}_j^\top \boldsymbol{Z}^{(j)} \operatorname{diag} \left( \boldsymbol{S}^\top \mathbf{e}_j \right) - \mathbf{e}_j^\top \boldsymbol{Q}^{(j-1)} \right) \\
=&\mathbf{e}_k^\top \boldsymbol{Q}^{(j-1)} + 0 \left( \mathbf{e}_j^\top \boldsymbol{Z}^{(j)} \operatorname{diag} \left( \boldsymbol{S}^\top \mathbf{e}_j \right) - \mathbf{e}_j^\top \boldsymbol{Q}^{(j-1)} \right) \\
=&\mathbf{e}_k^\top \boldsymbol{Q}^{(j-1)}.
\end{aligned} \tag{21}$$

Recursively applying Eq. 21, for $1 \le j \le c, 1 \le k \le c$,

$$
\mathbf{e}_k^\top \boldsymbol{Q}^{(j)} = \begin{cases} \mathbf{e}_k^\top \boldsymbol{Q}^{(k)} & \text{if } 1 \le k \le j \le c, \\ \mathbf{e}_k^\top \boldsymbol{Q}^{(0)} = \mathbf{e}_k^\top \boldsymbol{W}^{(0)} & \text{if } 1 \le j < k \le c. \end{cases}
\tag{22}
$$

Similar to Eq. 22, with Eq. 10, for $1 \le j \le c, 1 \le k \le c$,

$$
\mathbf{e}_k^\top \hat{\boldsymbol{Q}}^{(j)} = \begin{cases} \mathbf{e}_k^\top \hat{\boldsymbol{Q}}^{(k)} & \text{if } 1 \le k \le j \le c, \\ \mathbf{e}_k^\top \hat{\boldsymbol{Q}}^{(0)} = \mathbf{e}_k^\top \hat{\boldsymbol{W}}^{(0)} & \text{if } 1 \le j < k \le c. \end{cases}
\tag{23}
$$

With Eq. 23, for $1 \le j \le c, 1 \le k \le c$,

$$
\begin{aligned}
\mathbf{e}_k^\top \boldsymbol{\Delta}^{(j)} =& \mathbf{e}_k^\top \left( \hat{\boldsymbol{Q}}^{(j)} - \hat{\boldsymbol{W}}^{(0)} \right) & \text{(Eq. 15)} \\
=& \mathbf{e}_k^\top \hat{\boldsymbol{Q}}^{(j)} - \mathbf{e}_k^\top \hat{\boldsymbol{W}}^{(0)} \\
=& \begin{cases} \mathbf{e}_k^\top \hat{\boldsymbol{Q}}^{(k)} - \mathbf{e}_k^\top \hat{\boldsymbol{W}}^{(0)} = \mathbf{e}_k^\top \boldsymbol{\Delta}^{(k)} & \text{if } 1 \le k \le j \le c, \\ \mathbf{e}_k^\top \hat{\boldsymbol{W}}^{(0)} - \mathbf{e}_k^\top \hat{\boldsymbol{W}}^{(0)} = \mathbf{e}_k^\top \boldsymbol{\Delta}^{(0)} = \mathbf{0} & \text{if } 1 \le j < k \le c. \end{cases}
\end{aligned}
\tag{24}
$$

For $1 \le k \le j \le c$,

$$
\begin{aligned}
& \mathbf{e}_k^\top \boldsymbol{L} \boldsymbol{\Delta}^{(j)} \\
=& \mathbf{e}_k^\top \boldsymbol{L} \boldsymbol{I} \boldsymbol{\Delta}^{(j)} \\
=& \mathbf{e}_k^\top \boldsymbol{L} \left( \sum_{k'=1}^{c} \mathbf{e}_{k'} \mathbf{e}_{k'}^\top \right) \boldsymbol{\Delta}^{(j)} \\
=& \sum_{k'=1}^{c} \mathbf{e}_k^\top \boldsymbol{L} \mathbf{e}_{k'} \mathbf{e}_{k'}^\top \boldsymbol{\Delta}^{(j)} \\
=& \left( \sum_{k'=1}^{k} \mathbf{e}_k^\top \boldsymbol{L} \mathbf{e}_{k'} \mathbf{e}_{k'}^\top \boldsymbol{\Delta}^{(j)} \right) + \left( \sum_{k'=k+1}^{c} \mathbf{e}_k^\top \boldsymbol{L} \mathbf{e}_{k'} \mathbf{e}_{k'}^\top \boldsymbol{\Delta}^{(j)} \right) \\
=& \left( \sum_{k'=1}^{k} \mathbf{e}_k^\top \boldsymbol{L} \mathbf{e}_{k'} \mathbf{e}_{k'}^\top \boldsymbol{\Delta}^{(k')} \right) + \left( \sum_{k'=k+1}^{c} \mathbf{0} \mathbf{e}_{k'}^\top \boldsymbol{\Delta}^{(j)} \right) & \text{(Eqs. 18, 24)} \\
=& \left( \sum_{k'=1}^{k} \mathbf{e}_k^\top \boldsymbol{L} \mathbf{e}_{k'} \mathbf{e}_{k'}^\top \boldsymbol{\Delta}^{(k)} \right) + \left( \sum_{k'=k+1}^{c} \mathbf{0} \mathbf{e}_{k'}^\top \boldsymbol{\Delta}^{(k)} \right) & \text{(Eq. 24)} \\
=& \left( \sum_{k'=1}^{k} \mathbf{e}_k^\top \boldsymbol{L} \mathbf{e}_{k'} \mathbf{e}_{k'}^\top \boldsymbol{\Delta}^{(k)} \right) + \left( \sum_{k'=k+1}^{c} \mathbf{e}_k^\top \boldsymbol{L} \mathbf{e}_{k'} \mathbf{e}_{k'}^\top \boldsymbol{\Delta}^{(k)} \right) & \text{(Eq. 18)} \\
=& \sum_{k'=1}^{c} \mathbf{e}_k^\top \boldsymbol{L} \mathbf{e}_{k'} \mathbf{e}_{k'}^\top \boldsymbol{\Delta}^{(k)} \\
=& \mathbf{e}_k^\top \boldsymbol{L} \left( \sum_{k'=1}^{c} \mathbf{e}_{k'} \mathbf{e}_{k'}^\top \right) \boldsymbol{\Delta}^{(k)} \\
=& \mathbf{e}_k^\top \boldsymbol{L} \boldsymbol{I} \boldsymbol{\Delta}^{(k)} \\
=& \mathbf{e}_k^\top \boldsymbol{L} \boldsymbol{\Delta}^{(k)}.
\end{aligned}
\tag{25}
$$

For $1 \leq j \leq c$,

$$
\begin{aligned}
&\mathbf{e}_j^\top \boldsymbol{L}^{-1} \boldsymbol{\Delta}^{(j-1)} \\
={}&\mathbf{e}_j^\top \boldsymbol{L}^{-1} \mathbf{I} \boldsymbol{\Delta}^{(j-1)} \\
={}&\mathbf{e}_j^\top \boldsymbol{L}^{-1} \left( \sum_{k=1}^{c} \mathbf{e}_k \mathbf{e}_k^\top \right) \boldsymbol{\Delta}^{(j-1)} \\
={}&\sum_{k=1}^{c} \mathbf{e}_j^\top \boldsymbol{L}^{-1} \mathbf{e}_k \mathbf{e}_k^\top \boldsymbol{\Delta}^{(j-1)} \\
={}&\left( \sum_{k=1}^{j-1} \mathbf{e}_j^\top \boldsymbol{L}^{-1} \mathbf{e}_k \mathbf{e}_k^\top \boldsymbol{\Delta}^{(j-1)} \right) + \mathbf{e}_j^\top \boldsymbol{L}^{-1} \mathbf{e}_j \mathbf{e}_j^\top \boldsymbol{\Delta}^{(j-1)} + \left( \sum_{k=j+1}^{c} \mathbf{e}_j^\top \boldsymbol{L}^{-1} \mathbf{e}_k \mathbf{e}_k^\top \boldsymbol{\Delta}^{(j-1)} \right) \\
={}&\left( \sum_{k=1}^{j-1} \mathbf{e}_j^\top \boldsymbol{L}^{-1} \mathbf{e}_k \mathbf{e}_k^\top \boldsymbol{\Delta}^{(j-1)} \right) + \mathbf{e}_j^\top \boldsymbol{L}^{-1} \mathbf{e}_j \mathbf{0} + \left( \sum_{k=j+1}^{c} \mathbf{0} \mathbf{e}_k^\top \boldsymbol{\Delta}^{(j-1)} \right)
\end{aligned}
$$

(Eqs. 18, 24)

$$
\begin{aligned}
={}&\left( \sum_{k=1}^{j-1} \mathbf{e}_j^\top \boldsymbol{L}^{-1} \mathbf{e}_k \mathbf{e}_k^\top \boldsymbol{\Delta}^{(j-1)} \right) + \left( \sum_{k=j+1}^{c} \mathbf{0} \mathbf{e}_k^\top \boldsymbol{\Delta}^{(j-1)} \right) + \mathbf{e}_j^\top \boldsymbol{\Delta}^{(j)} - \mathbf{e}_j^\top \boldsymbol{\Delta}^{(j)} \\
={}&\left( \sum_{k=1}^{j-1} \mathbf{e}_j^\top \boldsymbol{L}^{-1} \mathbf{e}_k \mathbf{e}_k^\top \boldsymbol{\Delta}^{(j)} \right) + \left( \sum_{k=j+1}^{c} \mathbf{e}_j^\top \boldsymbol{L}^{-1} \mathbf{e}_k \mathbf{e}_k^\top \boldsymbol{\Delta}^{(j)} \right) + \mathbf{e}_j^\top \boldsymbol{L}^{-1} \mathbf{e}_j \mathbf{e}_j^\top \boldsymbol{\Delta}^{(j)} - \mathbf{e}_j^\top \boldsymbol{\Delta}^{(j)}
\end{aligned}
$$

(Eqs. 19, 24)

$$
\begin{aligned}
={}&\left( \sum_{k=1}^{c} \mathbf{e}_j^\top \boldsymbol{L}^{-1} \mathbf{e}_k \mathbf{e}_k^\top \boldsymbol{\Delta}^{(j)} \right) - \mathbf{e}_j^\top \boldsymbol{\Delta}^{(j)} \\
={}&\mathbf{e}_j^\top \boldsymbol{L}^{-1} \left( \sum_{k=1}^{c} \mathbf{e}_k \mathbf{e}_k^\top \right) \boldsymbol{\Delta}^{(j)} - \mathbf{e}_j^\top \boldsymbol{\Delta}^{(j)} \\
={}&\mathbf{e}_j^\top \boldsymbol{L}^{-1} \mathbf{I} \boldsymbol{\Delta}^{(j)} - \mathbf{e}_j^\top \boldsymbol{\Delta}^{(j)} \\
={}&\mathbf{e}_j^\top \left( \boldsymbol{L}^{-1} - \mathbf{I} \right) \boldsymbol{\Delta}^{(j)}.
\end{aligned}
$$

(26)

According to the assumption, for $1 \leq k \leq j_* < c$, we have

$$
\mathbf{e}_k^\top \boldsymbol{W}^{(k-1)} = \boldsymbol{\omega}^{(k)} = \hat{\boldsymbol{\omega}}^{(k)} = \mathbf{e}_k^\top \hat{\boldsymbol{W}}^{(k-1)}
\tag{27}
$$

and

$$
\boldsymbol{Q}^{(k)} = \hat{\boldsymbol{Q}}^{(k)}.
\tag{28}
$$

For $1 \leq k \leq j_*$,

$$
\begin{aligned}
\varepsilon^{(k)} &= \mathbf{e}_k^\top \boldsymbol{Q}^{(k)} - \boldsymbol{\omega}^{(k)} && \text{(Eq. 11)} \\
&= \mathbf{e}_k^\top \boldsymbol{Q}^{(k)} - \mathbf{e}_k^\top \boldsymbol{W}^{(k-1)} \\
&= \mathbf{e}_k^\top \left( \boldsymbol{Q}^{(k)} - \boldsymbol{W}^{(k-1)} \right) \\
&= \mathbf{e}_k^\top \left( \hat{\boldsymbol{Q}}^{(k)} - \hat{\boldsymbol{W}}^{(k-1)} \right) && \text{(Eqs. 27, 28)} \\
&= \mathbf{e}_k^\top \left( \hat{\boldsymbol{Q}}^{(k)} - \left( \hat{\boldsymbol{W}}^{(0)} - \boldsymbol{L}^{-1} \boldsymbol{\Delta}^{(k-1)} \right) \right) && \text{(Eq. 16)} \\
&= \mathbf{e}_k^\top \left( \left( \hat{\boldsymbol{Q}}^{(k)} - \hat{\boldsymbol{W}}^{(0)} \right) + \boldsymbol{L}^{-1} \boldsymbol{\Delta}^{(k-1)} \right) \\
&= \mathbf{e}_k^\top \left( \boldsymbol{\Delta}^{(k)} + \boldsymbol{L}^{-1} \boldsymbol{\Delta}^{(k-1)} \right) && \text{(Eq. 15)} \\
&= \mathbf{e}_k^\top \left( \boldsymbol{\Delta}^{(k)} + \left( \boldsymbol{L}^{-1} - \mathbf{I} \right) \boldsymbol{\Delta}^{(k)} \right) && \text{(Eq. 26)} \\
&= \mathbf{e}_k^\top \boldsymbol{L}^{-1} \boldsymbol{\Delta}^{(k)} \\
&= \mathbf{e}_k^\top \boldsymbol{L}^{-1} \boldsymbol{\Delta}^{(j_*)} && \text{(Eq. 25).}
\end{aligned}
\tag{29}
$$

$$
\begin{aligned}
\boldsymbol{\omega}^{(j_*+1)} &= \mathbf{e}_{j_*+1}^\top \boldsymbol{W}^{(j_*)} && \text{(Eq. 7)} \\
&= \mathbf{e}_{j_*+1}^\top \left( \boldsymbol{W}^{(j_*-1)} + \boldsymbol{L} \mathbf{e}_{j_*} \varepsilon^{(j_*)} \right) && \text{(Eq. 13)} \\
&= \mathbf{e}_{j_*+1}^\top \left( \boldsymbol{W}^{(0)} + \left( \sum_{k=1}^{j_*} \boldsymbol{L} \mathbf{e}_k \varepsilon^{(k)} \right) \right) && \text{(Eq. 13)} \\
&= \mathbf{e}_{j_*+1}^\top \left( \hat{\boldsymbol{W}}^{(0)} + \left( \sum_{k=1}^{j_*} \boldsymbol{L} \mathbf{e}_k \mathbf{e}_k^\top \boldsymbol{L}^{-1} \boldsymbol{\Delta}^{(j_*)} \right) \right) && \text{(Eq. 29)} \\
&= \mathbf{e}_{j_*+1}^\top \left( \hat{\boldsymbol{W}}^{(0)} + \boldsymbol{L} \left( \sum_{k=1}^{j_*} \mathbf{e}_k \mathbf{e}_k^\top \right) \boldsymbol{L}^{-1} \boldsymbol{\Delta}^{(j_*)} \right) \\
&= \mathbf{e}_{j_*+1}^\top \left( \hat{\boldsymbol{W}}^{(0)} + \left( \boldsymbol{L} - \mathbf{I} \right) \boldsymbol{L}^{-1} \boldsymbol{\Delta}^{(j_*)} \right) && \text{(Eq. 20)} \\
&= \mathbf{e}_{j_*+1}^\top \left( \hat{\boldsymbol{W}}^{(0)} - \boldsymbol{L}^{-1} \boldsymbol{\Delta}^{(j_*)} + \boldsymbol{\Delta}^{(j_*)} \right) \\
&= \mathbf{e}_{j_*+1}^\top \left( \hat{\boldsymbol{W}}^{(0)} - \boldsymbol{L}^{-1} \boldsymbol{\Delta}^{(j_*)} + \mathbf{0} \right) && \text{(Eq. 24)} \\
&= \mathbf{e}_{j_*+1}^\top \left( \hat{\boldsymbol{W}}^{(0)} - \boldsymbol{L}^{-1} \boldsymbol{\Delta}^{(j_*)} \right) \\
&= \mathbf{e}_{j_*+1}^\top \hat{\boldsymbol{W}}^{(j_*)} && \text{(Eq. 16)} \\
&= \hat{\boldsymbol{\omega}}^{(j_*+1)} && \text{(Eq. 8).}
\end{aligned}
\tag{30}
$$

Eq. 3 holds for $j = j_* + 1$. $\blacksquare$

### C.2  STEP 2

Algorithm 7 (back-to-front order) is generated by reversing the iteration direction of Algorithm 6. Besides changing the direction of the index $j$, we also need to change the LDL decomposition to a so-called "UDU" decomposition so that the error propagation is correctly applied to the not-yet-quantized weights in the front dimensions.

**Justification**

Let $\mathbf{P}$ be the anti-diagonal permutation matrix with $\mathbf{P} = \mathbf{P}^\top = \mathbf{P}^{-1}$. Let $\hat{\boldsymbol{L}}$ be the LDL decomposition of the permuted Hessian inverse $\mathbf{P} \boldsymbol{H}^{-1} \mathbf{P} = \hat{\boldsymbol{L}} \hat{\boldsymbol{D}}_{\mathrm{L}}^{\frac{1}{2}} \hat{\boldsymbol{D}}_{\mathrm{L}}^{\frac{1}{2}} \hat{\boldsymbol{L}}^\top$ where $\hat{\boldsymbol{L}}$ is a lower triangular matrix with all diagonal elements being 1, and $\hat{\boldsymbol{D}}_{\mathrm{L}}^{\frac{1}{2}}$ is a non-negative diagonal matrix.

Since we are changing the iteration direction instead of applying the permutation, we permute the matrix $\hat{L}$ back, yielding $U = P\hat{L}P$. Alternatively, $U$ can be calculated using the decomposition $H^{-1} = P\hat{L}PP\hat{D}_L^{\frac{1}{2}}PP\hat{D}_L^{\frac{1}{2}}PP\hat{L}^\top P = UD_U^{\frac{1}{2}}D_U^{\frac{1}{2}}U^\top$ where $U$ is an upper triangular matrix with all diagonal elements being 1, and $D_U^{\frac{1}{2}} = P\hat{D}_L^{\frac{1}{2}}P$ is a non-negative diagonal matrix.

The decomposition to calculate $U$ from $H^{-1}$ is what we call "UDU" decomposition, which can be considered as a variant of the LDL decomposition.

## C.3 STEP 3

To distinguish the variables using the same symbol in Algorithms 7 and 8, we use symbols with ˆ to denote the symbols in Algorithm 7, and use the symbols with ˜ for Algorithm 8.

We have the Cholesky decomposition of $H$: $H = \left(H^{-1}\right)^{-1} = \left(UD_U^{\frac{1}{2}}D_U^{\frac{1}{2}}U^\top\right)^{-1} = \left(D_U^{-\frac{1}{2}}U^{-1}\right)^\top D_U^{-\frac{1}{2}}U^{-1}$, so that $A = D_U^{-\frac{1}{2}}U^{-1}$.

**Claim**

$$\hat{\omega}_j = \tilde{\omega}_j, \quad 1 \leq j \leq c, \tag{31}$$

and consequently,

$$\hat{Z}^{(j)} = \tilde{Z}^{(j)}, \quad 1 \leq j \leq c+1, \tag{32}$$

and

$$\hat{Q}^{(j)} = \tilde{Q}^{(j)}, \quad 1 \leq j \leq c+1. \tag{33}$$

**Proof Eq. 31 by Induction**

For $1 \leq j \leq c$,

$$
\begin{aligned}
\tilde{\omega}^{(j)} &= \frac{\mathbf{e}_j^\top Y^{(j+1)}}{\mathbf{e}_j^\top A\mathbf{e}_j} \\
&= \frac{\mathbf{e}_j^\top Y^{(j+1)}}{\mathbf{e}_j^\top D_U^{-\frac{1}{2}}U^{-1}\mathbf{e}_j} \\
&= \frac{\mathbf{e}_j^\top Y^{(j+1)}}{D_U^{-\frac{1}{2}}[j,j]} \\
&= D_U^{\frac{1}{2}}[j,j]\mathbf{e}_j^\top Y^{(j+1)} \\
&= \mathbf{e}_j^\top D_U^{\frac{1}{2}}Y^{(j+1)}.
\end{aligned}
\tag{34}
$$

The following equalities are held by the design of Algorithms 6 and 8:

$$\hat{Q}^{(c+1)} = \tilde{Q}^{(c+1)} = \hat{W}^{(c+1)} = \tilde{W}. \tag{35}$$

$$Y^{(c+1)} = A\tilde{W} = D_U^{-\frac{1}{2}}U^{-1}\tilde{W}. \tag{36}$$

$$\hat{\omega}^{(j)} = \mathbf{e}_j^\top \hat{W}^{(j+1)}, \quad 1 \leq j \leq c. \tag{37}$$

$$\hat{Q}^{(j)} = \hat{Q}^{(j+1)} + \mathbf{e}_j \left(\mathbf{e}_j^\top \hat{Z}^{(j)} \operatorname{diag}\left(S^\top \mathbf{e}_j\right) - \mathbf{e}_j^\top \hat{Q}^{(j+1)}\right), \quad 1 \leq j \leq c. \tag{38}$$

$$\tilde{Q}^{(j)} = \tilde{Q}^{(j+1)} + \mathbf{e}_j \left(\mathbf{e}_j^\top \tilde{Z}^{(j)} \operatorname{diag}\left(S^\top \mathbf{e}_j\right) - \mathbf{e}_j^\top \tilde{Q}^{(j+1)}\right), \quad 1 \leq j \leq c. \tag{39}$$

$$\Delta^{(j)} = \hat{Q}^{(j)} - \hat{W}^{(c+1)}, \quad 1 \leq j \leq c. \tag{40}$$

$$\hat{W}^{(j)} = \hat{W}^{(c+1)} - U^{-1}\Delta^{(j)}, \quad 1 \leq j \leq c. \tag{41}$$

$$Y^{(j)} = Y^{(j+1)} - A\mathbf{e}_j\mathbf{e}_j^\top \tilde{Q}^{(j)} = Y^{(j+1)} - D_U^{-\frac{1}{2}}U^{-1}\mathbf{e}_j\mathbf{e}_j^\top \tilde{Q}^{(j)}, \quad 1 \leq j \leq c. \tag{42}$$

Because $U$ is an upper triangular matrix with all diagonal elements being 1, $U^{-1}$ is also an upper triangular matrix with all diagonal elements being 1.

For $1 \leq k < j \leq c$,

$$\mathbf{e}_j^\top U \mathbf{e}_k = \mathbf{e}_j^\top U^{-1} \mathbf{e}_k = 0. \tag{43}$$

$$\mathbf{e}_c^\top U = \mathbf{e}_c^\top. \tag{44}$$

For $1 \leq j \leq c$,

$$\mathbf{e}_j^\top U \mathbf{e}_j = \mathbf{e}_j^\top U^{-1} \mathbf{e}_j = 1. \tag{45}$$

(1) Eq. 31 holds for $j = c$:

Using Eqs. 34, 35, 36, 37, 44,

$$\tilde{\boldsymbol{\omega}}^{(c)} = \mathbf{e}_c^\top \boldsymbol{D}_\mathrm{U}^{\frac{1}{2}} \boldsymbol{Y}^{(c+1)} = \mathbf{e}_c^\top \boldsymbol{D}_\mathrm{U}^{\frac{1}{2}} \boldsymbol{D}_\mathrm{U}^{-\frac{1}{2}} \boldsymbol{U}^{-1} \tilde{\boldsymbol{W}} = \mathbf{e}_c^\top \boldsymbol{U}^{-1} \tilde{\boldsymbol{W}} = \mathbf{e}_c^\top \tilde{\boldsymbol{W}} = \mathbf{e}_c^\top \hat{\boldsymbol{W}}^{(c+1)} = \hat{\boldsymbol{\omega}}^{(c)}. \tag{46}$$

(2) Assume Eq. 31 holds for all $j \geq j_*$, $1 < j_* \leq c$.

With Eq. 38, for $1 \leq j \leq c, 1 \leq k \leq c$ and $j \neq k$,

$$
\begin{aligned}
\mathbf{e}_k^\top \hat{\boldsymbol{Q}}^{(j)} &= \mathbf{e}_k^\top \left( \hat{\boldsymbol{Q}}^{(j+1)} + \mathbf{e}_j \left( \mathbf{e}_j^\top \boldsymbol{Z}^{(j)} \operatorname{diag} \left( \boldsymbol{S}^\top \mathbf{e}_j \right) - \mathbf{e}_j^\top \hat{\boldsymbol{Q}}^{(j+1)} \right) \right) \\
&= \mathbf{e}_k^\top \hat{\boldsymbol{Q}}^{(j+1)} + \mathbf{e}_k^\top \mathbf{e}_j \left( \mathbf{e}_j^\top \boldsymbol{Z}^{(j)} \operatorname{diag} \left( \boldsymbol{S}^\top \mathbf{e}_j \right) - \mathbf{e}_j^\top \hat{\boldsymbol{Q}}^{(j+1)} \right) \\
&= \mathbf{e}_k^\top \hat{\boldsymbol{Q}}^{(j+1)} + 0 \left( \mathbf{e}_j^\top \boldsymbol{Z}^{(j)} \operatorname{diag} \left( \boldsymbol{S}^\top \mathbf{e}_j \right) - \mathbf{e}_j^\top \hat{\boldsymbol{Q}}^{(j+1)} \right) \\
&= \mathbf{e}_k^\top \hat{\boldsymbol{Q}}^{(j+1)}.
\end{aligned}
\tag{47}
$$

Recursively applying Eq. 47, for $1 \leq j \leq c, 1 \leq k \leq c$,

$$
\mathbf{e}_k^\top \hat{\boldsymbol{Q}}^{(j)} = \begin{cases} \mathbf{e}_k^\top \hat{\boldsymbol{Q}}^{(k)} & \text{if } 1 \leq j \leq k \leq c, \\ \mathbf{e}_k^\top \hat{\boldsymbol{Q}}^{(c+1)} = \mathbf{e}_k^\top \hat{\boldsymbol{W}}^{(c+1)} & \text{if } 1 \leq k < j \leq c. \end{cases}
\tag{48}
$$

Similar to Eq. 48, with Eq. 39, for $1 \leq j \leq c, 1 \leq k \leq c$,

$$
\mathbf{e}_k^\top \tilde{\boldsymbol{Q}}^{(j)} = \begin{cases} \mathbf{e}_k^\top \tilde{\boldsymbol{Q}}^{(k)} & \text{if } 1 \leq j \leq k \leq c, \\ \mathbf{e}_k^\top \tilde{\boldsymbol{Q}}^{(c+1)} = \mathbf{e}_k^\top \tilde{\boldsymbol{W}} & \text{if } 1 \leq k < j \leq c. \end{cases}
\tag{49}
$$

For $1 \leq j \leq c$,

$$
\begin{aligned}
\boldsymbol{Y}^{(j)} &= \boldsymbol{Y}^{(j+1)} - \boldsymbol{D}_\mathrm{U}^{-\frac{1}{2}} \boldsymbol{U}^{-1} \mathbf{e}_j \mathbf{e}_j^\top \tilde{\boldsymbol{Q}}^{(j)} && \text{(Eq. 42)} \\
&= \boldsymbol{Y}^{(c+1)} - \left( \sum_{k=j}^{c} \boldsymbol{D}_\mathrm{U}^{-\frac{1}{2}} \boldsymbol{U}^{-1} \mathbf{e}_k \mathbf{e}_k^\top \tilde{\boldsymbol{Q}}^{(k)} \right) && \text{(Eq. 42)} \\
&= \boldsymbol{D}_\mathrm{U}^{-\frac{1}{2}} \boldsymbol{U}^{-1} \tilde{\boldsymbol{W}} - \left( \sum_{k=j}^{c} \boldsymbol{D}_\mathrm{U}^{-\frac{1}{2}} \boldsymbol{U}^{-1} \mathbf{e}_k \mathbf{e}_k^\top \tilde{\boldsymbol{Q}}^{(j)} \right) && \text{(Eq. 36)} \\
&= \boldsymbol{D}_\mathrm{U}^{-\frac{1}{2}} \boldsymbol{U}^{-1} \left( \tilde{\boldsymbol{W}} - \left( \sum_{k=j}^{c} \mathbf{e}_k \mathbf{e}_k^\top \right) \tilde{\boldsymbol{Q}}^{(j)} \right)
\end{aligned}
\tag{50}
$$

For $1 \leq j < c$,

$$\tilde{\boldsymbol{\omega}}^{(j)} = \mathbf{e}_j^\top \boldsymbol{D}_{\mathrm{U}}^{\frac{1}{2}} \boldsymbol{Y}^{(j+1)}$$

$$\text{(Eq. 34)}$$

$$= \mathbf{e}_j^\top \boldsymbol{D}_{\mathrm{U}}^{\frac{1}{2}} \boldsymbol{D}_{\mathrm{U}}^{-\frac{1}{2}} \boldsymbol{U}^{-1} \left( \tilde{\boldsymbol{W}} - \left( \sum_{k=j+1}^{c} \mathbf{e}_k \mathbf{e}_k^\top \right) \tilde{\boldsymbol{Q}}^{(j+1)} \right)$$

$$\text{(Eq. 50)}$$

$$= \mathbf{e}_j^\top \boldsymbol{U}^{-1} \left( \tilde{\boldsymbol{W}} - \left( \sum_{k=j+1}^{c} \mathbf{e}_k \mathbf{e}_k^\top \right) \tilde{\boldsymbol{Q}}^{(j+1)} \right)$$

$$= \mathbf{e}_j^\top \boldsymbol{U}^{-1} \tilde{\boldsymbol{W}} - \left( \sum_{k=j+1}^{c} \mathbf{e}_j^\top \boldsymbol{U}^{-1} \mathbf{e}_k \mathbf{e}_k^\top \right) \tilde{\boldsymbol{Q}}^{(j+1)}$$

$$= \mathbf{e}_j^\top \boldsymbol{U}^{-1} \tilde{\boldsymbol{W}} - \left( \left( \sum_{k=1}^{c} \mathbf{e}_j^\top \boldsymbol{U}^{-1} \mathbf{e}_k \mathbf{e}_k^\top \right) - \left( \sum_{k=1}^{j-1} \mathbf{e}_j^\top \boldsymbol{U}^{-1} \mathbf{e}_k \mathbf{e}_k^\top \right) - \mathbf{e}_j^\top \boldsymbol{U}^{-1} \mathbf{e}_j \mathbf{e}_j^\top \right) \tilde{\boldsymbol{Q}}^{(j+1)}$$

$$= \mathbf{e}_j^\top \boldsymbol{U}^{-1} \tilde{\boldsymbol{W}} - \left( \left( \sum_{k=1}^{c} \mathbf{e}_j^\top \boldsymbol{U}^{-1} \mathbf{e}_k \mathbf{e}_k^\top \right) - \left( \sum_{k=1}^{j-1} 0 \mathbf{e}_k^\top \right) - 1 \mathbf{e}_j^\top \right) \tilde{\boldsymbol{Q}}^{(j+1)}$$

$$\text{(Eqs. 43, 45)}$$

$$= \mathbf{e}_j^\top \boldsymbol{U}^{-1} \tilde{\boldsymbol{W}} - \left( \sum_{k=1}^{c} \mathbf{e}_j^\top \boldsymbol{U}^{-1} \mathbf{e}_k \mathbf{e}_k^\top \right) \tilde{\boldsymbol{Q}}^{(j+1)} + \mathbf{e}_j^\top \tilde{\boldsymbol{Q}}^{(j+1)}$$

$$= \mathbf{e}_j^\top \boldsymbol{U}^{-1} \tilde{\boldsymbol{W}} - \left( \sum_{k=1}^{c} \mathbf{e}_j^\top \boldsymbol{U}^{-1} \mathbf{e}_k \mathbf{e}_k^\top \right) \tilde{\boldsymbol{Q}}^{(j+1)} + \mathbf{e}_j^\top \tilde{\boldsymbol{W}}$$

$$\text{(Eq. 49)}$$

$$= \mathbf{e}_j^\top \left( \tilde{\boldsymbol{W}} - \boldsymbol{U}^{-1} \left( \left( \sum_{k=1}^{c} \mathbf{e}_k \mathbf{e}_k^\top \right) \tilde{\boldsymbol{Q}}^{(j+1)} - \tilde{\boldsymbol{W}} \right) \right)$$

$$= \mathbf{e}_j^\top \left( \tilde{\boldsymbol{W}} - \boldsymbol{U}^{-1} \left( \mathbf{I} \tilde{\boldsymbol{Q}}^{(j+1)} - \tilde{\boldsymbol{W}} \right) \right)$$

$$= \mathbf{e}_j^\top \left( \tilde{\boldsymbol{W}} - \boldsymbol{U}^{-1} \left( \tilde{\boldsymbol{Q}}^{(j+1)} - \tilde{\boldsymbol{W}} \right) \right).$$

$$\text{(51)}$$

Because $\mathbf{e}_c^\top \left( \tilde{\boldsymbol{W}} - \boldsymbol{U}^{-1} \left( \tilde{\boldsymbol{Q}}^{(c+1)} - \tilde{\boldsymbol{W}} \right) \right) = \mathbf{e}_c^\top \tilde{\boldsymbol{W}} = \tilde{\boldsymbol{\omega}}^{(c)}$, Eq. 51 can be extended for $j = c$,

$$\tilde{\boldsymbol{\omega}}^{(j)} = \mathbf{e}_j^\top \left( \tilde{\boldsymbol{W}} - \boldsymbol{U}^{-1} \left( \tilde{\boldsymbol{Q}}^{(j+1)} - \tilde{\boldsymbol{W}} \right) \right), \quad 1 \leq j \leq c. \tag{52}$$

According to the assumption, for $1 < j_* \leq k \leq c$, we have

$$\hat{\boldsymbol{Q}}^{(k)} = \tilde{\boldsymbol{Q}}^{(k)}. \tag{53}$$

$$\begin{aligned}
\tilde{\boldsymbol{\omega}}^{(j_*-1)} &= \mathbf{e}_{j_*-1}^\top \left( \tilde{\boldsymbol{W}} - \boldsymbol{U}^{-1} \left( \tilde{\boldsymbol{Q}}^{(j_*)} - \tilde{\boldsymbol{W}} \right) \right) & \text{(Eq. 52)} \\
&= \mathbf{e}_{j_*-1}^\top \left( \hat{\boldsymbol{W}}^{(c+1)} - \boldsymbol{U}^{-1} \left( \hat{\boldsymbol{Q}}^{(j_*)} - \hat{\boldsymbol{W}}^{(c+1)} \right) \right) & \text{(Eq. 53)} \\
&= \mathbf{e}_{j_*-1}^\top \left( \hat{\boldsymbol{W}}^{(c+1)} - \boldsymbol{U}^{-1} \boldsymbol{\Delta}^{(j_*)} \right) & \text{(Eq. 40)} \\
&= \mathbf{e}_{j_*-1}^\top \hat{\boldsymbol{W}}^{(j_*)} & \text{(Eq. 41)} \\
&= \hat{\boldsymbol{\omega}}^{(j_*-1)} & \text{(Eq. 37).}
\end{aligned} \tag{54}$$

Eq. 31 holds for $j = j_* - 1$. ∎

### C.4 PROOF OF INEFFECTIVENESS OF ADDITIONAL GPTQ REFINEMENT ON BABAI'S ALGORITHM

We may try to apply further GPTQ updates in Babai's algorithm by changing Line 9 in Algorithm 8 to

$$\boldsymbol{Y}'^{(j)} \leftarrow \boldsymbol{Y}^{(j)} + \boldsymbol{A}\boldsymbol{U}\mathbf{e}_j\boldsymbol{\varepsilon}^{(j)} = \boldsymbol{Y}^{(j+1)} - \boldsymbol{A}\mathbf{e}_j\mathbf{e}_j^\top\tilde{\boldsymbol{Q}}^{(j)} + \boldsymbol{A}\boldsymbol{U}\mathbf{e}_j\boldsymbol{\varepsilon}^{(j)} \tag{55}$$

However, as $\boldsymbol{A} = \boldsymbol{D}_{\mathrm{U}}^{-\frac{1}{2}}\boldsymbol{U}^{-1}$, the $\tilde{\boldsymbol{\omega}}^{(j-1)}$ remains the same:

$$
\begin{aligned}
\tilde{\boldsymbol{\omega}}'^{(j-1)} &= \mathbf{e}_{j-1}^\top \boldsymbol{D}_{\mathrm{U}}^{\frac{1}{2}} \boldsymbol{Y}'^{(j)} && \text{(Eq. 34)} \\
&= \mathbf{e}_{j-1}^\top \boldsymbol{D}_{\mathrm{U}}^{\frac{1}{2}} \left( \boldsymbol{Y}^{(j)} + \boldsymbol{D}_{\mathrm{U}}^{-\frac{1}{2}}\boldsymbol{U}^{-1}\boldsymbol{U}\mathbf{e}_j\boldsymbol{\varepsilon}^{(j)} \right) \\
&= \mathbf{e}_{j-1}^\top \boldsymbol{D}_{\mathrm{U}}^{\frac{1}{2}} \boldsymbol{Y}^{(j)} + \mathbf{e}_{j-1}^\top \boldsymbol{D}_{\mathrm{U}}^{\frac{1}{2}} \boldsymbol{D}_{\mathrm{U}}^{-\frac{1}{2}}\boldsymbol{U}^{-1}\boldsymbol{U}\mathbf{e}_j\boldsymbol{\varepsilon}^{(j)} \\
&= \mathbf{e}_{j-1}^\top \boldsymbol{D}_{\mathrm{U}}^{\frac{1}{2}} \boldsymbol{Y}^{(j)} + \mathbf{e}_{j-1}^\top \mathbf{e}_j\boldsymbol{\varepsilon}^{(j)} && (56) \\
&= \mathbf{e}_{j-1}^\top \boldsymbol{D}_{\mathrm{U}}^{\frac{1}{2}} \boldsymbol{Y}^{(j)} + 0\boldsymbol{\varepsilon}^{(j)} \\
&= \mathbf{e}_{j-1}^\top \boldsymbol{D}_{\mathrm{U}}^{\frac{1}{2}} \boldsymbol{Y}^{(j)} \\
&= \tilde{\boldsymbol{\omega}}^{(j-1)} && \text{(Eq. 34).}
\end{aligned}
$$

$\blacksquare$

## D    FURTHER DISCUSSION ON QUANTIZATION ERROR BOUND

### D.1    PROOF OF ABSOLUTE AND RELATIVE GPTQ QUANTIZATION ERROR BOUNDS

We prove Theorem 5 as follows.

Denote the basis $\boldsymbol{B}_{(i)} = \boldsymbol{\mathcal{X}} \operatorname{diag}(\boldsymbol{s}_i)$, $\boldsymbol{y}_{(i)} = \boldsymbol{\mathcal{X}} \boldsymbol{w}_i$ as in Section 4.1 so that the quantization problem becomes the CVP minimizing $\left\| \boldsymbol{B}_{(i)} \boldsymbol{z}_i - \boldsymbol{y}_{(i)} \right\|^2$. Applying permutation $\boldsymbol{T}$ gives the permuted basis $\boldsymbol{A}_{(i)} = \boldsymbol{B}_{(i)} \boldsymbol{T} = \boldsymbol{\mathcal{X}} \operatorname{diag}(\boldsymbol{s}_i) \boldsymbol{T} = \boldsymbol{\mathcal{X}} \boldsymbol{T} \operatorname{diag}(\boldsymbol{T}^{-1} \boldsymbol{s}_i)$. Write the unnormalized Gram-Schmidt vectors of $\boldsymbol{A}_{(i)}$ as $\tilde{\boldsymbol{A}}_{(i)} = [\tilde{\boldsymbol{a}}_{(i)1}, \ldots, \tilde{\boldsymbol{a}}_{(i)c}]$. Babai's guarantee therefore yields the tight bound $\left\| \boldsymbol{B}_{(i)} \boldsymbol{z}_i - \boldsymbol{y}_{(i)} \right\|^2 = \left\| \boldsymbol{A}_{(i)} (\boldsymbol{T}^{-1} \boldsymbol{z}_i) - \boldsymbol{y}_{(i)} \right\|^2 \leq \frac{1}{4} \sum_{j=1}^{c} \left\| \tilde{\boldsymbol{a}}_{(i)j} \right\|^2$.

We may, without loss of generality, use Theorem 1 to rotate $\boldsymbol{\mathcal{X}}$ so that $\boldsymbol{A}_{(i)}$ is upper triangular. In that case, the norm $\left\| \tilde{\boldsymbol{a}}_{(i)j} \right\|$ simplifies to $\left| \boldsymbol{A}_{(i)}[j,j] \right|$. Let $\boldsymbol{D}_{(i)}$ be the diagonal matrix of the LDL decomposition of $\boldsymbol{A}_{(i)}^{\top} \boldsymbol{A}_{(i)}$ such that $\boldsymbol{D}_{(i)}[j,j] = \left| \boldsymbol{A}_{(i)}[j,j] \right|^2 = \left\| \tilde{\boldsymbol{a}}_{(i)j} \right\|^2$. The summation $\sum_{j=1}^{c} \left\| \tilde{\boldsymbol{a}}_{(i)j} \right\|^2$ can then be expressed as $\operatorname{tr}(\boldsymbol{D}_{(i)})$. Let $\boldsymbol{\mathcal{L}}$ be the lower triangular matrix in the LDL decomposition of $\boldsymbol{T}^{\top} \boldsymbol{X}^{\top} \boldsymbol{X} \boldsymbol{T} = \boldsymbol{\mathcal{L}} \boldsymbol{\mathcal{D}} \boldsymbol{\mathcal{L}}^{\top}$, so that the LDL decomposition of $\boldsymbol{A}_{(i)}^{\top} \boldsymbol{A}_{(i)} = \operatorname{diag}(\boldsymbol{T}^{-1} \boldsymbol{s}_i) \boldsymbol{T}^{\top} \boldsymbol{X}^{\top} \boldsymbol{X} \boldsymbol{T} \operatorname{diag}(\boldsymbol{T}^{-1} \boldsymbol{s}_i) = \boldsymbol{\mathcal{L}}_{(i)} \boldsymbol{D}_{(i)} \boldsymbol{\mathcal{L}}_{(i)}^{\top}$ has $\boldsymbol{D}_{(i)} = \operatorname{diag}(\boldsymbol{T}^{-1} \boldsymbol{s}_i) \boldsymbol{D} \operatorname{diag}(\boldsymbol{T}^{-1} \boldsymbol{s}_i)$ and $\boldsymbol{\mathcal{L}}_{(i)} = \operatorname{diag}(\boldsymbol{T}^{-1} \boldsymbol{s}_i) \boldsymbol{\mathcal{L}} \operatorname{diag}(\boldsymbol{T}^{-1} \boldsymbol{s}_i)^{-1}$. The absolute no-clipping error bound is therefore $\frac{1}{4} \sum_{j=1}^{c} \left\| \tilde{\boldsymbol{a}}_{(i)j} \right\|^2 = \frac{1}{4} \operatorname{tr}(\boldsymbol{D}_{(i)}) = \frac{1}{4} (\boldsymbol{T}^{-1} \boldsymbol{s}_i)^{\top} \boldsymbol{D} (\boldsymbol{T}^{-1} \boldsymbol{s}_i)$.

For the relative no-clipping quantization error bound, we can plug in $\left\| \tilde{\boldsymbol{a}}_{(i)j} \right\| = \left| \boldsymbol{A}_{(i)}[j,j] \right| = \sqrt{\boldsymbol{D}_{(i)}[j,j]} = \sqrt{(\operatorname{diag}(\boldsymbol{T}^{-1} \boldsymbol{s}_i) \boldsymbol{D} \operatorname{diag}(\boldsymbol{T}^{-1} \boldsymbol{s}_i))[j,j]} = \sqrt{\boldsymbol{D}[j,j]} \left| (\boldsymbol{T}^{-1} \boldsymbol{s}_i)[j] \right| := d_j$ into Babai's relative error bound in Section 3.2.

### D.2    EXPECTED QUANTIZATION ERROR OVER A UNIFORM HYPER-CUBOID

We have shown that, when clipping is disabled, Babai's nearest-plane (hence back-to-front GPTQ) ensures the tight worst-case bound

$$\left\| \boldsymbol{X} \operatorname{diag}(\boldsymbol{s}_i) \boldsymbol{z}_i - \boldsymbol{X} \boldsymbol{w}_i \right\|^2 \leq \frac{1}{4} \sum_{j=1}^{c} \left\| \tilde{\boldsymbol{a}}_j \right\|^2, \quad \tilde{\boldsymbol{A}} = [\tilde{\boldsymbol{a}}_1, \ldots, \tilde{\boldsymbol{a}}_c] \tag{57}$$

where $\tilde{\boldsymbol{a}}_j$ are the unnormalized Gram-Schmidt vectors of the permuted lattice basis $\boldsymbol{A}$.

Introduce the half-edge lengths

$$a_j = \frac{1}{2} \left\| \tilde{\boldsymbol{a}}_j \right\|, \quad j = 1, \ldots, c, \tag{58}$$

so that the Babai residual always lies in the axis-aligned hyper-cuboid $\prod_{j=1}^{c} [-a_j, a_j]$ and Eq. 57 is rewritten as

$$\epsilon_{\text{worst}} = \sum_{j=1}^{c} a_j^2. \tag{59}$$

**Uniform prior on the unknown weight vector.** Assume now that the continuous, not-yet-quantized weight offset $\boldsymbol{u} = \boldsymbol{X}(\boldsymbol{w}_i - \operatorname{diag}(\boldsymbol{s}_i)\boldsymbol{z}_i)$ is uniformly distributed inside this hyper-cuboid, i.e., each coordinate $u_j \sim \operatorname{Uniform}(-a_j, a_j)$ and the coordinates are independent. The squared error becomes the random variable

$$\epsilon = \sum_{j=1}^{c} u_j^2. \tag{60}$$

**Lemma 7** *For a scalar $u \sim \operatorname{Uniform}(-a, a)$ one has $\mathbb{E}[u^2] = \frac{a^2}{3}$.*

**Proof**

$$\mathbb{E}[u^2] = \frac{1}{2a} \int_{-a}^{a} u^2 \mathrm{d}u = \frac{1}{2a} \left[ \frac{1}{3} x^3 \right]_{-a}^{a} = \frac{a^2}{3}. \tag{61}$$

■

**Expected residual norm.** Using Lemma 7,

$$\mathbb{E}[\epsilon] = \sum_{j=1}^{c} \mathbb{E}\left[u_j^2\right] = \frac{1}{3} \sum_{j=1}^{c} a_j^2. \tag{62}$$

**Ratio to the worst-case bound.** Comparing Eq. 62 with Eq. 59 gives

$$\boxed{\mathbb{E}[\epsilon] = \frac{1}{3}\epsilon_{\text{worst}}} \quad \Longrightarrow \quad \mathbb{E}\left[\|\boldsymbol{X}\operatorname{diag}(\boldsymbol{s}_i)\,\boldsymbol{z}_i - \boldsymbol{X}\boldsymbol{w}_i\|^2\right] = \frac{1}{12} \sum_{j=1}^{c} \|\tilde{\boldsymbol{a}}_j\|^2. \tag{63}$$

Hence, under a uniform prior on the weights inside Babai's orthogonal hyper-cuboid, the average layer-wise quantization error is exactly $\frac{1}{3}$ of the worst-case guarantee stated in Theorem 5.

### D.3 EMPIRICAL VERIFICATION ON QUANTIZATION ORDER AND ERROR BOUND

Changing the quantization order alters the diagonal matrix $\boldsymbol{D}$ of the LDL decomposition of the permuted Hessian and therefore the no-clipping GPTQ/Babai bound (see Section 4.5). When per-group scales are approximately uniform, minimizing $\operatorname{tr}(\boldsymbol{D})$ is a good proxy for tightening this bound. To assess different orders (back-to-front, front-to-back, random order, GPTQ's act-order, and our min-pivot order), we run the calibration dataset from Section E.2 through the full-precision Qwen3-8B model and compute per-layer Hessians and calculate the $\operatorname{tr}(\boldsymbol{D})$. For the random order, we average the results over 100 runs. Table 2 reports $\operatorname{tr}(\boldsymbol{D})$ for the layers in transformer block 18; other blocks and models show similar patterns. In block 18, act-order already reduces $\operatorname{tr}(\boldsymbol{D})$ relative to the back-to-front/front-to-back/random baselines, especially in the Q·K·V and Gate·Up layers ($\approx$35-50% lower). Our min-pivot heuristic consistently attains the smallest trace. In practice, this tightens the theoretical layer-wise error bound and yields modest but consistent improvements. We can use act-order as a cheap option and reserve min-pivot for cases where a tighter bound is required.

Table 2: $\operatorname{tr}(\boldsymbol{D})$ with different quantization orders of layers in Qwen3-8B block 18.

| Order | Q·K·V | O | Gate·Up | Down |
|---|---|---|---|---|
| back-to-front | 1.169e+08 | 1.824e+08 | 1.181e+08 | 1.323e+09 |
| front-to-back | 1.161e+08 | 1.841e+08 | 1.202e+08 | 1.320e+09 |
| random (averaged) | 1.168e+08 | 1.856e+08 | 1.194e+08 | 1.322e+09 |
| act-order | 7.400e+07 | 1.786e+08 | 6.052e+07 | 1.222e+09 |
| min-pivot | 7.323e+07 | 1.772e+08 | 5.990e+07 | 1.221e+09 |

## E  FURTHER APPLICATIONS AND EXPERIMENTAL RESULTS

### E.1  OVERFLOW-TOLERANT QUANTIZATION ALGORITHMS

Algorithms 9, 11 and 12 are the pseudocodes of our proposed SSQR, HPTQ, and HRTN algorithms in Section 5. Additional notations are as follows. $\rho \in [0, 1]$ is the target outlier rate in SSQR. $\Xi = [\xi_1, \ldots, \xi_r] \in \mathbb{R}^{c \times r}$ is the sparse weight matrix in SSQR. $h \in \mathbb{R}_{>0}$ is the target average bitwidth in HPTQ and HRTN.

---

**Algorithm 9:** SSQR

**Input:** $W, X, P, \lambda, \mathbb{Z}_\dagger, \rho$
**Output:** $Z, S, \Xi, Q$
1   $S_{\text{MSE}} \leftarrow$ compute the MSE scale using $W$ and $\mathbb{Z}_\dagger$
2   $s_{\min}, s_{\max} \leftarrow 0^r, 2^r$ // initialize the binary search boundary per output channel
3   $s \leftarrow (s_{\min} + s_{\max})/2$ // the scale for scale
4   **while** $s$ not converge **do**
5      $S \leftarrow S_{\text{MSE}} \operatorname{diag}(s)$ // output-channel-wisely proportionally adjust the scale
6      $Z, \Xi, Q \leftarrow \text{SSQRINNERPROCEDURE}(W, S, X, P, \lambda, \mathbb{Z}_\dagger)$ // Algorithm 10
7      $s_{\min}[i], s_{\max}[i] \leftarrow \begin{cases} s_{\min}[i], s[i] & \text{if } \|\Xi[:, i]\|_0 < \rho c \\ s[i], s_{\max}[i] & \text{otherwise} \end{cases}$    for $i \in \{1, \ldots, r\}$
8      $s \leftarrow (s_{\min} + s_{\max})/2$
9   **end**

---

**Algorithm 10:** SSQR Inner Procedure (GPTQ with overflowed elements in floating-point)

**Input:** $W, S, X, P, \lambda, \mathbb{Z}_\dagger$
**Output:** $Z, \Xi, Q$
1   $H \leftarrow P^\top (X^\top X + \lambda I) P$
2   $L \leftarrow \text{LDL}(H^{-1})$
3   $W, S \leftarrow P^{-1} W, P^{-1} S$
4   $Q, Z \leftarrow W, 0$
5   **for** $j \leftarrow 1$ to $c$ **do**
6      $\zeta \leftarrow W[j, :]/S[j, :]$
7      $Z[j, :] \leftarrow \text{ROUND}(\zeta, \mathbb{Z}_\dagger)$
8      $\Xi[j, i] \leftarrow \begin{cases} W[j, i] - Z[j, i] * S[j, i] & \text{if } Z[j, i] \neq \text{ROUND}(\zeta[i], \mathbb{Z}) \\ 0 & \text{otherwise} \end{cases}$ // new
9      $Q[j, :] \leftarrow Z[j, :] * S[j, :] + \Xi[j, :]$ // new
10      $\varepsilon \leftarrow Q[j, :] - W[j, :]$
11      $W[j:, :] \leftarrow W[j:, :] + L[j:, j]\varepsilon$
12   **end**
13   $Z, \Xi, Q \leftarrow PZ, P\Xi, PQ$ // new

---

---

**Algorithm 11:** HPTQ

    **Input:** $\boldsymbol{W}, \boldsymbol{X}, \boldsymbol{P}, \lambda, h$
    **Output:** $\boldsymbol{Z}, s, \boldsymbol{Q}$
**1** $s_{\min}, s_{\max} \leftarrow 0, \|\boldsymbol{W}\|_\infty$ // initialize the binary search boundary
**2** $s \leftarrow (s_{\min} + s_{\max})/2$ // the scale
**3** **while** $s$ not converge **do**
**4**     $\boldsymbol{S} \leftarrow s \cdot \boldsymbol{1}^{c \times r}$ // broadcast the scale
**5**     $\boldsymbol{Z}, \boldsymbol{Q} \leftarrow \text{GPTQ}(\boldsymbol{W}, \boldsymbol{S}, \boldsymbol{X}, \boldsymbol{P}, \lambda, \mathbb{Z})$ // Algorithm 1
**6**     $h' \leftarrow$ average Huffman encoding bitwidth of $\boldsymbol{Z}$
**7**     **if** $h' < h$ **then**
**8**         $s_{\max} \leftarrow s$ // too few bits, try smaller scale
**9**     **end**
**10**    **else**
**11**        $s_{\min} \leftarrow s$ // too many bits, try larger scale
**12**    **end**
**13**    $s \leftarrow (s_{\min} + s_{\max})/2$
**14** **end**

---

**Algorithm 12:** HRTN

    **Input:** $\boldsymbol{W}, h$
    **Output:** $\boldsymbol{Z}, s, \boldsymbol{Q}$
**1** $s_{\min}, s_{\max} \leftarrow 0, \|\boldsymbol{W}\|_\infty$ // initialize the binary search boundary with min and max
**2** $s \leftarrow (s_{\min} + s_{\max})/2$ // the scale
**3** **while** $s$ not converge **do**
**4**     $\boldsymbol{Z} \leftarrow \text{ROUND}(\boldsymbol{W}/s, \mathbb{Z})$ // round-to-nearest
**5**     $\boldsymbol{Q} \leftarrow s\boldsymbol{Z}$
**6**     $h' \leftarrow$ average Huffman encoding bitwidth of $\boldsymbol{Z}$
**7**     **if** $h' < h$ **then**
**8**         $s_{\max} \leftarrow s$ // too few bits, try smaller scale
**9**     **end**
**10**    **else**
**11**        $s_{\min} \leftarrow s$ // too many bits, try larger scale
**12**    **end**
**13**    $s \leftarrow (s_{\min} + s_{\max})/2$
**14** **end**

---

## E.2 EXPERIMENT SETUP

We work with the Qwen3 family of models, which come in a range of sizes. We focus on the Qwen3-8B model for detailed head-to-head comparisons, while the other variants, Qwen3-0.6B, Qwen3-1.7B, Qwen3-4B, and Qwen3-14B, help us assess how our method performs across different model scales.

We construct the calibration dataset for the GPTQ algorithm using the FineWeb-Edu dataset (HuggingFaceFW/fineweb-edu, subset sample-10BT). The dataset is streamed and shuffled with a fixed seed for reproducibility. After tokenizing the text samples, our 256 sequences are accumulated into non-overlapping sequences of length 2048.

We use WikiText-2 and C4 for perplexity evaluations. For WikiText-2, the entire test split is first concatenated using two line breaks as separators and then tokenized with the default HuggingFace tokenizer for each model. For C4, we sample individual documents from the selected shard, tokenize them, and randomly extract sequences of the desired length. In both cases, sequences shorter than the target length (2048 tokens) are discarded, and sequences longer than the target length are cropped to the specified window.

### E.3 ACCURACY RESULTS FOR QWEN3 MODELS

We compare the perplexity results between RTN, GPTQ, HRTN, HPTQ, and SSQR using the Qwen3-8B model in Table 3. In addition, the perplexity results for other variants of Qwen3 with HPTQ are shown in Table 4.

Table 5 shows additional zero-shot results on the Qwen3-8B model for RTN, GPTQ, HRTN, and HPTQ. Additional HPTQ results on other Qwen3 models are in Tables 6 to 10.

Table 3: Perplexity of Qwen3-8B model under HPTQ, GPTQ, HRTN, RTN, and SSQR with different bitwidths.

| Method | Avg. Bitwidth | Perplexity | |
| --- | --- | --- | --- |
| | | WikiText-2 | C4 |
| BF16 Baseline | 16.000 | 9.73 | 13.55 |
| HPTQ | 4.125 | 9.81 | 13.64 |
| | 3.125 | 10.34 | 14.23 |
| | 2.125 | 13.97 | 16.89 |
| GPTQ | 4.125 | 10.10 | 13.92 |
| | 3.125 | 12.77 | 15.61 |
| | 2.125 | 57.51 | 36.14 |
| HRTN | 4.125 | 9.90 | 13.80 |
| | 3.125 | 10.75 | 14.63 |
| | 2.125 | 593.05 | 503.00 |
| RTN | 4.125 | 10.30 | 15.20 |
| | 3.125 | 16.30 | 21.08 |
| | 2.125 | 2e10 | 2e10 |
| SSQR-1% | 4.445 | 10.00 | 13.83 |
| | 3.445 | 10.64 | 14.71 |
| | 2.445 | 22.30 | 27.07 |
| SSQR-2% | 4.765 | 9.96 | 13.76 |
| | 3.765 | 10.57 | 14.56 |
| | 2.765 | 16.55 | 20.80 |
| SSQR-3% | 5.085 | 9.92 | 13.76 |
| | 4.085 | 10.42 | 14.32 |
| | 3.085 | 14.05 | 18.57 |
| SSQR-4% | 5.405 | 9.84 | 13.71 |
| | 4.405 | 10.34 | 14.29 |
| | 3.405 | 13.12 | 17.60 |
| SSQR-5% | 5.725 | 9.80 | 13.67 |
| | 4.725 | 10.32 | 14.22 |
| | 3.725 | 12.88 | 16.85 |

Table 4: Perplexity of Qwen3 models under HPTQ for different bitwidths.

| Model | Avg. Bitwidth | Perplexity | |
|---|---|---|---|
| | | WikiText-2 | C4 |
| 0.6B | 16.000 | 20.96 | 26.37 |
| | 4.125 | 22.72 | 28.35 |
| | 3.125 | 31.43 | 37.92 |
| | 2.125 | 156.45 | 171.38 |
| 1.7B | 16.000 | 16.72 | 19.92 |
| | 4.125 | 18.18 | 20.99 |
| | 3.125 | 19.72 | 23.15 |
| | 2.125 | 46.94 | 51.96 |
| 4B | 16.000 | 13.66 | 17.07 |
| | 4.125 | 14.26 | 17.39 |
| | 3.125 | 14.55 | 18.17 |
| | 2.125 | 24.40 | 26.46 |
| 8B | 16.000 | 9.73 | 13.55 |
| | 4.125 | 9.81 | 13.64 |
| | 3.125 | 10.34 | 14.23 |
| | 2.125 | 13.97 | 16.89 |
| 14B | 16.000 | 8.65 | 12.23 |
| | 4.125 | 8.76 | 12.12 |
| | 3.125 | 9.06 | 13.97 |
| | 2.125 | 11.36 | 15.50 |

Table 5: Zero-shot evaluation results (%) for Qwen3-8B under different quantization methods across six benchmarks.

| Method | Avg. Bitwidth | Wino | MMLU | PiQA | SciQ | TQA | |
| --- | --- | --- | --- | --- | --- | --- | --- |
| | | | | | | MC1 | MC2 |
| BF16 Baseline | 16.000 | 68.11 | 73.02 | 77.80 | 95.7 | 36.35 | 54.50 |
| HPTQ | 4.125 | 67.17 | 72.28 | 77.42 | 95.6 | 35.01 | 53.36 |
| | 3.125 | 66.93 | 70.96 | 77.53 | 95.4 | 36.11 | 54.73 |
| | 2.125 | 59.19 | 52.99 | 72.52 | 86.8 | 31.09 | 49.01 |
| GPTQ | 4.125 | 68.82 | 71.76 | 77.58 | 95.3 | 36.35 | 54.55 |
| | 3.125 | 68.35 | 65.80 | 75.46 | 75.46 | 36.11 | 55.21 |
| | 2.125 | 52.25 | 34.25 | 57.83 | 57.83 | 28.40 | 46.91 |
| HRTN | 4.125 | 67.56 | 72.15 | 76.99 | 94.2 | 36.47 | 56.46 |
| | 3.125 | 66.22 | 67.85 | 76.12 | 93.7 | 35.13 | 53.68 |
| | 2.125 | 51.22 | 33.91 | 65.78 | 76.8 | 30.48 | 51.78 |
| RTN | 4.125 | 67.17 | 69.71 | 75.90 | 94.5 | 36.84 | 55.77 |
| | 3.125 | 57.93 | 47.90 | 70.89 | 87.1 | 34.03 | 52.76 |
| | 2.125 | 49.08 | 22.95 | 51.63 | 21.2 | 24.11 | 47.33 |
| SSQR-1% | 4.445 | 68.43 | 72.12 | 77.04 | 95.2 | 37.58 | 55.81 |
| | 3.445 | 68.11 | 68.46 | 75.84 | 95.5 | 38.19 | 55.95 |
| | 2.445 | 51.85 | 26.71 | 61.64 | 69.8 | 28.40 | 43.88 |
| SSQR-2% | 4.765 | 67.25 | 72.27 | 77.97 | 95.5 | 35.62 | 53.47 |
| | 3.765 | 67.40 | 69.66 | 76.22 | 95.1 | 33.90 | 53.05 |
| | 2.765 | 55.72 | 37.48 | 66.76 | 83.8 | 27.54 | 45.54 |
| SSQR-3% | 5.085 | 67.72 | 71.89 | 77.53 | 95.6 | 36.47 | 54.46 |
| | 4.085 | 65.59 | 69.88 | 77.31 | 94.3 | 37.82 | 55.34 |
| | 3.085 | 59.19 | 49.32 | 69.59 | 86.4 | 29.50 | 48.53 |
| SSQR-4% | 5.405 | 69.53 | 72.63 | 77.31 | 95.1 | 36.23 | 53.60 |
| | 4.405 | 67.48 | 69.51 | 76.61 | 94.9 | 37.21 | 54.81 |
| | 3.405 | 61.25 | 54.07 | 72.80 | 89.5 | 31.33 | 50.46 |
| SSQR-5% | 5.725 | 68.27 | 72.23 | 77.42 | 95.2 | 35.86 | 53.76 |
| | 4.725 | 67.48 | 70.76 | 76.71 | 95.5 | 35.37 | 52.91 |
| | 3.725 | 62.59 | 58.67 | 73.23 | 90.8 | 31.21 | 50.25 |

Table 6: TruthfullQA (%) zero-shot results (MC1/MC2) for Qwen3 models quantized with HPTQ.

| Avg. Bitwidth | 0.6B | 1.7B | 4B | 8B | 14B |
| --- | --- | --- | --- | --- | --- |
| 16.000 | 27.17/42.80 | 29.50/45.88 | 37.33/54.83 | 36.35/54.50 | 40.76/58.62 |
| 4.125 | 26.19/41.56 | 28.76/45.17 | 36.72/54.46 | 35.01/53.36 | 40.51/58.28 |
| 3.125 | 25.34/41.95 | 29.62/46.13 | 35.25/53.83 | 36.11/54.73 | 39.90/58.33 |
| 2.125 | 23.99/46.39 | 28.15/48.25 | 31.70/50.67 | 31.09/49.01 | 36.84/54.93 |

Table 7: MMLU (%) zero-shot results for Qwen3 models quantized with HPTQ.

| Avg. Bitwidth | 0.6B | 1.7B | 4B | 8B | 14B |
|---|---|---|---|---|---|
| 16.000 | 40.34 | 55.44 | 68.38 | 73.02 | 77.10 |
| 4.125 | 29.84 | 53.95 | 67.45 | 72.28 | 76.27 |
| 3.125 | 32.92 | 47.49 | 62.70 | 70.96 | 75.53 |
| 2.125 | 24.58 | 23.87 | 40.83 | 52.99 | 64.31 |

Table 8: PiQA (%) zero-shot results for Qwen3 models quantized with HPTQ.

| Avg. Bitwidth | 0.6B | 1.7B | 4B | 8B | 14B |
|---|---|---|---|---|---|
| 16.000 | 67.30 | 72.31 | 74.92 | 77.80 | 79.87 |
| 4.125 | 66.00 | 70.78 | 75.30 | 77.42 | 79.54 |
| 3.125 | 62.08 | 68.44 | 73.01 | 77.53 | 78.78 |
| 2.125 | 54.13 | 57.40 | 66.76 | 72.52 | 75.46 |

Table 9: WinoGrande (%) zero-shot results for Qwen models quantized with HPTQ.

| Avg. Bitwidth | 0.6B | 1.7B | 4B | 8B | 14B |
|---|---|---|---|---|---|
| 16.000 | 56.43 | 61.48 | 65.27 | 68.11 | 72.53 |
| 4.125 | 54.38 | 59.67 | 64.09 | 67.17 | 73.01 |
| 3.125 | 52.72 | 58.72 | 64.80 | 66.93 | 71.19 |
| 2.125 | 49.80 | 49.96 | 53.04 | 59.19 | 66.06 |

Table 10: SciQ (%) zero-shot results for Qwen3 models quantized with HPTQ, with internal reasoning disabled.

| Avg. Bitwidth | 0.6B | 1.7B | 4B | 8B | 14B |
|---|---|---|---|---|---|
| 16.000 | 83.5 | 91.2 | 93.5 | 95.7 | 96.8 |
| 4.125 | 80.7 | 88.9 | 93.3 | 95.6 | 97.1 |
| 3.125 | 76.6 | 89.9 | 92 | 95.4 | 96.8 |
| 2.125 | 40.8 | 62.8 | 81.2 | 86.8 | 93.8 |

### E.4 ACCURACY RESULTS FOR LLAMA MODELS

Tables 11 to 15 report the evaluation results for Llama-3.2-3B-Instruct, Llama-3.1-8B-Instruct, and Llama-2-7B models under the same setups as in Section E.3.

Table 11: Perplexity of Llama-3.2-3B-Instruct model under HPTQ, GPTQ, and SSQR with different bitwidths.

| Method | Avg. Bitwidth | Perplexity | |
|---|---|---|---|
| | | WikiText-2 | C4 |
| BF16 Baseline | 16.000 | 11.01 | 13.49 |
| HPTQ | 4.125 | 11.27 | 14.64 |
| | 3.125 | 12.51 | 15.81 |
| | 2.125 | 22.58 | 29.82 |
| GPTQ | 4.125 | 11.96 | 15.37 |
| | 3.125 | 15.20 | 18.99 |
| | 2.125 | 357.69 | 172.89 |
| SSQR-1% | 4.445 | 11.38 | 14.95 |
| | 3.445 | 13.48 | 18.38 |
| | 2.445 | 83.41 | 67.19 |
| SSQR-2% | 4.765 | 11.50 | 14.77 |
| | 3.765 | 13.20 | 16.65 |
| | 2.765 | 45.93 | 41.69 |
| SSQR-3% | 5.085 | 11.39 | 14.64 |
| | 4.085 | 12.50 | 16.10 |
| | 3.085 | 37.41 | 30.74 |
| SSQR-4% | 5.405 | 11.53 | 14.69 |
| | 4.405 | 12.33 | 15.96 |
| | 3.405 | 23.74 | 27.59 |
| SSQR-5% | 5.725 | 11.47 | 14.69 |
| | 4.725 | 12.29 | 15.81 |
| | 3.725 | 22.94 | 25.44 |

Table 12: Perplexity of Llama-3.1-8B-Instruct model under HPTQ, GPTQ, and SSQR with different bitwidths.

| Method | Avg. Bitwidth | Perplexity | |
|---|---|---|---|
| | | WikiText-2 | C4 |
| BF16 Baseline | 16.000 | 7.20 | 9.09 |
| HPTQ | 4.125 | 7.37 | 9.99 |
| | 3.125 | 7.84 | 11.04 |
| | 2.125 | 11.89 | 16.37 |
| GPTQ | 4.125 | 7.56 | 10.46 |
| | 3.125 | 9.44 | 13.16 |
| | 2.125 | 148.15 | 71.33 |
| SSQR-1% | 4.445 | 7.50 | 10.30 |
| | 3.445 | 8.67 | 12.35 |
| | 2.445 | 57.26 | 39.96 |
| SSQR-2% | 4.765 | 7.48 | 10.20 |
| | 3.765 | 8.32 | 11.75 |
| | 2.765 | 25.18 | 25.21 |
| SSQR-3% | 5.085 | 7.41 | 10.11 |
| | 4.085 | 8.16 | 11.54 |
| | 3.085 | 17.27 | 20.03 |
| SSQR-4% | 5.405 | 7.39 | 10.05 |
| | 4.405 | 8.01 | 11.31 |
| | 3.405 | 13.22 | 17.77 |
| SSQR-5% | 5.725 | 7.38 | 10.03 |
| | 4.725 | 7.98 | 11.13 |
| | 3.725 | 12.12 | 16.13 |

Table 13: Perplexity of Llama-2-7B model under HPTQ, GPTQ, and SSQR-1% with different bitwidths.

| Method | Avg. Bitwidth | Perplexity | |
|---|---|---|---|
| | | WikiText-2 | C4 |
| FP16 Baseline | 16.000 | 5.50 | 6.24 |
| HPTQ | 4.125 | 5.53 | 6.73 |
| | 3.125 | 5.77 | 7.04 |
| | 2.125 | 7.45 | 9.43 |
| GPTQ | 4.125 | 5.70 | 6.90 |
| | 3.125 | 6.75 | 8.08 |
| | 2.125 | 28.07 | 26.13 |
| SSQR-1% | 4.445 | 5.60 | 6.81 |
| | 3.445 | 6.09 | 7.52 |
| | 2.445 | 14.58 | 15.85 |

Table 14: Zero-shot evaluation results (%) for Llama-3.2-3B-Instruct under different quantization methods.

| Method | Avg. Bitwidth | Wino | MMLU | PiQA | SciQ | HSwag | |
| --- | --- | --- | --- | --- | --- | --- | --- |
| | | | | | | acc | acc$_{norm}$ |
| BF16 Baseline | 16.000 | 68.75 | 62.18 | 76.17 | 95.4 | 53.27 | 71.65 |
| HPTQ | 4.125 | 68.03 | 61.57 | 76.55 | 95.0 | 53.02 | 71.28 |
| | 3.125 | 68.35 | 58.50 | 7497 | 95.5 | 51.76 | 70.00 |
| | 2.125 | 60.85 | 42.75 | 69.15 | 89.9 | 44.54 | 60.29 |
| GPTQ | 4.125 | 68.11 | 59.81 | 75.73 | 95.5 | 52.29 | 70.54 |
| | 3.125 | 66.06 | 49.13 | 72.58 | 94.0 | 47.25 | 63.93 |
| | 2.125 | 50.59 | 22.96 | 53.65 | 63.4 | 28.06 | 30.58 |
| SSQR-1% | 4.445 | 68.19 | 60.94 | 76.12 | 95.7 | 52.37 | 70.88 |
| | 3.445 | 66.93 | 54.10 | 74.92 | 95.6 | 50.55 | 68.86 |
| | 2.445 | 51.70 | 23.97 | 58.22 | 64.5 | 31.14 | 36.90 |
| SSQR-2% | 4.765 | 68.03 | 61.17 | 76.33 | 95.2 | 52.49 | 70.99 |
| | 3.765 | 65.51 | 56.37 | 74.43 | 94.4 | 50.88 | 68.85 |
| | 2.765 | 53.12 | 23.91 | 60.01 | 78.3 | 34.12 | 42.99 |
| SSQR-3% | 5.085 | 68.27 | 61.68 | 76.82 | 95.4 | 53.03 | 71.29 |
| | 4.085 | 66.69 | 57.65 | 75.03 | 95.0 | 50.98 | 69.00 |
| | 3.085 | 58.48 | 34.20 | 65.61 | 90.5 | 39.87 | 52.43 |
| SSQR-4% | 5.405 | 68.90 | 61.11 | 76.28 | 95.5 | 52.80 | 71.03 |
| | 4.405 | 66.77 | 57.73 | 75.03 | 95.3 | 51.08 | 68.79 |
| | 3.405 | 57.85 | 33.74 | 66.49 | 90.1 | 40.66 | 54.44 |
| SSQR-5% | 5.725 | 68.35 | 61.67 | 75.57 | 95.3 | 52.88 | 70.97 |
| | 4.725 | 66.69 | 57.02 | 75.52 | 95.3 | 51.32 | 69.70 |
| | 3.725 | 57.38 | 37.24 | 65.56 | 91.5 | 41.37 | 54.96 |

Table 15: Zero-shot evaluation results (%) for Llama-3.1-8B-Instruct under different quantization methods.

| Method | Avg. Bitwidth | Wino | MMLU | PiQA | SciQ | HSwag | |
| --- | --- | --- | --- | --- | --- | --- | --- |
| | | | | | | acc | $acc_{norm}$ |
| BF16 Baseline | 16.000 | 73.72 | 68.31 | 80.14 | 97.3 | 59.81 | 79.59 |
| HPTQ | 4.125 | 73.56 | 67.90 | 79.49 | 97.7 | 59.57 | 79.25 |
| | 3.125 | 72.77 | 64.58 | 79.16 | 96.9 | 58.42 | 78.21 |
| | 2.125 | 63.69 | 45.01 | 69.15 | 90.8 | 49.84 | 67.98 |
| GPTQ | 4.125 | 73.80 | 65.68 | 79.27 | 97.2 | 58.61 | 78.36 |
| | 3.125 | 72.45 | 58.19 | 77.37 | 95.5 | 55.21 | 74.57 |
| | 2.125 | 54.93 | 24.67 | 54.46 | 75.1 | 31.77 | 37.79 |
| SSQR-1% | 4.445 | 74.43 | 66.78 | 79.65 | 96.9 | 59.18 | 78.93 |
| | 3.445 | 72.45 | 60.14 | 77.97 | 96.3 | 56.74 | 76.24 |
| | 2.445 | 52.80 | 23.07 | 58.49 | 74.1 | 33.25 | 40.05 |
| SSQR-2% | 4.765 | 73.80 | 67.21 | 79.49 | 97.2 | 58.94 | 78.53 |
| | 3.765 | 73.24 | 63.13 | 78.78 | 96.4 | 57.63 | 77.22 |
| | 2.765 | 54.30 | 27.08 | 61.04 | 82.5 | 38.41 | 50.41 |
| SSQR-3% | 5.085 | 72.93 | 67.38 | 79.54 | 96.9 | 59.64 | 79.07 |
| | 4.085 | 73.09 | 63.77 | 79.11 | 96.6 | 57.62 | 77.40 |
| | 3.085 | 54.54 | 26.15 | 58.81 | 83.6 | 38.34 | 49.52 |
| SSQR-4% | 5.405 | 73.24 | 66.95 | 79.92 | 96.9 | 59.32 | 79.06 |
| | 4.405 | 73.24 | 62.92 | 78.73 | 96.5 | 57.61 | 77.47 |
| | 3.405 | 54.54 | 29.95 | 54.95 | 82.3 | 39.80 | 51.87 |
| SSQR-5% | 5.725 | 74.03 | 67.91 | 80.52 | 97.2 | 59.49 | 79.39 |
| | 4.725 | 73.40 | 64.14 | 79.05 | 97.0 | 58.16 | 77.63 |
| | 3.725 | 64.25 | 42.59 | 72.58 | 88.7 | 49.94 | 68.20 |

## E.5 Comparison with Other Quantization Methods

We compare zero-shot WinoGrande and PiQA accuracies of our methods (HPTQ, SSQR) against GPTQ and state-of-the-art post-training, weight-only quantizers AQLM (Egiazarian et al., 2024), QuIP# (Tseng et al., 2024a), and QTIP (Tseng et al., 2024b) on Llama-2-7B. Results are reported in Table 16, sorted by average bitwidth. Metrics for AQLM, QuIP#, and QTIP are taken from their respective papers.

As shown in Table 16, for average bitwidth $\geq 4$, all methods yield accuracy close to the full-precision baseline. In the 3-4 bit regime, vanilla GPTQ falls behind recent methods; however, HPTQ and SSQR close this gap, bringing a scalar quantization approach to parity with vector quantization methods (AQLM, QuIP#, QTIP). In the 2-3 bit regime, HPTQ remains competitive with the state of the art.

Table 16: Comparing the zero-shot results of different quantization methods on Llama-2-7B.

| Method | Avg. Bitwidth | WinoGrande | PiQA |
|---|---|---|---|
| FP16 Baseline | 16.000 | 69.46 | 78.13 |
| AQLM | 5.020 | 67.40 | 78.29 |
| *SSQR-1%* | 4.445 | 68.82 | 78.35 |
| *HPTQ* | 4.125 | 69.61 | 77.75 |
| GPTQ | 4.125 | 68.82 | 77.97 |
| AQLM | 4.040 | 67.32 | 78.24 |
| QuIP# | 4.000 | 67.60 | 78.40 |
| QTIP | 4.000 | 67.10 | 78.40 |
| *SSQR-1%* | 3.445 | 65.43 | 77.15 |
| *HPTQ* | 3.125 | 67.72 | 77.80 |
| GPTQ | 3.125 | 64.96 | 73.88 |
| AQLM | 3.040 | 66.93 | 76.88 |
| QuIP# | 3.000 | 66.50 | 77.30 |
| QTIP | 3.000 | 66.90 | 78.10 |
| *SSQR-1%* | 2.445 | 50.04 | 56.15 |
| AQLM | 2.290 | 65.67 | 74.92 |
| *HPTQ* | 2.125 | 65.82 | 73.56 |
| GPTQ | 2.125 | 49.64 | 56.20 |
| AQLM | 2.020 | 65.67 | 74.76 |
| QuIP# | 2.000 | 64.90 | 75.10 |
| QTIP | 2.000 | 64.70 | 75.90 |

### E.6 Technical Details and Performance of SSQR's CUDA Kernel

The kernel is specialized for two regimes: in the low-batch regime, the kernel utilizes SIMT GPU cores exclusively, while tensor cores are utilized when batch size is $\geq 8$, the smallest outer dimension where tensor cores can be utilized without padding, and with 16-bit operands and 32-bit floating-point accumulators. For both regimes, sparse outliers are handled with SIMT cores.

To handle the dense inliers, we apply two reordering schemes here. First, the weights are reordered for memory movement involving tensor cores. Second, we apply an additional reordering scheme to enable batched conversion between 2-4-bit integers into their 16-bit counterparts.

To handle the sparse outliers, we group sparse outliers in groups of 16 rows (matching the outer tensor core dimension), then store them in column-major row order with padding to account for differences between non-zero counts across rows in the group.

Figure 5 shows the layer-wise speedup of the SSQR kernel on NVIDIA RTX 6000 GPU compared to the PyTorch BF16 matrix multiplication baseline across different layer shapes in the Qwen3-8B model (layers with the same input are merged), inlier bitwidths, outlier rates, and batch sizes. We observe the largest gains in the low-batch regime, with up to $4\times$ speedup when <1% outliers are present. As the outlier rate increases, the speedup diminishes, but the kernel consistently outperforms the BF16 baseline across all settings.

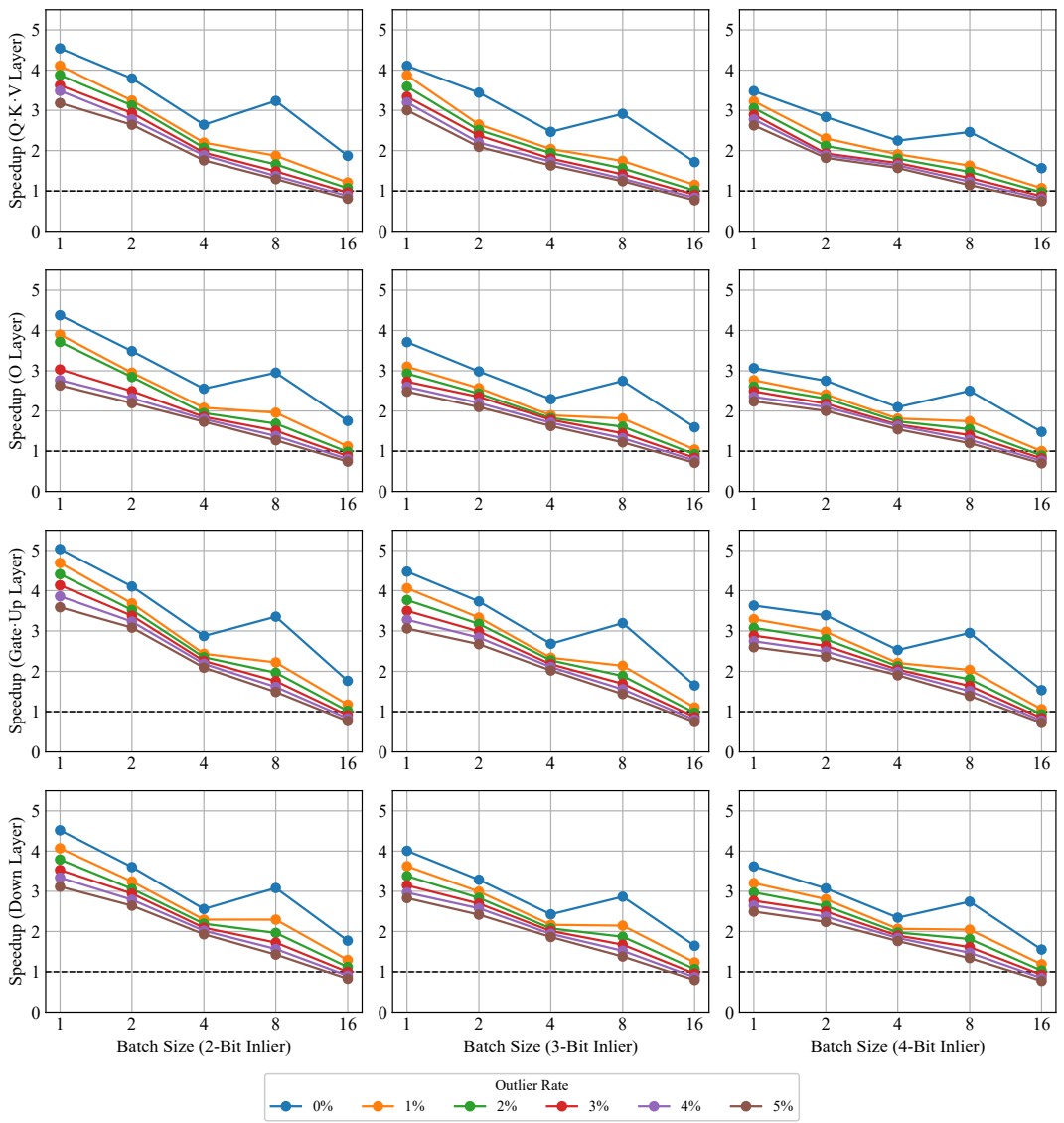

Figure 5: Layer-wise inference speedup of the SSQR kernel over the PyTorch BF16 baseline on Qwen3-8B across inlier bitwidths, outlier rates, and batch sizes on A6000 GPU.

## F LLM USAGE

LLM was used to aid and polish the writing of this paper, e.g., correcting grammar and rephrasing sentences.

