# OpenReview forum: "The Geometry of LLM Quantization: GPTQ as Babai's Nearest Plane Algorithm"
_ICLR.cc/2026/Conference — ICLR 2026 Poster_

### Official Review · Reviewer_PGNw · 2025-10-31

**Soundness:** 4
**Presentation:** 4
**Contribution:** 3
**Rating:** 8
**Confidence:** 3

**Summary:**

The paper shows how GPTQ, a well-established method for model weight quantization, is mathematically equivalent to Babai's nearest plane algorithm, a technique for tackling the closest vector problem on a lattice. In the case of GPTQ, the lattice is defined by the Hessian matrix of the layer's inputs. By leveraging the connection, the authors are able to apply the error guarantees from Babai's algorithm to the GPTQ algorithm, as well as give a geometric interpretation to its quantization steps.

**Strengths:**

The paper is well-written and easy to follow. I was familiar with the GPTQ algorithm and its usage in quantization, but not with the nearest vector on a lattice problem; I believe the authors were able to clearly establish the connection, and introduce both techniques in a clear and instructive way. Even if the connection might have limited algorithmic implications (in terms of improvements to GPTQ), I believe it still offers valuable insight into a widely-used quantization technique. I do not think the connection is necessarily obvious to the quantization community, and thus I believe this is a solid contribution and a helpful paper. A few specific comments worth highlighting:
- The paper is well-structured and the mathematical notation is clear and easy to follow.
- Figures 1-3 are very helpful in explaining the geometric interpretation of the algorithm.
- The authors highlight the importance of "no clipping" for full equivalence of the two algorithms, and provide solutions using the SpQR method for detecting outliers, and Huffman encoding for lossless compression.
- The authors helpfully provide an optimised kernel for the SSQR technique as part of their submission.

**Weaknesses:**

I view the paper's main contribution as the establishment of the theoretical connection between the two widely used algorithms. The algorithmic contributions (SSQR and HPTQ) are relatively incremental, as SSQR is a direct application of SpQR to the GPTQ method, and Huffman coding has indeed been applied to quantization problems previously (see e.g., https://arxiv.org/abs/1612.01543). This however is clearly indicated by the authors, and does not diminish the contribution of the paper.

**Questions:**

- Reading Algorithms 1-3 without the corresponding sections is a bit difficult — I wonder if captions detailing the procedures might help readability? (I leave this at authors' discretion)
- The authors provide an optimised kernel for SSQR. It would be useful to also highlight the implications of the Huffman encoding/decoding in terms of practical efficiency/speed.
- Minor: On line 158 the authors mention parameter "delta", but this is not defined/mentioned elsewhere so it wasn't quite clear to me as I wasn't deeply familiar with the LLL algorithm.

---

> ### Author Response · Authors · 2025-11-21
>
> We thank the reviewer for the encouraging review. We address the comments and answer the questions below. We mark edits to the manuscript in blue font.
>
> ***W. Algorithmic contributions***
>
> We agree that our primary contribution is the theoretical connection. SSQR and HPTQ are intended as practical illustrations of how the lattice view guides the design for improvements. In the revised manuscript, we explicitly cite this prior work on Huffman coding for network compression (arXiv:1612.01543).
>
> ***Q1. Algorithm details***
>
> We have made Algorithms 1-3 more self-contained by adding descriptive comments that state inputs, outputs, and the role of each step in the revised manuscript.
>
> ***Q2. Huffman encoding/decoding efficiency***
>
> We consider the Huffman encoding as a way to explore the compression limit of GPTQ. The encoding is done only once during the compression, and the decoding is done streamingly when it is being loaded into an integer tensor. This means it will be helpful in gaining speedups in extremely memory or I/O bound settings. We leave this as future work.
>
> ***Q3. The delta parameter in LLL***
>
> In LLL, the parameter $\\delta \\in \\left( \\frac{1}{4}, 1 \\right)$ controls how strict the Lovász condition is: larger $\\delta$ enforces stronger reduction, producing a more orthogonal (and shorter) basis but requiring more computation. It thus balances reduction quality versus runtime.
>
> A pseudocode of this algorithm with the parameter $\\delta$ can be conveniently found at
> [Lenstra–Lenstra–Lovász lattice basis reduction algorithm \- Wikipedia](https://en.wikipedia.org/wiki/Lenstra%E2%80%93Lenstra%E2%80%93Lov%C3%A1sz_lattice_basis_reduction_algorithm#LLL_algorithm_pseudocode).

---

> > ### Comment · Reviewer_PGNw · 2025-11-25
> >
> > I'd like to thank the authors for their response to my comments. As noted in my initial evaluation, I am quite positive about the overall contribution of the paper, and am happy to recommend it for acceptance.

---

### Official Review · Reviewer_ej6W · 2025-11-01

**Soundness:** 3
**Presentation:** 3
**Contribution:** 3
**Rating:** 8
**Confidence:** 3

**Summary:**

The authors identify the equivalence between GPTQ and Babai's nearest plane algorithm (Section 4.1 and 4.3), and derive a tight error bounds for GPTQ in the no-clipping regime (Section 4.4). Based on this understanding, they design two variants of GPTQ (SSQR and HPTQ; Section 5) with better accuracy than GPTQ and implement it in CUDA with considerable speedup.

**Strengths:**

1. Provides a geometric interpretation and theoretical guarantee for GPTQ (not quite due to order and clipping, but close enough!)
1. This is a bridge paper that connects the modern LLM quantization algorithms with the classical CVP literature
1. The mathematical proofs (both equivalence and error bound) are non-trivial
1. The CUDA inference kernels for SSQR and HPTQ have impressive speedup despite GPU-unfriendly designs such as outlier storage in CSR and Huffman coding.

**Weaknesses:**

1. Gap between theory and practice:
    1. People typically run GPTQ front-to-back, which is not consistent with the back-to-front order studied in the paper
    1. Clipping is inevitable in practice due to the limited range of quantized types, but the authors focus on the no-clipping scheme
    1. From my understanding, Babai's algorithm is most attractive when preceded by LLL/BKZ basis reduction, but GPTQ is equivalent to "Babai’s algorithm without basis reduction".
1. SSQR and HPTQ only has modest improvement over GPTQ, and far from SToA weight-only quantizers like [QTIP](https://arxiv.org/abs/2406.11235).
1. Insufficient experiments: all experiments are ran on Qwen3-8B, but ideally we want to see the performance across different scales and/or model families.

**Questions:**

1. You are claiming your bound (Theorem 5) is "tight". Can you construct a "hard case" that attains your error bound?
1. What's the counterpart for GPTQ for running Babai's algorithm with LLL/BKZ basis reduction?
1. Why did everyone suddenly starts to notice the equivalence between GPTQ and Babai’s algorithm? The authors mentioned a [concurrent work](https://arxiv.org/abs/2508.01077), and I assume that's not a coincidence?
    - You don't have to answer this if you are feeling uncomfortable; I'm just curious about how you discovered the connection between GPTQ and Babai's algorithm.

---

> ### Author Response · Authors · 2025-11-21
> **Official Comment by Authors (1/2)**
>
> We thank the reviewer for the thoughtful review and positive outlook. We address the raised points and answer the questions below. We mark edits to the manuscript in blue font.
>
> ***W1. Gap between theory and practice***
>
> The three differences the reviewer lists reflect mathematical choices rather than contradictions between theory and practice.
>
> 1\. Babai runs back-to-front by convention, whereas many GPTQ implementations iterate front-to-back or use act-order. These are related by a simple permutation. In practice, reversing the GPTQ loop or applying a reverse permutation to inputs/outputs reproduces the Babai iteration exactly; most inference kernels (e.g., ExLlamaV2 \[1\]) already accept an explicit ordering index (e.g., act-order index), so matching orders requires only flipping that index. Furthermore, the front-to-back and back-to-front orders do not differ much from a random order in practice. See the numerical results in the table below.
>
> 2\. Clipping in standard GPTQ is used to force outliers into a fixed grid, but it invalidates the preconditions of Babai's error guarantees and could make the **layer-wise error potentially unbounded**. In practice, there are quantization methods that avoid clipping, so the error bound is operational in the intended regime. Our proposed two formats, **SSQR** and **HPTQ**, are designed to **avoid clipping** while retaining low effective bit-rates and high accuracies.
> Second, we emphasize that the state-of-the-art 4-bit floating-point formats (e.g., MXFP4 and NVFP4) \[2\] are essentially **no-clipping**: since they use very small quantization groups (32 and 16, respectively), the near-optimal choice of scale is AbsMax per-group, which leads to no weight being clipped. As such, our analysis would apply directly to these formats.
>
> 3\. As formalized in Theorem 5, Babai's algorithm **without** basis reduction already yields a layer-wise error bound that is stronger than simple round-to-nearest (RTN).
>
> We also discussed in Appendix A.2 that applying Babai for GPTQ quantization on a **reduced** basis would require computing an LLL/BKZ transform **for each output channel** (because the reduction depends on the per-group scales), then quantizing in the reduced basis and mapping back to the original basis. This breaks GPTQ's batched reuse and introduces substantial overhead: The complexity of GPTQ itself is cubic in the number of input dimensions, whereas a single LLL run is quartic (with optimized implementations) and BKZ is exponential in its block size.
>
> We were actually interested in this point as well in our investigation, and tested LLL empirically. We simplify the setting by choosing the group size equal to the number of input channels so that the LLL step can be shared across the output channels. For comparison with different orderings, we use the same AbsMax scale without clipping. On the layers we evaluated (Qwen3-8B block 18), LLL provided only **marginal L2-loss improvements** relative to strong ordering heuristics (see the table below). Moreover, LLL is not guaranteed to be optimal in general; on some layers, it underperforms a well-chosen ordering.
>
> | Layer | q\_proj | k\_proj | v\_proj | o\_proj | gate\_proj | up\_proj | down\_proj |
> | :---- | ----- | ----- | ----- | ----- | ----- | ----- | ----- |
> | front-to-back order | 0.00437 | 0.00495 | 0.00475 | 0.00762 | 0.00502 | 0.00504 | 0.07691 |
> | back-to-front order | 0.00434 | 0.00491 | 0.00478 | 0.00769 | 0.00508 | 0.00510 | 0.07675 |
> | random order (mean) | 0.00431 | 0.00507 | 0.00480 | 0.00775 | 0.00507 | 0.00508 | 0.07687 |
> | act-order | 0.00286 | 0.00324 | 0.00316 | 0.00746 | 0.00287 | 0.00287 | 0.07110 |
> | min-pivot order | 0.00284 | 0.00321 | 0.00313 | 0.00740 | 0.00284 | 0.00285 | 0.07102 |
> | LLL basis reduction | 0.00283 | 0.00321 | 0.00313 | 0.00738 | 0.00290 | 0.00291 | 0.07134 |

---

> ### Author Response · Authors · 2025-11-21
> **Official Comment by Authors (2/2)**
>
> ***W2. Comparison with other methods***
>
> This paper is primarily **theoretical**, showing a new and non-trivial connection between two very different areas of computer science. It establishes a precise equivalence between GPTQ and **Babai's nearest plane algorithm**, yielding (i) an intuitive geometric account of GPTQ's error propagation and (ii) a **tight** layer-wise error bound in the no-clipping regime. We are not claiming (or aiming) to break SOTA quantization methods.
>
> The point of the experiments is to show that our connection can yield improvements to the GPTQ algorithm that are **drop-in** for popular GPTQ toolchains. The better quantization orders (min-pivot) and no-clipping representations (SSQR/HPTQ) also apply to GPTQ-based methods like QuIP, QuaRot, and SpinQuant, where GPTQ is used as a subroutine. Additionally, our insights (e.g., the ordering guidance) can be transferred to GPTQ/LDLQ-based methods such as QuIP\# and QTIP, where the weights are updated block-wise using the second-order information, which can be considered as a blocked variant of GPTQ.
>
> For comparison, we have added **Appendix Section D.6** to compare our methods (HPTQ and SSQR) with the **AQLM**, **QuIP\#**, and **QTIP** results on the Llama-2-7B model in the revised manuscript. For average bitwidth $\\ge$4, all methods yield accuracy close to the full-precision baseline. In the 3-4 bit regime, vanilla GPTQ falls behind recent methods; however, HPTQ and SSQR close this gap, bringing a scalar quantization approach to parity with vector quantization methods (AQLM, QuIP\#, QTIP). In the 2-3 bit regime, HPTQ remains competitive with the strong baselines.
>
> ***W3. Evaluations on other models***
>
> In our initial submission, we already have Qwen3 evaluation results in the **sizes** of 0.6B, 1.7B, 4B, 8B, and 14B in Figure 4(b) and Appendix D.3.
>
> **Additional models.** In the revised manuscript, we have added Appendix Section D.5 that includes **Llama-3.2-3B-Instruct**, **Llama-3.1-8B-Instruct**, and **Llama-2-7B** results using the same evaluation protocol. The resulting trends are similar in terms of accuracy improvements.
>
> ***Q1. Hard case to attain the error bound***
>
> We can construct a hard case as follows.
> Take identity calibration activation $\boldsymbol{X}=\mathbf{I}$, identity permutation $\boldsymbol{T}=\mathbf{I}$, and all-one scale $\boldsymbol{s}\_{i} = \mathbf{1}$. The Hessian is $\boldsymbol{X}^\top \boldsymbol{X} = \mathbf{I}$, so the error propagation is always $0$, and GPTQ/Babai degenerates to integer RTN.
> If the weight $\boldsymbol{w}\_{i} - \frac{1}{2} \in \mathbb{Z}^{c}$, i.e., the weights are integers plus $\frac{1}{2}$, every element will then have a quantization error of $\frac{1}{2}$, and the L2 loss of $\boldsymbol{X}\boldsymbol{w}_{i}$ is $\left\\| \boldsymbol{X} \operatorname{diag} \left( \boldsymbol{s}\_{i} \right) \boldsymbol{z}\_{i} - \boldsymbol{X} \boldsymbol{w}\_{i} \right\\|^{2} = \sum\_{j=1}^c \left( \frac{1}{2} \right)^{2} = \frac{c}{4}$.
> $\boldsymbol{D}$ is the diagonal matrix of the LDL decomposition of $\boldsymbol{T}^\top \boldsymbol{X}^\top \boldsymbol{X} \boldsymbol{T} = \mathbf{I}$, so $\boldsymbol{D} = \mathbf{I}$ .
> The L2 loss matches the worst-case bound $\frac{1}{4} \left( \boldsymbol{T}^{-1} \boldsymbol{s}\_{i} \right)^\top \boldsymbol{D} \left( \boldsymbol{T}^{-1} \boldsymbol{s}\_{i} \right) = \frac{1}{4} \left( \mathbf{I}^{-1} \mathbf{1} \right)^\top \mathbf{I} \left( \mathbf{I}^{-1} \mathbf{1} \right) = \frac{1}{4} \mathbf{1}^\top \mathbf{1} = \frac{c}{4}$ exactly.
>
> ***Q2. GPTQ counterpart to Babai+LLL/BKZ***
>
> Please refer to the answer in W1 point 3\.
>
> ***Q3. Concurrent work***
>
> We are the first to discover the equivalence. We think this is a pure coincidence, and we were surprised when we saw the concurrent work. Compared to the concurrent work, our paper is more complete and additionally includes an analysis of quantization ordering, practical no-clipping schemes with kernels, and numerical evaluation results.
>
> ***References***
>
> \[1\] ExLlamaV2. URL [https://github.com/turboderp-org/exllamav2](https://github.com/turboderp-org/exllamav2).
> \[2\] Vage Egiazarian, Roberto L. Castro, Denis Kuznedelev, Andrei Panferov, Eldar Kurtic, Shubhra Pandit, Alexandre Marques, Mark Kurtz, Saleh Ashkboos, Torsten Hoefler, and Dan Alistarh. Bridging the gap between promise and performance for microscaling fp4 quantization, 2025\. URL [https://arxiv.org/abs/2509.23202](https://arxiv.org/abs/2509.23202).

---

### Official Review · Reviewer_fgCD · 2025-11-01

**Soundness:** 3
**Presentation:** 3
**Contribution:** 2
**Rating:** 4
**Confidence:** 4

**Summary:**

This paper establishes a formal equivalence between the GPTQ algorithm and Babai's nearest plane algorithm from lattice theory. This connection provides a theoretical foundation for a method that has thus far been understood primarily as a sequence of heuristic, greedy updates. The core theoretical insight is that when GPTQ is executed in a back-to-front order, its process of quantizing a weight and propagating the error is mathematically identical to Babai's algorithm solving the Closest Vector Problem (CVP). Motivated by the new foundation, they propose a new method: Huffman-encoded post-training quantization (HPTQ).

**Strengths:**

1. The paper's primary contribution is establishing a rigorous mathematical equivalence between the GPTQ quantization algorithm and Babai's nearest plane algorithm from lattice theory.

2. The theoretical foundation leads to practicaly improved method, including a formal error bound for GPTQ and the design of new, improved quantization methods like HPTQ. These methods demonstrably outperform the original GPTQ.

3. The presenation is clear and well-written.

**Weaknesses:**

1.  The experimental evaluation primarily benchmarks the proposed methods (HPTQ, SSQR) against GPTQ and Round-to-Nearest (RTN). They didn't compare with strong baseline method like QuaRot, QuIP, SpinQuant, OmniQuant. Although GPTQ is a well known quantization method, without comparing with other recent, strong baselines limits the practical value of this work.

2. The paper derives a worst-case error bound, but its utility is questionable. The bound is only valid in the no-clipping regime, which is not the standard use case. Besides, the proposed "min-pivot" ordering is shown to provide only a modest practical improvement over the much cheaper "act-order" heuristic.

3. The paper introduces non-trivial computational costs without a thorough analysis. HPTQ requires a binary search over scales to hit a target Huffman bitrate, and SSQR requires a similar search to meet an outlier budget.

**Questions:**

NA

---

> ### Author Response · Authors · 2025-11-21
> **Official Comment by Authors (1/2)**
>
> We thank the reviewer for the positive assessment of our presentation and for the constructive feedback. We address each concern below and mark edits to the manuscript in blue font.
>
> ***W1. Comparison with other methods***
>
> This paper is primarily **theoretical**, showing a new and non-trivial connection between two very different areas of computer science. It establishes a precise equivalence between GPTQ and **Babai's nearest plane algorithm**, yielding (i) an intuitive geometric account of GPTQ's error propagation and (ii) a **tight** layer-wise error bound in the no-clipping regime. We are not claiming (or aiming) to break SOTA quantization methods.
>
> The point of the experiments is to show that our connection can yield improvements to the GPTQ algorithm that are **drop-in** for popular GPTQ toolchains. The better quantization orders (min-pivot) and no-clipping representations (SSQR/HPTQ) also apply to GPTQ-based methods like QuIP, QuaRot, and SpinQuant, where GPTQ is used as a subroutine. Additionally, our insights (e.g., the ordering guidance) can be transferred to GPTQ/LDLQ-based methods such as QuIP\# and QTIP, where the weights are updated block-wise using the second-order information, which can be considered as a blocked variant of GPTQ.
>
> For comparison, we have added **Appendix Section D.6** to compare our methods (HPTQ and SSQR) with the **AQLM**, **QuIP\#**, and **QTIP** results on the Llama-2-7B model in the revised manuscript. For average bitwidth $\\ge$4, all methods yield accuracy close to the full-precision baseline. In the 3-4 bit regime, vanilla GPTQ falls behind recent methods; however, HPTQ and SSQR close this gap, bringing a scalar quantization approach to parity with vector quantization methods (AQLM, QuIP\#, QTIP). In the 2-3 bit regime, HPTQ remains competitive with the strong baselines.
>
> Our evaluation focus is on the **one-shot** **weight-only** quantization regime (like GPTQ/RTN). The requested baselines, SpinQuant and OmniQuant, involve **data-dependent tuning** beyond one-shot, which are not fair comparisons to our methods. Also, QuaRot and SpinQuant are **Weight-Activation-KV** quantization methods and are not suitable for comparison to our weight-only quantization methods.
>
> ***W2. Worst-case error bound and min-pivot order***
>
> Clipping in standard GPTQ is used to force outliers into a fixed grid, but it invalidates the preconditions of Babai's error guarantees and could make the **layer-wise error potentially unbounded**. In practice, there are quantization methods that avoid clipping, so the error bound is operational in the intended regime. Our proposed two formats, **SSQR** and **HPTQ**, are designed to **avoid clipping** while retaining low effective bit-rates and high accuracies.
> Second, we emphasize that the state-of-the-art 4-bit floating-point formats (e.g., MXFP4 and NVFP4) \[1\] are essentially **no-clipping**: since they use very small quantization groups (32 and 16, respectively), the near-optimal choice of scale is AbsMax per-group, which leads to no weight being clipped. As such, our analysis would apply directly to these formats.
>
> As an **additional note**, the worst-case guarantee also meaningfully controls typical behavior: in Appendix C.2, we prove that, under no-clipping and some smooth distributions, the expected absolute layer-wise error is exactly $\\frac{1}{3}$ of the worst-case bound.
>
> The **min-pivot** order is more principled and more robust under ill-conditioned Hessians compared to the act-order heuristic. Although the models we evaluated are largely well-conditioned, other settings may be ill-conditioned and thus benefit more from min-pivot. Crucially, min-pivot does not change GPTQ's asymptotic complexity, and substituting it for act-order only adds a moderate overhead to the time. We detail the computational cost in the next paragraph.

---

> ### Author Response · Authors · 2025-11-21
> **Official Comment by Authors (2/2)**
>
> ***W3. Computational cost***
>
> We have created Triton kernels for GPTQ and several other components, and provided them in the supplementary materials. The **GPTQ** computation itself is fast, and the overall time is bottlenecked by propagating the activations through the layers and calculating the Hessian matrices. **HPTQ** only requires a few iterations of GPTQ in the binary search, which only leads to a marginal increase in time. We have not implemented the Triton kernel for SSQR, but its run time will be significantly reduced once a kernel is produced. Even without the kernel, **SSQR** takes only about an hour to quantize an 8B model on an A6000 GPU, which is still fast for post-training quantization.
>
> We measure the time for each component on an A6000 GPU and provide the table below for the comparisons. The time is summed over the same component for all layers within a **transformer block** of Qwen3-8B. The table does not include the pre-processing (e.g., tokenization) and the post-processing (e.g., packing), considering which would make the overall time difference of different methods even less significant.
>
> | Component | Time \[sec\] |
> | :---- | ----- |
> | *Order* |  |
> | act-order | 0.002 |
> | min-pivot (Triton) | 15.180 |
> | *Quantization* |  |
> | GPTQ (Triton) | 0.741 |
> | HPTQ (Triton) | 3.115 |
> | SSQR | 95.502 |
> | *Others* |  |
> | Activation, Hessian (Triton), etc. | 10.316 |
>
> ***References***
>
> \[1\] Vage Egiazarian, Roberto L. Castro, Denis Kuznedelev, Andrei Panferov, Eldar Kurtic, Shubhra Pandit, Alexandre Marques, Mark Kurtz, Saleh Ashkboos, Torsten Hoefler, and Dan Alistarh. Bridging the gap between promise and performance for microscaling fp4 quantization, 2025\. URL [https://arxiv.org/abs/2509.23202](https://arxiv.org/abs/2509.23202).

---

### Official Review · Reviewer_64UP · 2025-11-01

**Soundness:** 3
**Presentation:** 3
**Contribution:** 3
**Rating:** 6
**Confidence:** 5

**Summary:**

The paper considers the GPTQ algorithm, and mentioned how running it back-to-front is identical to Babai’s nearest-place algorithm. The author show several theoretical results and also propose practical schemes for algos that utilize these insights.

**Strengths:**

1. The intro of the paper is clearly written
2. Algos 1 and 2 paint a clear picture, although GPTQ is a little more involved than what Algo 1 depicts.
3. The theoretical results are solid, i went over the theorems and corollaries. The equivalence b/w GPTQ and Babai’s nearest-plane is clean.
4. The experiments are conducted on the Qwen3 series of models, representing near-state of the art models.
5. HPTQ seems to be competitive in experiments

**Weaknesses:**

1. I dont agree with the statement that GPTQ’s inner workings are “adhoc”. GPTQ is inspired from the Optimal Brain Surgeon/Compression literature (OBS). OBS is quite principled and its update can be derived using KKT conditions, making it quite rigorous, Except the optimization bit for GPTQ which is quite sloppy, GPTQ is a pretty grounded method.
2. The role of clipping is a bit confusing, maybe required in practice. Although the results on Qwen3 are good, would be good to extend to other model families.
3. I am not sure if the experiment results offer something practical. I am confused about the main contribution of this paper. Are you claiming that this is a theoretical paper? Or are you claiming that there is a better version of GPTQ? Because several papers have already practically bettered GPTQ

**Questions:**

See weaknesses

---

> ### Author Response · Authors · 2025-11-21
>
> We thank the reviewer for the positive assessment of our exposition, theory, and experiments. We respond to the concerns below and mark edits to the manuscript in blue font.
>
> ***Clarification to Algorithm 1***
>
> As noted in Section 3.1, Algorithm 1 is identical to the pseudocode in the original GPTQ paper \[1\] except for missing the blocking mechanism that only affects the memory access pattern and computational speed, but not the **numerical** results. We have also added more descriptive comments to our pseudocodes in the updated manuscript to make them easier to read.
>
> ***W1. On the "ad-hoc" characterization of GPTQ***
>
> Our intent was definitely not to dismiss GPTQ/OBS as unprincipled. Each **local** (one-coordinate) update step in OBS/GPTQ is well-motivated (e.g., via second-order reasoning) and minimizes the loss of the **current step**. Our statement was meant to highlight the lack of **global guarantees**. Weight quantization with a layerwise objective is equivalent to a CVP instance, which is NP-hard. The sequential local-optimal update schemes of OBS/GPTQ are not global-optimal in general and do not automatically provide **layer-wise global error guarantees**. Our contribution is to supply such a guarantee by proving the exact geometric equivalence to Babai's nearest plane and importing Babai's error bound.
>
> In brief, we agree that the term "ad-hoc" might be confusing. We have **removed it** in the revised manuscript.
>
> ***W2. Clipping in practice; results on other models***
>
> Clipping in standard GPTQ is used to force outliers into a fixed grid, but it invalidates the preconditions of Babai's error guarantees and could make the **layer-wise error potentially unbounded**. In practice, there are quantization methods that avoid clipping, so the error bound is operational in the intended regime. Our proposed two formats, **SSQR** and **HPTQ**, are designed to **avoid clipping** while retaining low effective bit-rates and high accuracies.
>
> Second, we emphasize that the state-of-the-art 4-bit floating-point formats (e.g., MXFP4 and NVFP4) \[2\] are essentially **no-clipping**: since they use very small quantization groups (32 and 16, respectively), the near-optimal choice of scale is AbsMax per-group, which leads to no weight being clipped. As such, our analysis would apply directly to these formats.
>
> **Additional models.** In the revised manuscript, we have added Appendix Section D.5 that includes **Llama-3.2-3B-Instruct**, **Llama-3.1-8B-Instruct**, and **Llama-2-7B** results using the same evaluation protocol. The resulting trends are similar in terms of accuracy improvements.
>
> ***W3. Contribution of this paper***
>
> This paper is primarily **theoretical**, showing a new and non-trivial connection between two very different areas of computer science. It establishes a precise equivalence between GPTQ and **Babai's nearest plane algorithm**, yielding (i) an intuitive geometric account of GPTQ's error propagation and (ii) a **tight** layer-wise error bound in the no-clipping regime. We are not claiming (or aiming) to break SOTA quantization methods.
>
> The point of the experiments is to show that our connection can yield improvements to the GPTQ algorithm that are **drop-in** for popular GPTQ toolchains. The better quantization orders (min-pivot) and no-clipping representations (SSQR/HPTQ) also apply to GPTQ-based methods like QuIP, QuaRot, and SpinQuant, where GPTQ is used as a subroutine. Additionally, our insights (e.g., the ordering guidance) can be transferred to GPTQ/LDLQ-based methods such as QuIP\# and QTIP, where the weights are updated block-wise using the second-order information, which can be considered as a blocked variant of GPTQ.
>
> ***References***
>
> \[1\] Elias Frantar, Saleh Ashkboos, Torsten Hoefler, and Dan Alistarh. OPTQ: Accurate quantization for generative pre-trained transformers. In The Eleventh International Conference on Learning Representations, 2023\. URL [https://openreview.net/forum?id=tcbBPnfwxS](https://openreview.net/forum?id=tcbBPnfwxS).
> \[2\] Vage Egiazarian, Roberto L. Castro, Denis Kuznedelev, Andrei Panferov, Eldar Kurtic, Shubhra Pandit, Alexandre Marques, Mark Kurtz, Saleh Ashkboos, Torsten Hoefler, and Dan Alistarh. Bridging the gap between promise and performance for microscaling fp4 quantization, 2025\. URL [https://arxiv.org/abs/2509.23202](https://arxiv.org/abs/2509.23202).

---

### Author Response · Authors · 2025-12-03
**General Response to All Reviewers and ACs**

We are extremely grateful to all reviewers and ACs for taking the time to review this paper, and we thank all reviewers for their valuable feedback. The paper receives mostly positive ratings and reviews. We have addressed every concern in the individual response to each reviewer. We provide this general comment as a summary of the common concerns by the reviewers and our responses to these common concerns.

***1\. Meaning of the experiments; evaluation on additional models and comparison to other methods***

This paper is primarily **theoretical**, showing a new and non-trivial connection between two very different areas of computer science. It establishes a precise equivalence between **GPTQ** and **Babai's nearest plane algorithm**, yielding (i) an intuitive geometric account of GPTQ's error propagation and (ii) a **tight** layer-wise error bound in the no-clipping regime. We are not claiming (or aiming) to break SOTA quantization methods.

The point of the experiments is to show that our connection can yield improvements to the GPTQ algorithm that are **drop-in** for popular GPTQ toolchains. The improved quantization order (min-pivot) and no-clipping representations (SSQR/HPTQ) also apply to GPTQ-based methods like QuIP, QuaRot, and SpinQuant, where GPTQ is used as a subroutine. Additionally, our insights (e.g., the ordering guidance) can be transferred to GPTQ/LDLQ-based methods such as QuIP\# and QTIP, where the weights are updated block-wise using the second-order information, which can be considered as a blocked variant of GPTQ.

To fully address the reviewers' concern, we have added Sections D.5 and D.6 in the appendix and marked major modifications in blue font in the revised manuscript.

**Appendix Section D.5:** Evaluation results on **additional models** (Llama-3.2-3B-Instruct, Llama-3.1-8B-Instruct, and Llama-2-7B) are added. The resulting trends are similar to the Qwen3 model family metrics in our initial submission in terms of accuracy improvements.

**Appendix Section D.6:** Comparison to **additional quantization methods** (AQLM, QuIP\#, and QTIP) is added. We compare our proposed methods (HPTQ and SSQR) with those methods on the Llama-2-7B model. For average bitwidth $\\ge$4, all methods yield accuracy close to the full-precision baseline. In the 3-4 bit regime, vanilla GPTQ falls behind recent methods; however, HPTQ and SSQR close this gap, bringing a scalar quantization approach to parity with vector quantization methods (AQLM, QuIP\#, and QTIP). In the 2-3 bit regime, HPTQ remains competitive with the state of the art.

***2\. The role of clipping in theory and practice***

Clipping in standard GPTQ is used to force outliers into a fixed grid, but it invalidates the preconditions of Babai's error guarantees and could make the **layer-wise error potentially unbounded**. In practice, there are quantization methods that avoid clipping, so the error bound is operational in the intended regime. Our proposed two formats, **SSQR** and **HPTQ**, are designed to **avoid clipping** while retaining low effective bit-rates and high accuracies. Second, we emphasize that the state-of-the-art 4-bit floating-point formats (e.g., MXFP4 and NVFP4) \[1\] are essentially **no-clipping**: since they use very small quantization groups (32 and 16, respectively), the near-optimal choice of scale is AbsMax per-group, which leads to no weight being clipped. As such, our analysis would apply directly to these formats.

***References***

\[1\] Vage Egiazarian, Roberto L. Castro, Denis Kuznedelev, Andrei Panferov, Eldar Kurtic, Shubhra Pandit, Alexandre Marques, Mark Kurtz, Saleh Ashkboos, Torsten Hoefler, and Dan Alistarh. Bridging the gap between promise and performance for microscaling fp4 quantization, 2025\. URL [https://arxiv.org/abs/2509.23202](https://arxiv.org/abs/2509.23202).

---

### Meta-Review · Area_Chair_PwXW · 2026-01-01

**Summary:**

Reviewers’ concerns focused on three main points: (1) clarification of the paper’s intended contribution, as it was initially unclear whether the work aimed to propose a new state-of-the-art quantization method or a primarily theoretical analysis; (2) the practical relevance of the derived error bound, which holds under a no-clipping assumption that differs from standard GPTQ usage; and (3) the scope of experimental evaluation, including comparisons to stronger recent baselines, coverage of additional model families, and clarification of computational overhead.

These concerns were addressed in the rebuttal and revision by clearly positioning the paper as a theory-first contribution, removing potentially misleading language, justifying the relevance of the no-clipping regime (including connections to modern quantization formats), and adding additional experiments, baselines, and runtime discussion. With these clarifications and additions, the reviewers’ concerns were largely resolved and informed a positive recommendation.

**Reviewer Concerns:**

Concerns addressed by the rebuttal and revision: (1) Unclear contribution scope (theory vs. practice): The rebuttal clearly positions the paper as a theory-first contribution, removes potentially misleading language (e.g., describing GPTQ as “ad-hoc”), and clarifies that the experiments are meant to demonstrate how the theory informs principled extensions rather than to claim SOTA performance. (2) Role of clipping and validity of the error bound: The authors convincingly explain that clipping violates the assumptions behind Babai’s guarantees, justify the relevance of the no-clipping regime in practice, and propose SSQR and HPTQ as concrete mechanisms to make the theory operational. Additional discussion further motivates the usefulness of the bound beyond worst-case analysis. (3) Limited experimental coverage and missing baselines: The revised manuscript adds evaluations on additional model families and comparisons to stronger recent baselines, and provides clarification of computational overhead, addressing the main experimental concerns.

Concerns partially or still outstanding: (1) While the added experiments show that the proposed methods are competitive, the practical improvements over existing methods remain modest in some regimes. However, this is consistent with the paper’s clarified theoretical focus. (2)  The error bound remains formally limited to the no-clipping setting, and its applicability to heavily clipped or highly non-uniform quantization regimes is not addressed. This is a known and acknowledged limitation rather than an unresolved issue.

Overall, no major reviewer concerns remain unaddressed, and the remaining limitations are aligned with the paper’s stated scope.

**Reviewer Scores:**

1. Reviewer 64UP (initial rating: 6 – marginally above acceptance threshold):
Likely increase to 8 or unchanged. The rebuttal directly addressed concerns about contribution scope, clarified the role of theory vs. practice, removed confusing language, and added results on additional model families. These changes resolve the reviewer’s main sources of hesitation.


2. Reviewer fgCD (initial rating: 4 – marginally below acceptance threshold):
Likely increase to 6. The added comparisons to stronger baselines, expanded experiments, clarification of computational cost, and clearer justification of the no-clipping regime substantially address this reviewer’s primary weaknesses, though enthusiasm may remain tempered by the modest practical gains

---

### Decision · Program_Chairs · 2026-01-26

Accept (Poster)